# PROVABLY EFFICIENT POLICY-REWARD CO-PRETRAINING FOR ADVERSARIAL IMITATION LEARNING

## ABSTRACT

Adversarial imitation learning (AIL) achieves superior expert sample efficiency compared to behavioral cloning (BC) but requires extensive online environment interactions. Recent empirical works have attempted to mitigate this limitation by augmenting AIL with BC—for instance, initializing AIL algorithms with BC-pretrained policies. Despite certain empirical successes, systematic theoretical analysis of the provable efficiency gains remains lacking. This paper provides rigorous theoretical guarantees and develops effective algorithms to accelerate AIL. First, we develop a theoretical analysis for AIL with policy pretraining alone, revealing a critical but theoretically unexplored limitation: the absence of reward pretraining. Building on this insight, we derive a principled reward pretraining method grounded in reward-shaping-based analysis. Crucially, our analysis reveals a fundamental connection between the expert policy and shaping reward, naturally giving rise to CoPT-AIL, an approach that jointly pretrains policies and rewards through a single BC procedure. Theoretical results demonstrate that CoPT-AIL achieves an improved imitation gap bound compared to standard AIL without pretraining, providing the first theoretical guarantee for the benefits of pretraining in AIL. Experimental evaluation confirms CoPT-AIL's superior performance over prior AIL methods.

## 1 INTRODUCTION

Imitation learning (IL) (Argall et al., 2009; Osa et al., 2018) is an essential technique in artificial intelligence that enables machines to learn complex behaviors by mimicking expert demonstrations. This approach has achieved significant success across diverse domains, including autonomous driving (Pan et al., 2017), generalist robot learning (Brohan et al., 2023; Mees et al., 2024), and language modeling (Brown et al., 2020).

IL comprises two primary methodological categories: behavioral cloning (BC) and adversarial imitation learning (AIL). BC represents an offline approach that directly applies supervised learning to learn policies from demonstrations (Pomerleau, 1991; Ross et al., 2011; Brantley et al., 2020). While conceptually straightforward, BC is vulnerable to compounding errors (Syed & Schapire, 2010), resulting in poor expert sample efficiency. In contrast, AIL (Abbeel & Ng, 2004; Syed & Schapire, 2007; Ho & Ermon, 2016; Kostrikov et al., 2019) seeks to match the expert's state-action distribution through a minimax optimization framework. The method alternates between recovering an adversarial reward function that maximizes the policy value gap between expert and learner, and updating the policy to minimize this gap. Since this optimization typically requires online environment interactions, AIL is classified as an online method. Both theoretical analysis (Rajaraman et al., 2020; Xu et al., 2020) and empirical evidence (Ho & Ermon, 2016; Kostrikov et al., 2019; Ghasemipour et al., 2019) demonstrate that AIL effectively mitigates BC's compounding error problem, achieving superior expert sample efficiency.

While AIL demonstrates high sample efficiency with expert demonstrations, its reliance on extensive online environment interactions presents a significant limitation (Ho & Ermon, 2016). To mitigate this limitation, researchers have explored various approaches to combine AIL with BC (Jena et al., 2021; Orsini et al., 2021; Haldar et al., 2023; Watson et al., 2023; Yue et al., 2024). The most intuitive

approach involves pretraining policies using BC, then finetuning them with AIL through online interactions (Ho & Ermon, 2016). However, empirical studies consistently show that this strategy provides minimal benefits (Sasaki et al., 2018; Jena et al., 2021; Orsini et al., 2021; Yue et al., 2024). The pretrained policy's performance typically degrades during early AIL training, negating most advantages from the initial BC phase.

To overcome this limitation, several alternative integration strategies have emerged. Some approaches augment the AIL objective with BC regularization terms (Jena et al., 2021; Haldar et al., 2023), while others learn additional reward functions using either prior policies (Watson et al., 2023) or supplementary datasets (Yue et al., 2024). Despite the empirical successes in certain scenarios, there remains a notable absence of systematic theoretical studies, particularly in terms of *imitation gap* (i.e., performance difference between the expert and learner), which may hinder deep understanding and impede future algorithmic advances.

This paper aims to bridge the gap between theory and practice by providing rigorous theoretical guarantees and developing effective algorithms to accelerate AIL. Our key contributions are threefold.

- First, we develop a theoretical analysis for AIL with policy pretraining alone, establishing a sharp theoretical baseline. Specifically, we prove that while policy pre-training effectively reduces the cumulative policy error, the specific reward error induced by random reward initialization remains a persistent bottleneck. This rigorous diagnosis serves a dual purpose: it mathematically isolates the reward error as the precise target for our algorithmic design, and establishes a tight theoretical baseline for the subsequent theoretical comparison.

- Guided by this theoretical diagnosis, we derive a principled reward pretraining method, grounded in reward shaping theory (Ng et al., 1999). We prove that inferring a shaping reward—rather than the original true reward—is already sufficient to reduce reward error, thereby circumventing the reward ambiguity issue. Crucially, our analysis reveals a fundamental connection between the expert policy and shaping reward, naturally giving rise to the approach of jointly pretraining policies and rewards through a single BC procedure. This yields our complete algorithm CoPT-AIL, **AIL** with Policy-Reward **Co**-**Pre**training.

- Finally, we provide a rigorous theoretical analysis demonstrating CoPT-AIL's superiority over prior AIL approaches. Our theoretical results show that CoPT-AIL can provably reduce reward error through reward pretraining, achieving an improved imitation gap bound compared to standard AIL without pretraining under mild assumptions. To our best knowledge, this represents the first theoretical guarantee for the efficiency gains of pretraining in AIL. Experimental evaluation confirms CoPT-AIL's superior performance over existing methods.

## 2 PRELIMINARIES

**Markov Decision Process.** We consider episodic Markov Decision Processes (MDPs) represented by the tuple $\mathcal{M} = (\mathcal{S}, \mathcal{A}, P, r^\star, H, \rho)$, where $\mathcal{S}$ and $\mathcal{A}$ denote the state and action spaces, respectively, $H$ is the planning horizon, and $\rho$ is the initial state distribution. Following (Ross & Bagnell, 2010), we consider the non-stationary transition dynamics characterized by $P = \{P_1, \ldots, P_H\}$, where $P_h(s_{h+1}|s_h, a_h)$ gives the probability of transitioning to state $s_{h+1}$ from state $s_h$ upon taking action $a_h$ at step $h \in [H]$. The non-stationary reward is defined by $r^\star = \{r_1^\star, \ldots, r_H^\star\}$, where without loss of generality, $r_h^\star : \mathcal{S} \times \mathcal{A} \to [0, 1]$ for all $h \in [H]$.

A non-stationary policy $\pi = \{\pi_1, \ldots, \pi_H\}$ maps states to action distributions, with $\pi_h : \mathcal{S} \to \Delta(\mathcal{A})$, where $\Delta(\mathcal{A})$ denotes the probability simplex over actions. Here, $\pi_h(a|s)$ represents the probability of selecting action $a$ in state $s$ at step $h$.

The interaction protocol proceeds as follows: each episode begins with the environment sampling an initial state $s_1 \sim \rho$. At each step $h$, the agent observes state $s_h$, selects action $a_h \sim \pi_h(\cdot|s_h)$, receives reward $r_h^\star(s_h, a_h)$, and transitions to the next state $s_{h+1} \sim P_h(\cdot|s_h, a_h)$. The episode terminates after $H$ steps.

We evaluate policy performance using the expected cumulative reward:

$$V^\pi := \mathbb{E}\left[\sum_{h=1}^{H} r_h^\star(s_h, a_h)\middle| a_h \sim \pi_h(\cdot|s_h), s_{h+1} \sim P_h(\cdot|s_h, a_h), \forall h \in [H]\right].$$

The Q-function is defined as $Q_h^\pi(s, a) := \mathbb{E}\left[\sum_{h'=h}^{H} r_{h'}^\star(s_{h'}, a_{h'})\middle|(s_h, a_h) = (s, a), \pi\right]$. We also define the state visitation distribution $d_h^\pi(s) := \mathbb{P}^\pi(s_h = s)$ and state-action visitation distribution $d_h^\pi(s, a) := \mathbb{P}^\pi(s_h = s, a_h = a)$.

**Imitation Learning.** The goal of imitation learning (IL) is to acquire a high-quality policy *without* access to the reward function $r^\star$. To achieve this, we assume access to an expert policy $\pi^{\mathrm{E}}$ that generates a dataset of $N$ trajectories, each of length $H$:

$$\mathcal{D}^{\mathrm{E}} = \left\{\tau^i = \left(s_1^i, a_1^i, s_2^i, a_2^i, \ldots, s_H^i, a_H^i\right); a_h^i \sim \pi_h^{\mathrm{E}}(\cdot|s_h^i), s_{h+1}^i \sim P_h(\cdot|s_h^i, a_h^i), \forall h \in [H]\right\}_{i=1}^{N},$$

The learner uses this dataset $\mathcal{D}^{\mathrm{E}}$ to learn a policy that mimics the expert's behavior. We measure imitation quality using the *imitation gap* (Abbeel & Ng, 2004; Ross & Bagnell, 2010; Rajaraman et al., 2020), defined as $V^{\pi^{\mathrm{E}}} - V^\pi$, where $\pi$ is the learned policy. Essentially, we hope that the learned policy can perfectly mimic the expert such that the imitation gap is small.

Typical IL works (Ng & Russell, 2000; Abbeel & Ng, 2004) often assume that the expert policy is optimal regarding the true reward $r^\star$, which suffers from the issue that degenerated constant rewards can induce the same expert policy (Ziebart et al., 2008). Following (Ziebart et al., 2008; Bloem & Bambos, 2014), we avoid this issue by considering that the expert is a soft-optimal policy (Haarnoja et al., 2018; Geist et al., 2019) regarding $r^\star$. Formally, we can formulate the expert policy by

$$\pi_h^{\mathrm{E}}(a|s) = \exp\left(Q_h^{\star,\mathrm{soft}}(s, a) - V_h^{\star,\mathrm{soft}}(s)\right). \tag{1}$$

Here $Q_h^{\star,\mathrm{soft}}(s, a)$ and $V_h^{\star,\mathrm{soft}}(s)$ denote the soft-optimal Q-function and value function, respectively.

**Behavioral Cloning.** As a classical IL method, behavioral cloning (BC) (Pomerleau, 1991) performs maximum likelihood estimation (MLE) to mimic the expert.

$$\pi^{\mathrm{BC}} = \underset{\pi \in \Pi}{\mathrm{argmax}} \sum_{i=1}^{N} \sum_{h=1}^{H} \log\left(\pi_h(a_h^i|s_h^i)\right). \tag{2}$$

Here $\Pi$ is the set of all policies. This optimization problem can be solved entirely using pre-collected expert data without any environment interaction, making BC a purely offline method. However, this offline nature introduces a fundamental limitation: BC is susceptible to compounding errors (Ross & Bagnell, 2010), resulting in poor efficiency in terms of demonstrations.

**Adversarial Imitation Learning.** As another prominent class of IL methods, adversarial imitation learning (AIL) imitates expert behavior through a game-theoretic approach.

$$\max_{\pi \in \Pi} \min_{r \in \mathcal{R}} V_r^\pi - V_r^{\pi^{\mathrm{E}}}. \tag{3}$$

Here $V_r^\pi$ denotes the value of policy $\pi$ under reward $r$ and $\mathcal{R} := \{r : \forall(s, a, h) \in \mathcal{S} \times \mathcal{A} \times [H], r_h(s, a) \in [0, 1]\}$ denotes the reward class. In this minimax objective, AIL infers a reward function that maximizes the value gap between the expert policy and the learning policy. Subsequently, it learns a policy that minimizes this value gap using the inferred reward. Note that the outer optimization problem concerning the policy is equivalent to a reinforcement learning (RL) problem under the inferred reward $r$. Solving RL problems requires online environment interactions, marking AIL as an online approach. AIL has proven to mitigate the compounding errors issue in BC (Ho & Ermon, 2016; Kostrikov et al., 2019; Ghasemipour et al., 2019; Xu et al., 2020; Rajaraman et al., 2020), achieving a high expert sample efficiency. However, AIL relies on extensive online environment interactions, presenting a significant limitation in scenarios where such interactions are expensive.

## 3   THE CRITICAL ROLE OF REWARD PRETRAINING IN ADVERSARIAL IMITATION LEARNING

A natural approach to improve the interaction efficiency of AIL involves first pretraining policies via BC, then finetuning them through AIL with online interactions (Ho & Ermon, 2016). This intuitive strategy leverages BC to establish an acceptable initial policy before engaging in interaction-expensive adversarial learning. However, numerous empirical works (Sasaki et al., 2018; Jena et al., 2021; Orsini et al., 2021; Yue et al., 2024) have consistently found that policy pretraining alone provides minimal benefits. In particular, they observed that policy quality deteriorates rapidly at the beginning of AIL training, negating most advantages gained from the initial BC phase. This phenomenon suggests fundamental limits in standard AIL with policy pretraining that have yet to be theoretically understood.

Instead of merely observing that policy pre-training is insufficient, we develop a rigorous theoretical analysis for AIL with policy pre-training. Our analysis formally examines a standard AIL procedure with BC-pretrained policies, outlined in Algorithm 1.

---

**Algorithm 1** Adversarial Imitation Learning with Policy Pretraining Alone

**Input:** Randomly initialized reward $r^1$ and demonstrations $\mathcal{D}^{\mathrm{E}}$.
1: Pretrain a policy via BC based on Eq.(2): $\pi^1 \leftarrow \pi^{\mathrm{BC}}$.
2: **for** $k = 1, 2, \ldots, K-1$ **do**
3:    Calculate the Q-value function $\{Q_h^{\pi^k, r^k}\}_{h=1}^{H}$ for policy $\pi^k$.
4:    Update the policy by KL-regularized policy optimization:

$$\pi_h^{k+1}(\cdot|s) = \operatorname*{argmax}_{p \in \Delta(\mathcal{A})} \mathbb{E}_{a \sim p(\cdot)}\left[Q_h^{\pi^k, r^k}(s,a)\right] - \frac{1}{\eta} D_{\mathrm{KL}}\left(p(\cdot), \pi_h^k(\cdot|s)\right).$$

5:    Update the reward by solving the optimization problem of

$$r^{k+1} = \operatorname*{argmin}_{r \in \mathcal{R}} \mathbb{E}_{\tau \sim \pi^{k+1}}\left[\sum_{h=1}^{H} r_h(s_h, a_h)\right] - \mathbb{E}_{\tau \sim \mathcal{D}^{\mathrm{E}}}\left[\sum_{h=1}^{H} r_h(s_h, a_h)\right].$$

6: **end for**
**Output:** $\bar{\pi}$ sampled uniformly from $\{\pi^1, \ldots, \pi^K\}$.

---

Algorithm 1 operates in two stages. First, we pretrain policies through BC on expert demonstrations. Second, we conduct the online AIL process, which alternates between policy and reward updates. During policy updates, we employ KL-regularized policy optimization (Shani et al., 2020; Cai et al., 2020) to solve the outer RL problem in Eq.(3). During reward updates, with the newly recovered policy $\pi^{k+1}$, we update the reward by minimizing the policy value difference between $\pi^{k+1}$ and $\pi^{\mathrm{E}}$, i.e., $\min_{r \in \mathcal{R}} V_r^{\pi^{k+1}} - \widehat{V}_r^{\pi^{\mathrm{E}}}$, where $\widehat{V}_r^{\pi^{\mathrm{E}}} := \mathbb{E}_{\tau \sim \mathcal{D}^{\mathrm{E}}}[\sum_{h=1}^{H} r_h(s_h, a_h)]$ represents an empirical estimation of $V_r^{\pi^{\mathrm{E}}}$ based on demonstrations. Finally, following the standard online-to-batch conversion technique (Orabona, 2019), Algorithm 1 outputs a policy uniformly sampled from the recovered policies throughout training.

The following proposition provides the imitation gap bound of AIL with policy pretraining.

**Proposition 1.** *Consider adversarial imitation learning with policy pretraining shown in Algorithm 1. For any $\delta \in (0, 1)$, with probability at least $1 - \delta$, it holds that*

$$V^{\pi^{\mathrm{E}}} - V^{\bar{\pi}} \le \underbrace{\frac{1}{K}\left(V_{r^\star}^{\pi^{\mathrm{E}}} - V_{r^\star}^{\pi^1} - \left(V_{r^1}^{\pi^{\mathrm{E}}} - V_{r^1}^{\pi^1}\right)\right)}_{\text{reward error}} + 2\sqrt{\frac{2|\mathcal{S}||\mathcal{A}|H^2 \log(H/\delta)}{N}}$$

$$+ \underbrace{\frac{1}{\eta K}\mathbb{E}\left[\sum_{h=1}^{H} D_{\mathrm{KL}}(\pi_h^{\mathrm{E}}(\cdot|s_h), \pi_h^1(\cdot|s_h))\Big|\pi^{\mathrm{E}}\right]}_{\text{policy error}} + \frac{\eta}{2}H^3. \tag{4}$$

*Furthermore, consider pretraining policies via BC (i.e., $\pi^1 := \pi^{\mathrm{BC}}$) and choosing stepsize $\eta = \widetilde{\Theta}(\sqrt{(|\mathcal{S}||\mathcal{A}|)/(H^2 KN)})$, we have that*

$$
V^{\pi^{\mathrm{E}}} - V^{\bar{\pi}} \precsim \frac{1}{K}\left(V^{\pi^{\mathrm{E}}}_{r^\star} - V^{\pi^1}_{r^\star} - \left(V^{\pi^{\mathrm{E}}}_{r^1} - V^{\pi^1}_{r^1}\right)\right) + \sqrt{\frac{|\mathcal{S}||\mathcal{A}|H^2\log(H/\delta)}{N}}
$$
$$
+ \sqrt{\frac{|\mathcal{S}||\mathcal{A}|H^4\log^2(HN^2/\delta)}{KN}}.
$$
(5)

The complete proof is provided in Appendix A.1. Eq.(4) in Proposition 1 reveals that the imitation gap of AIL with policy pretraining contains two fundamental error components: reward error and policy error. The reward error quantifies the discrepancy between the true reward $r^\star$ and the initial reward $r^1$ through value difference. The policy error specifically measures the KL divergence between the expert policy $\pi^{\mathrm{E}}$ and the initial policy $\pi^1$. Besides, the second term in the RHS of Eq.(4) captures the statistical error arising from the finite number of expert demonstrations.

By pretraining the policy via BC (i.e., $\pi^1 \leftarrow \pi^{\mathrm{BC}}$), the KL divergence between $\pi^{\mathrm{E}}$ and $\pi^1$ can be notably reduced, as the BC policy is inherently closer to the expert than a randomly initialized policy. Formally, we can leverage the theoretical guarantee of BC (Tiapkin et al., 2024) to upper bound this divergence, yielding the sharper bound shown in Eq.(5). However, a critical limitation remains: the reward error persists at a large magnitude because the reward function $r^1$ is still randomly initialized and thus can be arbitrarily far from the true reward $r^\star$. This reward error can notably inflate the overall imitation gap, particularly in the early stages of training when $K$ is small.

This theoretical diagnosis serves a dual purpose: it mathematically isolates the reward error as the precise target for our algorithmic design, and establishes a tight theoretical baseline for the subsequent theoretical comparison.

## 4 POLICY-REWARD CO-PRETRAINING FOR ADVERSARIAL IMITATION LEARNING

Building on the theoretical insights from the previous section, we propose a joint pretraining approach for both policies and rewards to accelerate AIL. We first introduce a principled method for reward pretraining, then provide rigorous theoretical analysis demonstrating its effectiveness in reducing the imitation gap.

### 4.1 METHOD

Building on Proposition 1, we develop a reward pretraining method to reduce the key term $(V^{\pi^{\mathrm{E}}}_{r^\star} - V^\pi_{r^\star}) - (V^{\pi^{\mathrm{E}}}_r - V^\pi_r)$ in reward error. We refer to this term as the *relative policy evaluation error*, as it quantifies the discrepancy in evaluating the relative value difference between policies $\pi^{\mathrm{E}}$ and $\pi$. Based on the well-known simulation lemma (Kearns & Singh, 2002), a natural approach to reducing this error would be to pretrain a reward $r$ that closely approximates the original true reward $r^\star$, ensuring $|r^\star_h(s,a) - r_h(s,a)|$ is small. However, reward ambiguity fundamentally prevents recovering a reward function close to $r^\star$, even with complete knowledge of the expert policy and MDP (Cao et al., 2021; Metelli et al., 2021; Rolland et al., 2022).

To circumvent this limitation, we argue that learning a reward close to the original $r^\star$ is not necessary for reducing the relative policy evaluation error. Instead, we demonstrate that learning an accurate *shaping reward* (Ng et al., 1999) is already sufficient. We introduce the formal definition of the shaping reward as follows [1].

**Definition 1** (Shaping Reward (Ng et al., 1999))**.** *In an episodic MDP, for a reward function $r$ and potential shaping functions $\{\Phi_h : \mathcal{S} \to \mathbb{R}\}_{h=1}^{H+1}$ with $\Phi_{H+1} \equiv 0$, the shaping reward is defined as*

$$
\forall (s,a,h) \in \mathcal{S} \times \mathcal{A} \times [H], \; \widetilde{r}_h(s,a) := r_h(s,a) - \Phi_h(s) + \mathbb{E}_{s' \sim P_h(\cdot|s,a)}[\Phi_{h+1}(s')].
$$

---

[1] Ng et al. (1999) originally proposed shaping rewards for infinite-horizon discounted MDPs; we present the episodic adaptation here.

The core design principle involves a telescoping structure within the shaping functions, which theoretically guarantees that reward shaping preserves optimal policies (Ng et al., 1999).

Crucially, the following proposition shows that value differences $V^{\pi'} - V^{\pi}$ remain identical under both the original reward $r$ and its corresponding shaping reward $\widetilde{r}$.

**Proposition 2.** *For any pair of policies $\pi$ and $\pi'$, consider an arbitrary reward $r$ and its shaping reward $\widetilde{r}$ defined by $\widetilde{r}_h(s,a) := r_h(s,a) - \Phi_h(s) + \mathbb{E}_{s' \sim P_h(\cdot|s,a)}[\Phi_{h+1}(s')]$ with potential-based shaping functions $\{\Phi_h\}_{h=1}^{H+1}$, it holds that*

$$V_r^{\pi'} - V_r^{\pi} = V_{\widetilde{r}}^{\pi'} - V_{\widetilde{r}}^{\pi}.$$

The insight is that while individual policy values may differ between the original and shaping rewards, their relative differences remain invariant. Intuitively, according to the telescoping argument, the policy values of the original reward and the shaping reward only differ in the shaping value at the initial state, which cancels out when computing value differences. Proposition 2 has an important implication for our reward pretraining approach. It establishes that

$$(V_{r^\star}^{\pi^{\mathrm{E}}} - V_{r^\star}^{\pi}) - (V_r^{\pi^{\mathrm{E}}} - V_r^{\pi}) = (V_{\widetilde{r^\star}}^{\pi^{\mathrm{E}}} - V_{\widetilde{r^\star}}^{\pi}) - (V_r^{\pi^{\mathrm{E}}} - V_r^{\pi}),$$

where $\widetilde{r^\star}$ is certain shaping reward of $r^\star$. This reveals that learning a reward function close to any *shaping* reward $\widetilde{r}^\star$ is sufficient for reducing the relative policy evaluation error. As such, we do not need to recover the original reward $r^\star$ itself.

Having established that learning an accurate shaping reward is sufficient for reducing the reward error, we now develop a principled method to infer such a reward. According to Eq.(1) and the soft Bellman equation, we can characterize the true reward function as follows.

$$r_h^\star(s,a) = \log(\pi_h^{\mathrm{E}}(a|s)) + V_h^{\star,\mathrm{soft}}(s) - \mathbb{E}_{s' \sim P_h(\cdot|s,a)} \left[ V_{h+1}^{\star,\mathrm{soft}}(s') \right].$$

Crucially, we observe that $\widetilde{r}_h^\star(s,a) := \log(\pi_h^{\mathrm{E}}(a|s))$ is exactly a shaping reward of $r_h^\star(s,a)$ regarding the potential-based shaping functions $\{V_h^\star\}_{h=1}^{H+1}$ with $V_{H+1}^\star \equiv 0$. This shaping reward has an intuitive interpretation: it assigns greater values to actions with higher probabilities under the expert. This characterization naturally suggests our reward pretraining method. We first learn a BC policy $\pi^{\mathrm{BC}}$ and then pretrain the reward by setting $r_h^1(s,a) = \log(\pi_h^{\mathrm{BC}}(a|s))$. Since $\pi_h^{\mathrm{BC}}(a|s)$ approximates $\pi_h^{\mathrm{E}}(a|s)$ well based on maximum likelihood estimation, the pretrained reward $r_h^1(s,a)$ should be close to the target shaping reward $\widetilde{r}_h^\star(s,a)$. Equipping AIL with this joint pretraining of policies and rewards yields the overall algorithm termed **AIL** with Policy-Reward **Co**-**P**re**t**raining (CoPT-AIL), which is outlined in Algorithm 2.

The reward-shaping-based analysis reveals a fundamental connection between the expert policy and shaping reward, enabling a unified approach to policy and reward pretraining. This integration allows us to derive both components from a single learning procedure, eliminating the need for a separate reward learning step. The resulting computational efficiency gains are particularly valuable when working with large-parameter models.

### 4.2 THEORETICAL ANALYSIS

We now provide rigorous theoretical analysis demonstrating the effectiveness of CoPT-AIL.

**Theorem 1.** *Consider adversarial imitation learning with policy-reward co-pretraining shown in Algorithm 2. For any fixed $\delta \in (0,1)$, with probability at least $1-\delta$, the relative policy evaluation error is reduced as*

$$\frac{1}{K} \left( V_{r^\star}^{\pi^{\mathrm{E}}} - V_{r^\star}^{\pi^1} - \left( V_{r^1}^{\pi^{\mathrm{E}}} - V_{r^1}^{\pi^1} \right) \right) \lesssim \frac{C|\mathcal{S}||\mathcal{A}|H^2 \log^2 \left( |\mathcal{S}||\mathcal{A}|HN^2/\delta \right)}{KN}. \tag{6}$$

*Here $C := \max_{(s,h) \in \mathcal{S} \times [H]} d_h^{\pi^{\mathrm{BC}}}(s)/d_h^{\pi^{\mathrm{E}}}(s)$. Furthermore, the imitation gap satisfies that*

$$V^{\pi^{\mathrm{E}}} - V^{\bar{\pi}} \lesssim \frac{C|\mathcal{S}||\mathcal{A}|H^2 \log^2 \left( |\mathcal{S}||\mathcal{A}|HN^2/\delta \right)}{KN} + \sqrt{\frac{|\mathcal{S}||\mathcal{A}|H^2 \log(H/\delta)}{N}}$$

$$+ \sqrt{\frac{|\mathcal{S}||\mathcal{A}|H^4 \log^2(HN^2/\delta)}{KN}}. \tag{7}$$

---

**Algorithm 2** Adversarial Imitation Learning with Policy-Reward Co-Pretraining

---

**Input:** Demonstrations $\mathcal{D}^{\mathrm{E}}$.
1: Pretrain a policy via BC based on Eq.(2): $\pi^1 \leftarrow \pi^{\mathrm{BC}}$.
2: Pretrain a reward through $r_h^1(s,a) = \log(\pi_h^{\mathrm{BC}}(a|s))$.
3: **for** $k = 1, 2, \ldots, K-1$ **do**
4:     Calculate the Q-value function $\{Q_h^{\pi^k, r^k}\}_{h=1}^H$ for policy $\pi^k$.
5:     Update the policy by KL-regularized policy optimization:

$$\pi_h^{k+1}(\cdot|s) = \underset{p \in \Delta(\mathcal{A})}{\mathrm{argmax}} \, \mathbb{E}_{a \sim p(\cdot)} \left[ Q_h^{\pi^k, r^k}(s,a) \right] - \frac{1}{\eta} D_{\mathrm{KL}}\left(p(\cdot), \pi_h^k(\cdot|s)\right).$$

6:     Update the reward by solving the optimization problem of

$$r^{k+1} = \underset{r \in \mathcal{R}}{\mathrm{argmin}} \, \mathbb{E}_{\tau \sim \pi^{k+1}} \left[ \sum_{h=1}^H r_h(s_h, a_h) \right] - \mathbb{E}_{\tau \sim \mathcal{D}^{\mathrm{E}}} \left[ \sum_{h=1}^H r_h(s_h, a_h) \right].$$

7: **end for**
**Output:** $\overline{\pi}$ sampled uniformly from $\{\pi^1, \ldots, \pi^K\}$.

---

The complete proof is presented in Appendix A.3. Theorem 1 implies that our reward pretraining approach can reduce the relative policy evaluation error to $\widetilde{\mathcal{O}}(C|\mathcal{S}||\mathcal{A}|H^2/(KN))$, which decreases rapidly as the number of expert trajectories $N$ increases. This validates that our reward pretraining approach can effectively leverage expert demonstrations to infer a good initial reward.

Furthermore, Theorem 1 indicates that CoPT-AIL achieves an overall imitation gap bound of $\widetilde{\mathcal{O}}((C|\mathcal{S}||\mathcal{A}|H^2)/(KN) + \sqrt{|\mathcal{S}||\mathcal{A}|H^2/N} + \sqrt{|\mathcal{S}||\mathcal{A}|H^4/(KN)})$. In comparison, Shani et al. (2021) proved that standard AIL without pretraining achieves $\widetilde{\mathcal{O}}(\sqrt{|\mathcal{S}||\mathcal{A}|H^2/K} + \sqrt{|\mathcal{S}||\mathcal{A}|H^3/N} + \sqrt{H^4/K})$. Our analysis reveals that CoPT-AIL achieves a better imitation gap bound when the number of expert trajectories satisfies $N \succsim C\sqrt{|\mathcal{S}||\mathcal{A}|H^2/K}$ [2]. Intuitively, when a reasonable number of demonstrations are available, jointly pretraining both the policy and reward can achieve good initial performance, thereby effectively accelerating the AIL process. To our best knowledge, Theorem 1 provides the first theoretical guarantee for the efficiency gains of pretraining in AIL.

## 5 RELATED WORKS

**Adversarial Imitation Learning.** AIL (Abbeel & Ng, 2004; Syed & Schapire, 2007; Ho & Ermon, 2016; Ghasemipour et al., 2019; Kostrikov et al., 2019; 2020) represents a prominent class of IL methods that mimics expert behavior through a game-theoretic formulation. Although AIL demonstrates superior expert sample efficiency compared to BC, it typically requires extensive online environment interactions. To mitigate this limitation, recent studies have explored combining AIL with BC to enhance interaction efficiency (Jena et al., 2021; Haldar et al., 2023; Watson et al., 2023; Yue et al., 2024). Specifically, some approaches augment the AIL objective directly with the BC objective (Jena et al., 2021; Haldar et al., 2023), while others leverage additional prior policies (Watson et al., 2023) or supplementary datasets (Yue et al., 2024) to learn the reward function. However, these methods generally lack theoretical guarantees regarding the benefits of their proposed techniques. In contrast, this paper provides theoretical guarantees for the efficiency gains achieved by our proposed method.

On the theoretical aspect, several studies have analyzed the theoretical convergence of AIL in the online setting (Syed & Schapire, 2007; Shani et al., 2021; Liu et al., 2021; Xu et al., 2023; Viano et al., 2022; 2024). In particular, Shani et al. (2021) proposes employing online optimization methods (Shalev-Shwartz, 2007) to update the policy and reward, and provides the imitation gap bound in the tabular setup. Furthermore, Liu et al. (2021); Viano et al. (2024) extend this idea to the linear function approximation setting and Chen et al. (2024) offers theoretical support for the sample efficiency of

---

[2] The detailed comparison is provided in Appendix A.4

off-policy AIL algorithms. These works analyze AIL initialized with random policies and rewards, and focus on the iterative online learning process. None of these works theoretically identify the error from initial policies and rewards, or provide theoretical guarantees for any pre-training strategy. In contrast, this work provides the first systematic theoretical treatment of reward pre-training in AIL. Specifically, we theoretically reveal the critical impact of reward pre-training in AIL and introduce a joint pretraining approach for both the policy and reward. We prove that this joint pretraining approach leads to an improved imitation gap bound, enhancing the theoretical performance guarantees of AIL.

**Inverse Reinforcement Learning.**    IRL (Ng & Russell, 2000; Arora & Doshi, 2021) aims to recover the underlying reward function from expert demonstrations. Our reward pretraining method is situated within the offline IRL literature (Garg et al., 2021; Yue et al., 2023; Zeng et al., 2023; Wei et al., 2023). Unlike most prior approaches that require a supplementary, non-expert dataset to learn the reward (Yue et al., 2023; Zeng et al., 2023; Wei et al., 2023), our method operates using only the expert demonstrations. While other purely offline methods exist, such as the work of (Kostrikov et al., 2020), which first learns a Q-function and then derives the reward function through the inverse Bellman operator, our approach is distinct. Our reward-shaping-based analysis uncovers a connection between the expert policy and shaping reward, enabling us to simultaneously pretrain the reward function and policy from a single BC procedure.

# 6 SIMULATION STUDIES

This section validates the superiority of CoPT-AIL through simulation studies. We provide a brief overview of the experimental setup below, with detailed information available in Appendix C.

## 6.1 EXPERIMENT SETUP

**Environment.**    We conduct experiments across 6 tasks from the feature-based DMControl benchmark (Tassa et al., 2018), a widely adopted benchmark in imitation learning that provides diverse continuous control tasks. For each task, we train an agent using the online RL algorithm DrQ-v2 (Yarats et al., 2021) with sufficient environment interactions and treat the resulting policy as the expert policy. We then collect expert demonstrations by rolling out this expert policy. Each algorithm is evaluated across three trials with different random seeds, and policy performance is assessed using Monte Carlo approximation over 10 trajectories per evaluation.

**Baselines.**    We compare CoPT-AIL against established deep imitation learning methods, including BC (Pomerleau, 1991), IQLearn (Garg et al., 2021), PPIL (Viano et al., 2022), FILTER (Swamy et al., 2023), and HyPE (Ren et al., 2024), although most lack theoretical guarantees. Notably, FILTER, PPIL, and HyPE represent prior state-of-the-art (SOTA) deep AIL approaches. Implementation details are provided in Appendix C.

## 6.2 EXPERIMENT RESULTS

**Overall Performance.**    Figure 1 presents the learning curves regarding online environment interactions for different algorithms. All AIL approaches use 200k online interactions on simpler tasks `Cartpole Swingup` and `Finger Spin`, and 500k online interactions on other tasks. The results reveal that CoPT-AIL consistently matches or exceeds the convergence rates of prior SOTA AIL methods across all 6 tasks. Particularly, on `Cartpole Swingup`, `Hopper Hop`, `Hopper Stand` and `Finger Spin`, CoPT-AIL can achieve near-expert performance with significantly fewer online interactions than existing approaches. These empirical results corroborate our theoretical analysis that the proposed joint pretraining mechanism yields a superior imitation gap in CoPT-AIL.

**Ablation Study.**    To validate the effectiveness of our proposed joint pretraining mechanism, we conduct an ablation study comparing CoPT-AIL against two baselines: pure AIL without pretraining (AIL) and AIL with policy pretraining alone ($\pi$PT-AIL). Figure 2 presents the learning curves for these three algorithms. The results reveal that $\pi$PT-AIL achieves convergence rates similar to standard AIL, indicating limited improvement in interaction efficiency from policy pretraining alone. Furthermore, $\pi$PT-AIL exhibits instability, particularly on `Cartpole Swingup` and `Finger`

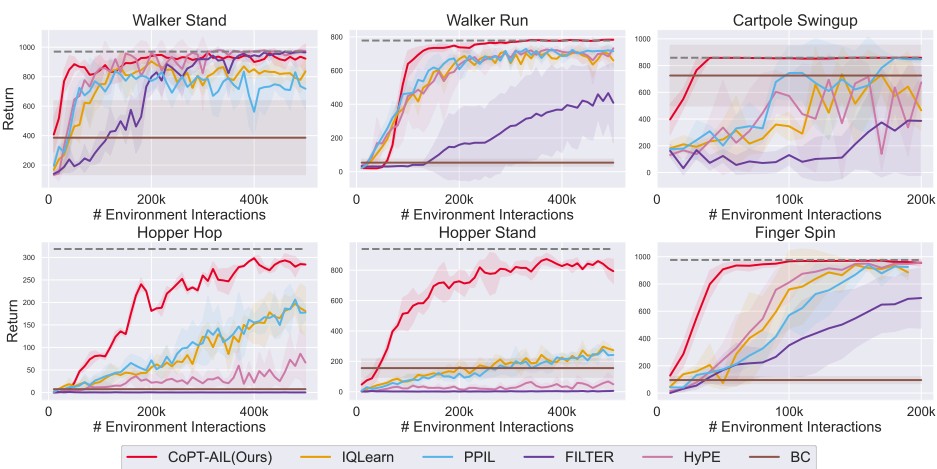

Figure 1: Learning curves regarding online environment interactions on 6 DMControl tasks. Here the $x$-axis is the number of environment interactions and the $y$-axis is the return.

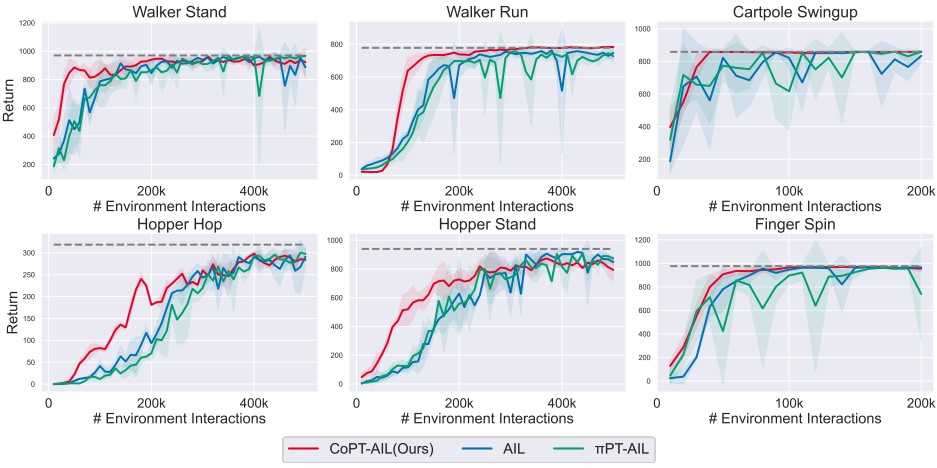

Figure 2: Learning curves regarding online environment interactions on 6 DMControl tasks. Here the $x$-axis is the number of environment interactions and the $y$-axis is the return.

Spin tasks. In contrast, CoPT-AIL demonstrates faster and more stable convergence across 6 tasks, with particularly pronounced improvements on Hopper Hop and Hopper Stand.

## 7 CONCLUSION

This paper proposes a principled policy-reward joint pretraining method to provably accelerate AIL. This paper begins with a theoretical analysis of AIL using policy pretraining alone, which isolates the reward error as the theoretical bottleneck and establishes a tight theoretical baseline. Guided by this theoretical diagnosis, we derive a principled reward pretraining method grounded in reward-shaping-based analysis. The analysis uncovers a fundamental connection between expert policy and shaping reward, naturally giving rise to our CoPT-AIL approach that jointly pretrains policies and rewards through a single BC procedure. Our theoretical results establish that CoPT-AIL achieves an improved imitation gap bound compared to standard AIL without pretraining, providing the first theoretical guarantee for the benefits of pretraining in AIL. Experimental evaluation confirms CoPT-AIL's superior performance over prior AIL methods.

Building on this work, there are several promising future directions deserving investigation. First, as a first step toward understanding the theoretical benefits of pretraining in AIL, this work focuses on the standard tabular setup. A valuable direction for future work is extending our theoretical results to function approximation scenarios. Besides, it would also be interesting to apply CoPT-AIL in more complex robot learning tasks, particularly in environments leveraging foundation models such as vision-language-action architectures.

## 8 ETHICS STATEMENT

This paper investigates the theoretical underpinnings of imitation learning and conforms with the ICLR Code of Ethics in every respect.

## 9 REPRODUCIBILITY STATEMENT

This paper provides all the information needed to reproduce the main results. For all theoretical results, the complete proof is provided in Section A and Section B. For experimental results, we present all implementation details in Section 6.1 and Section C. Code and scripts are also provided in the supplementary materials to reproduce experimental results.

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

# A OMITTED PROOF

## A.1 PROOF OF PROPOSITION 1

*Proof.* First, we can decompose the imitation gap into the following two terms.

$$V^{\pi^{\mathrm{E}}} - V^{\bar{\pi}} = \frac{1}{K} \sum_{k=1}^{K} \left( V_{r^\star}^{\pi^{\mathrm{E}}} - V_{r^\star}^{\pi^k} \right)$$

$$= \frac{1}{K} \sum_{k=1}^{K} \left( V_{r^\star}^{\pi^{\mathrm{E}}} - V_{r^\star}^{\pi^k} - \left( V_{r^k}^{\pi^{\mathrm{E}}} - V_{r^k}^{\pi^k} \right) \right) + \frac{1}{K} \sum_{k=1}^{K} \left( V_{r^k}^{\pi^{\mathrm{E}}} - V_{r^k}^{\pi^k} \right). \tag{8}$$

We first analyze the first term in the RHS.

$$\frac{1}{K} \sum_{k=1}^{K} \left( V_{r^\star}^{\pi^{\mathrm{E}}} - V_{r^\star}^{\pi^k} - \left( V_{r^k}^{\pi^{\mathrm{E}}} - V_{r^k}^{\pi^k} \right) \right)$$

$$= \frac{1}{K} \left( V_{r^\star}^{\pi^{\mathrm{E}}} - V_{r^\star}^{\pi^1} - \left( V_{r^1}^{\pi^{\mathrm{E}}} - V_{r^1}^{\pi^1} \right) \right) + \frac{1}{K} \sum_{k=2}^{K} \left( \widehat{V}_{r^\star}^{\pi^{\mathrm{E}}} - V_{r^\star}^{\pi^k} - \left( \widehat{V}_{r^k}^{\pi^{\mathrm{E}}} - V_{r^k}^{\pi^k} \right) \right) + V_{r^\star}^{\pi^{\mathrm{E}}} - \widehat{V}_{r^\star}^{\pi^{\mathrm{E}}}$$

$$+ \frac{1}{K} \sum_{k=2}^{K} \left( \widehat{V}_{r^k}^{\pi^{\mathrm{E}}} - V_{r^k}^{\pi^{\mathrm{E}}} \right)$$

$$\leq \frac{1}{K} \left( V_{r^\star}^{\pi^{\mathrm{E}}} - V_{r^\star}^{\pi^1} - \left( V_{r^1}^{\pi^{\mathrm{E}}} - V_{r^1}^{\pi^1} \right) \right) + \frac{1}{K} \sum_{k=2}^{K} \left( \widehat{V}_{r^\star}^{\pi^{\mathrm{E}}} - V_{r^\star}^{\pi^k} - \left( \widehat{V}_{r^k}^{\pi^{\mathrm{E}}} - V_{r^k}^{\pi^k} \right) \right)$$

$$+ 2 \max_{r \in \mathcal{R}} \left| V_r^{\pi^{\mathrm{E}}} - \widehat{V}_r^{\pi^{\mathrm{E}}} \right|.$$

For any reward $r \in \mathcal{R}$, $\widehat{V}_r^{\pi^{\mathrm{E}}} := \mathbb{E}_{\tau \sim \mathcal{D}^{\mathrm{E}}} \left[ \sum_{h=1}^{H} r_h(s_h, a_h) \right]$ denotes the empirical estimation of $V_r^{\pi^{\mathrm{E}}}$ based on demonstrations $\mathcal{D}^{\mathrm{E}}$. Furthermore, $\forall r \in \mathcal{R}$, we have that

$$\left| V_r^{\pi^{\mathrm{E}}} - \widehat{V}_r^{\pi^{\mathrm{E}}} \right| = \left| \sum_{h=1}^{H} \sum_{(s,a) \in \mathcal{S} \times \mathcal{A}} \left( d_h^{\pi^{\mathrm{E}}}(s,a) - \widehat{d_h^{\pi^{\mathrm{E}}}}(s,a) \right) r_h(s,a) \right|$$

$$\leq \sum_{h=1}^{H} \sum_{(s,a) \in \mathcal{S} \times \mathcal{A}} \left| d_h^{\pi^{\mathrm{E}}}(s,a) - \widehat{d_h^{\pi^{\mathrm{E}}}}(s,a) \right| |r_h(s,a)|$$

$$\leq \sum_{h=1}^{H} \sum_{(s,a) \in \mathcal{S} \times \mathcal{A}} \left| d_h^{\pi^{\mathrm{E}}}(s,a) - \widehat{d_h^{\pi^{\mathrm{E}}}}(s,a) \right|.$$

Here $d_h^{\pi^{\mathrm{E}}}(s,a) := \mathbb{P}^{\pi^{\mathrm{E}}}(s_h = s, a_h = a)$ represents the probability of visiting $(s,a)$ in time step $h$ by following $\pi^{\mathrm{E}}$. Besides, $\widehat{d_h^{\pi^{\mathrm{E}}}}(s,a) := n_h^{\mathrm{E}}(s,a)/N$ represents the empirical estimation based on demonstrations $\mathcal{D}^{\mathrm{E}}$, where $n_h^{\mathrm{E}}(s,a)$ denotes the number of times that $(s,a)$ is visited in time step $h$ in $\mathcal{D}^{\mathrm{E}}$. The last inequality follows that $r_h(s,a) \in [0,1]$.

Based on (Weissman et al., 2003) and union bound, for any $\delta \in (0,1)$, with probability at least $1 - \delta$, we have that

$$\forall h \in [H], \quad \sum_{(s,a) \in \mathcal{S} \times \mathcal{A}} \left| d_h^{\pi^{\mathrm{E}}}(s,a) - \widehat{d_h^{\pi^{\mathrm{E}}}}(s,a) \right| \leq \sqrt{\frac{2|\mathcal{S}||\mathcal{A}| \log(H/\delta)}{N}}.$$

Then it holds that

$$\forall r \in \mathcal{R}, \ \left| V_r^{\pi^{\mathrm{E}}} - \widehat{V}_r^{\pi^{\mathrm{E}}} \right| \leq H \sqrt{\frac{2|\mathcal{S}||\mathcal{A}| \log(H/\delta)}{N}}.$$

Then we can obtain the following upper bound.

$$\frac{1}{K}\sum_{k=1}^{K}\left(V_{r^\star}^{\pi^{\mathrm{E}}} - V_{r^\star}^{\pi^k} - \left(V_{r^k}^{\pi^{\mathrm{E}}} - V_{r^k}^{\pi^k}\right)\right)$$

$$\leq \frac{1}{K}\left(V_{r^\star}^{\pi^{\mathrm{E}}} - V_{r^\star}^{\pi^1} - \left(V_{r^1}^{\pi^{\mathrm{E}}} - V_{r^1}^{\pi^1}\right)\right) + \frac{1}{K}\sum_{k=2}^{K}\left(\widehat{V}_{r^\star}^{\pi^{\mathrm{E}}} - V_{r^\star}^{\pi^k} - \left(\widehat{V}_{r^k}^{\pi^{\mathrm{E}}} - V_{r^k}^{\pi^k}\right)\right)$$

$$+ 2H\sqrt{\frac{2|\mathcal{S}||\mathcal{A}|\log(H/\delta)}{N}}$$

$$\leq \frac{1}{K}\left(V_{r^\star}^{\pi^{\mathrm{E}}} - V_{r^\star}^{\pi^1} - \left(V_{r^1}^{\pi^{\mathrm{E}}} - V_{r^1}^{\pi^1}\right)\right) + 2H\sqrt{\frac{2|\mathcal{S}||\mathcal{A}|\log(H/\delta)}{N}}. \tag{9}$$

The last inequality follows that $\forall k \geq 2, r^k = \mathrm{argmin}_{r \in \mathcal{R}} V_r^{\pi^k} - \widehat{V}_r^{\pi^{\mathrm{E}}}$.

We proceed to analyze the second term in the RHS of Eq.(8). According to the policy difference lemma (Kakade & Langford, 2002), we can obtain that

$$\frac{1}{K}\sum_{k=1}^{K}\left(V_{r^k}^{\pi^{\mathrm{E}}} - V_{r^k}^{\pi^k}\right) = \frac{1}{K}\sum_{k=1}^{K}\mathbb{E}\left[\sum_{h=1}^{H}\langle Q_h^{\pi^k,r^k}(s_h,\cdot), \pi_h^{\mathrm{E}}(\cdot|s_h) - \pi_h^k(\cdot|s_h)\rangle\Big|\pi^{\mathrm{E}}\right]$$

$$= \frac{1}{K}\mathbb{E}\left[\sum_{h=1}^{H}\sum_{k=1}^{K}\langle Q_h^{\pi^k,r^k}(s_h,\cdot), \pi_h^{\mathrm{E}}(\cdot|s_h) - \pi_h^k(\cdot|s_h)\rangle\Big|\pi^{\mathrm{E}}\right].$$

For each $(s,h) \in \mathcal{S} \times [H]$, we analyze the error term of $\sum_{k=1}^{K}\langle Q_h^{\pi^k,r^k}(s,\cdot), \pi_h^{\mathrm{E}}(\cdot|s) - \pi_h^k(\cdot|s)\rangle$. For a simplex $p \in \Delta(\mathcal{A})$, we define the linear function $\ell_{s,h}^k(p) := -\sum_{a \in \mathcal{A}} p(a) Q_h^{\pi^k,r^k}(s,a)$. Then we can regard the above error term as the regret for the online optimization problem with loss functions $\{\ell_{s,h}^k(p)\}_{k=1}^{K}$.

$$\sum_{k=1}^{K}\langle Q_h^{\pi^k,r^k}(s,\cdot), \pi_h^{\mathrm{E}}(\cdot|s) - \pi_h^k(\cdot|s)\rangle = \sum_{k=1}^{K}\ell_{s,h}^k(\pi_h^k(\cdot|s)) - \ell_{s,h}^k(\pi_h^{\mathrm{E}}(\cdot|s)).$$

Furthermore, performing KL-regularized policy optimization is equivalent to applying online mirror descent (Orabona, 2019) on the loss functions $\{\ell_{s,h}^k(p)\}_{k=1}^{K}$. According to the regret bound on online mirror descent (e.g., (Orabona, 2019, Theorem 6.8)), we have that

$$\sum_{k=1}^{K}\ell_{s,h}^k(\pi_h^k(\cdot|s)) - \ell_{s,h}^k(\pi_h^{\mathrm{E}}(\cdot|s)) \leq \frac{D_{\mathrm{KL}}(\pi_h^{\mathrm{E}}(\cdot|s), \pi_h^1(\cdot|s))}{\eta} + \frac{\eta}{2}\sum_{k=1}^{K}\left\|Q_h^{\pi^k,r^k}(s,\cdot)\right\|_\infty^2$$

$$\leq \frac{D_{\mathrm{KL}}(\pi_h^{\mathrm{E}}(\cdot|s), \pi_h^1(\cdot|s))}{\eta} + \frac{\eta}{2}KH^2.$$

The last inequality follows that $Q_h^{\pi^k,r^k}(s,a) \in [0,H]$ because of $r_h^k(s,a) \in [0,1], \forall(s,a,h) \in \mathcal{S} \times \mathcal{A} \times [H]$. Then we can obtain that

$$\frac{1}{K}\sum_{k=1}^{K}\left(V_{r^k}^{\pi^{\mathrm{E}}} - V_{r^k}^{\pi^k}\right) = \frac{1}{K}\mathbb{E}\left[\sum_{h=1}^{H}\sum_{k=1}^{K}\langle Q_h^{\pi^k,r^k}(s_h,\cdot), \pi_h^{\mathrm{E}}(\cdot|s_h) - \pi^k(\cdot|s_h)\rangle\Big|\pi^{\mathrm{E}}\right]$$

$$\leq \frac{1}{K}\mathbb{E}\left[\sum_{h=1}^{H}\frac{D_{\mathrm{KL}}(\pi_h^{\mathrm{E}}(\cdot|s_h), \pi_h^1(\cdot|s_h))}{\eta} + \frac{\eta}{2}KH^2\Big|\pi^{\mathrm{E}}\right]$$

$$= \frac{1}{\eta K}\mathbb{E}\left[\sum_{h=1}^{H}D_{\mathrm{KL}}(\pi_h^{\mathrm{E}}(\cdot|s_h), \pi_h^1(\cdot|s_h))\Big|\pi^{\mathrm{E}}\right] + \frac{\eta}{2}H^3$$

$$= \frac{1}{\eta K}\mathbb{E}\left[\sum_{h=1}^{H}D_{\mathrm{KL}}(\pi_h^{\mathrm{E}}(\cdot|s_h), \pi_h^{\mathrm{BC}}(\cdot|s_h))\Big|\pi^{\mathrm{E}}\right] + \frac{\eta}{2}H^3. \tag{10}$$

Combining the bounds in Eq.(9) and Eq.(10) finishes the proof of Eq.(4).

$$V^{\pi^{\mathrm{E}}} - V^{\overline{\pi}} \leq \frac{1}{K}\left(V_{r^\star}^{\pi^{\mathrm{E}}} - V_{r^\star}^{\pi^1} - \left(V_{r^1}^{\pi^{\mathrm{E}}} - V_{r^1}^{\pi^1}\right)\right) + 2H\sqrt{\frac{2|\mathcal{S}||\mathcal{A}|\log(H/\delta)}{N}}$$

$$+ \frac{1}{\eta K}\mathbb{E}\left[\sum_{h=1}^{H} D_{\mathrm{KL}}(\pi_h^{\mathrm{E}}(\cdot|s_h), \pi_h^{\mathrm{BC}}(\cdot|s_h))\bigg|\pi^{\mathrm{E}}\right] + \frac{\eta}{2}H^3.$$

We proceed to prove Eq.(5) in Proposition 1. In particular, with the policy pre-trained via BC, we can leverage the guarantee of BC to analyze the KL divergence between the expert policy and the initial policy. Note that we have proved the following upper bound.

$$\frac{1}{K}\sum_{k=1}^{K}\left(V_{r^k}^{\pi^{\mathrm{E}}} - V_{r^k}^{\pi^k}\right) \leq \frac{1}{\eta K}\mathbb{E}\left[\sum_{h=1}^{H} D_{\mathrm{KL}}(\pi_h^{\mathrm{E}}(\cdot|s_h), \pi_h^{\mathrm{BC}}(\cdot|s_h))\bigg|\pi^{\mathrm{E}}\right] + \frac{\eta}{2}H^3.$$

According to (Tiapkin et al., 2024, Corollary 1), with probability at least $1 - \delta$, we have that

$$\frac{1}{K}\sum_{k=1}^{K}\left(V_{r^k}^{\pi^{\mathrm{E}}} - V_{r^k}^{\pi^k}\right) \leq \frac{6|\mathcal{S}||\mathcal{A}|H\cdot\log\left(2\mathrm{e}^4 N\right)\cdot\log\left(12HN^2/\delta\right)}{\eta KN} + \frac{18AH}{\eta KN} + \frac{\eta}{2}H^3$$

$$\leq \frac{24|\mathcal{S}||\mathcal{A}|H\cdot\log\left(2\mathrm{e}^4 N\right)\cdot\log\left(12HN^2/\delta\right)}{\eta KN} + \frac{\eta}{2}H^3$$

$$\leq \frac{24|\mathcal{S}||\mathcal{A}|H\log^2(2e^4 HN^2/\delta)}{\eta KN} + \frac{\eta}{2}H^3$$

$$= 4\sqrt{\frac{3|\mathcal{S}||\mathcal{A}|H^4\log^2(2e^4 HN^2/\delta)}{KN}}.$$

The last equation holds by choosing the step-size $\eta = \sqrt{(48|\mathcal{S}||\mathcal{A}|\log^2(2e^4 HN^2/\delta))/(H^2 KN)}$. Finally, by union bound, we have that

$$V^{\pi^{\mathrm{E}}} - V^{\overline{\pi}} \leq \frac{1}{K}\left(V_{r^\star}^{\pi^{\mathrm{E}}} - V_{r^\star}^{\pi^1} - \left(V_{r^1}^{\pi^{\mathrm{E}}} - V_{r^1}^{\pi^1}\right)\right) + 2H\sqrt{\frac{2|\mathcal{S}||\mathcal{A}|\log(2H/\delta)}{N}}$$

$$+ 4\sqrt{\frac{3|\mathcal{S}||\mathcal{A}|H^4\log^2(4e^4 HN^2/\delta)}{KN}}.$$

We complete the proof.

$\square$

## A.2 PROOF OF PROPOSITION 2

Recall the definition of shaping reward $\widetilde{r}_h(s, a) := r_h(s, a) - \Phi_h(s) + \mathbb{E}_{s'\sim P_h(\cdot|s,a)}[\Phi_{h+1}(s')]$. For any policy $\pi$, we have that

$$V_{\widetilde{r}}^{\pi} = \mathbb{E}\left[\sum_{h=1}^{H}\widetilde{r}_h(s_h, a_h)\bigg|\pi\right]$$

$$= \mathbb{E}\left[\sum_{h=1}^{H}\left(r_h(s_h, a_h) - \Phi_h(s_h) + \mathbb{E}_{s'\sim P_h(\cdot|s_h,a_h)}[\Phi_{h+1}(s')]\right)\bigg|\pi\right]$$

$$\overset{(a)}{=} \mathbb{E}\left[\sum_{h=1}^{H}\left(r_h(s_h, a_h) - \Phi_h(s_h) + \Phi_{h+1}(s_{h+1})\right)\bigg|\pi\right]$$

$$\overset{(b)}{=} \mathbb{E}\left[\sum_{h=1}^{H}r_h(s_h, a_h) - \Phi_1(s_1)\bigg|\pi\right]$$

$$= \mathbb{E} \left[ \sum_{h=1}^{H} r_h(s_h, a_h) \middle| \pi \right] - \mathbb{E}_{s_1 \sim \rho} \left[ \Phi(s_1) \right]$$

$$= V_r^{\pi} - \mathbb{E}_{s_1 \sim \rho} \left[ \Phi(s_1) \right].$$

Here Equation (a) follows the tower property and $s_{h+1} \sim P_h(\cdot | s_h, a_h)$. Equation (b) follows the telescoping argument with boundary condition $\Phi_{H+1} \equiv 0$. Then for any pair of policies $\pi$ and $\pi'$, it holds that

$$V_r^{\pi'} - V_r^{\pi} = (V_{\widetilde{r}}^{\pi'} + \mathbb{E}_{s_1 \sim \rho} \left[ \Phi(s_1) \right]) - (V_{\widetilde{r}}^{\pi} + \mathbb{E}_{s_1 \sim \rho} \left[ \Phi(s_1) \right]) = V_{\widetilde{r}}^{\pi'} - V_{\widetilde{r}}^{\pi}.$$

We complete the proof.

### A.3 PROOF OF THEOREM 1

We first analyze the reward error of $(1/K) \cdot (V_{r^\star}^{\pi^E} - V_{r^\star}^{\pi^1} - (V_{r^1}^{\pi^E} - V_{r^1}^{\pi^1}))$. Recall that $\widetilde{r}_h^\star(s, a) := \log(\pi_h^E(a|s))$ is exactly a shaping reward of $r_h^\star(s, a)$ regarding the potential-based shaping functions $\{V_h^\star\}$. According to 2, we can obtain that

$$V_{r^\star}^{\pi^E} - V_{r^\star}^{\pi^1} = V_{\widetilde{r}^\star}^{\pi^E} - V_{\widetilde{r}^\star}^{\pi^1}$$

$$= \mathbb{E} \left[ \sum_{h=1}^{H} \log(\pi_h^E(a_h|s_h)) \middle| \pi^E \right] - \mathbb{E} \left[ \sum_{h=1}^{H} \log(\pi_h^E(a_h|s_h)) \middle| \pi^1 \right].$$

Then we can obtain that

$$\left( V_{r^\star}^{\pi^E} - V_{r^\star}^{\pi^1} - \left( V_{r^1}^{\pi^E} - V_{r^1}^{\pi^1} \right) \right)$$

$$= \left( \mathbb{E} \left[ \sum_{h=1}^{H} \log(\pi_h^E(a_h|s_h)) \middle| \pi^E \right] - \mathbb{E} \left[ \sum_{h=1}^{H} \log(\pi_h^E(a_h|s_h)) \middle| \pi^1 \right] \right.$$

$$\left. - \left( \mathbb{E} \left[ \sum_{h=1}^{H} \log(\pi_h^{BC}(a_h|s_h)) \middle| \pi^E \right] - \mathbb{E} \left[ \sum_{h=1}^{H} \log(\pi_h^{BC}(a_h|s_h)) \middle| \pi^1 \right] \right) \right)$$

$$= \mathbb{E} \left[ \sum_{h=1}^{H} \log(\pi_h^E(a_h|s_h)) - \log(\pi_h^{BC}(a_h|s_h)) \middle| \pi^E \right]$$

$$- \mathbb{E} \left[ \sum_{h=1}^{H} \log(\pi_h^E(a_h|s_h)) - \log(\pi_h^{BC}(a_h|s_h)) \middle| \pi^1 \right]$$

$$= \mathbb{E} \left[ \sum_{h=1}^{H} D_{\mathrm{KL}}(\pi_h^E(\cdot|s_h), \pi_h^{BC}(\cdot|s_h)) \middle| \pi^E \right] + \mathbb{E} \left[ \sum_{h=1}^{H} D_{\mathrm{KL}}(\pi_h^{BC}(\cdot|s_h), \pi_h^E(\cdot|s_h)) \middle| \pi^{BC} \right].$$

Based on (Tiapkin et al., 2024, Corollary 1), we can upper bound the first term in the RHS. With probability at least $1 - \delta$, we have that

$$\mathbb{E} \left[ \sum_{h=1}^{H} D_{\mathrm{KL}}(\pi_h^E(\cdot|s_h), \pi_h^{BC}(\cdot|s_h)) \middle| \pi^E \right]$$

$$\leq \frac{6|\mathcal{S}||\mathcal{A}|H \cdot \log\left(2\mathrm{e}^4 N\right) \cdot \log\left(12HN^2/\delta\right)}{N} + \frac{18|\mathcal{A}|H}{N}$$

$$\leq \frac{24|\mathcal{S}||\mathcal{A}|H \cdot \log^2\left(2e^4 HN^2/\delta\right)}{N}.$$

We further upper bound the second term.

$$\mathbb{E} \left[ \sum_{h=1}^{H} D_{\mathrm{KL}}(\pi_h^{BC}(\cdot|s_h), \pi_h^E(\cdot|s_h)) \middle| \pi^{BC} \right] = \sum_{h=1}^{H} \sum_{s \in \mathcal{S}} d_h^{\pi^{BC}}(s) D_{\mathrm{KL}}(\pi_h^{BC}(\cdot|s), \pi_h^E(\cdot|s))$$

$$\leq C \sum_{h=1}^{H} \sum_{s \in \mathcal{S}} d_h^{\pi^{\mathrm{E}}}(s) D_{\mathrm{KL}}(\pi_h^{\mathrm{BC}}(\cdot|s), \pi_h^{\mathrm{E}}(\cdot|s))$$

Here $C := \max_{(s,h) \in \mathcal{S} \times [H]} d_h^{\pi^{\mathrm{BC}}}(s)/d_h^{\pi^{\mathrm{E}}}(s)$. According to Lemma 1, with probability at least $1 - \delta$, it holds that

$$\forall (s,h) \in \mathcal{S} \times [H],\ D_{\mathrm{KL}}(\pi_h^{\mathrm{BC}}(\cdot|s), \pi_h^{\mathrm{E}}(\cdot|s)) \leq \frac{H|\mathcal{A}| \log(4|\mathcal{S}||\mathcal{A}|H(N+1)/\delta)}{N_h(s) + |\mathcal{A}|}.$$

Besides, with Lemma 4 and union bound, with probability at least $1 - \delta$,

$$\forall (s,h) \in \mathcal{S} \times [H], \frac{d_h^{\pi^{\mathrm{E}}}(s)}{\max\{N_h(s), 1\}} \leq \frac{12 \log(2|\mathcal{S}|H/\delta)}{N}.$$

With union bound, the above two events happen with probability at least $1 - 2\delta$. Conditioned on these two events, we have that

$$\mathbb{E}\left[\sum_{h=1}^{H} D_{\mathrm{KL}}(\pi_h^{\mathrm{BC}}(\cdot|s_h), \pi_h^{\mathrm{E}}(\cdot|s_h)) \Big| \pi^{\mathrm{BC}}\right]$$

$$\leq C \sum_{h=1}^{H} \sum_{s \in \mathcal{S}} d_h^{\pi^{\mathrm{E}}}(s) \frac{|\mathcal{A}|H \log(4|\mathcal{S}||\mathcal{A}|H(N+1)/\delta)}{N_h(s) + |\mathcal{A}|}$$

$$= C|\mathcal{A}|H \log(4|\mathcal{S}||\mathcal{A}|H(N+1)/\delta) \sum_{h=1}^{H} \sum_{s \in \mathcal{S}} \frac{d_h^{\pi^{\mathrm{E}}}(s)}{N_h(s) + |\mathcal{A}|}$$

$$\leq C|\mathcal{A}|H \log(4|\mathcal{S}||\mathcal{A}|H(N+1)/\delta) \sum_{h=1}^{H} \sum_{s \in \mathcal{S}} \frac{d_h^{\pi^{\mathrm{E}}}(s)}{\max\{N_h(s), 1\}}$$

$$\leq 12 \frac{C|\mathcal{S}||\mathcal{A}|H^2 \log(4|\mathcal{S}||\mathcal{A}|H(N+1)/\delta) \log(2|\mathcal{S}|H/\delta)}{N}$$

$$\leq 12 \frac{C|\mathcal{S}||\mathcal{A}|H^2 \log^2(4|\mathcal{S}||\mathcal{A}|H(N+1)/\delta)}{N}.$$

By union bound, with probability at least $1 - \delta$, it holds that

$$\frac{1}{K}\left(V_{r^\star}^{\pi^{\mathrm{E}}} - V_{r^\star}^{\pi^1} - \left(V_{r^1}^{\pi^{\mathrm{E}}} - V_{r^1}^{\pi^1}\right)\right)$$

$$\leq \frac{24|\mathcal{S}||\mathcal{A}|H \cdot \log^2\left(6e^4 HN^2/\delta\right)}{KN} + 12 \frac{C|\mathcal{S}||\mathcal{A}|H^2 \log^2(12|\mathcal{S}||\mathcal{A}|H(N+1)/\delta)}{KN}$$

$$\leq 48 \frac{C|\mathcal{S}||\mathcal{A}|H^2 \log^2\left(6e^4|\mathcal{S}||\mathcal{A}|HN^2/\delta\right)}{KN}.$$

We complete the proof of Eq.(6). Furthermore, Algorithm 2 differs from Algorithm 1 only in the reward initialization. Therefore, by following the same analysis in the proof of Proposition 1, we can obtain that

$$V^{\pi^{\mathrm{E}}} - V^{\bar{\pi}} \leq \frac{1}{K}\left(V_{r^\star}^{\pi^{\mathrm{E}}} - V_{r^\star}^{\pi^1} - \left(V_{r^1}^{\pi^{\mathrm{E}}} - V_{r^1}^{\pi^1}\right)\right) + 4\sqrt{\frac{3|\mathcal{S}||\mathcal{A}|H^4 \log^2(4e^4 HN^2/\delta)}{KN}}$$

$$+ 2H\sqrt{\frac{2|\mathcal{S}||\mathcal{A}| \log(2H/\delta)}{N}}$$

$$\leq 48 \frac{C|\mathcal{S}||\mathcal{A}|H^2 \log^2\left(6e^4|\mathcal{S}||\mathcal{A}|HN^2/\delta\right)}{KN} + 4\sqrt{\frac{3|\mathcal{S}||\mathcal{A}|H^4 \log^2(4e^4 HN^2/\delta)}{KN}}$$

$$+ 2H\sqrt{\frac{2|\mathcal{S}||\mathcal{A}| \log(2H/\delta)}{N}}.$$

We complete the proof of Eq.(7).

### A.4 BOUND COMPARISON

In this part, we compare the imitation gap bound of CoPT-AILwith that of OAL (Shani et al., 2021), a standard AIL algorithm without pretraining. In particular, without any pretraining, OAL updates the reward function via online projected gradient descent (Orabona, 2019) and updates the policy via KL-regularized policy optimization. Similar to CoPT-AIL, we consider that OAL can compute the Q-value function of the current policy. Now, we are ready to perform the bound comparison. Shani et al. (2021) decomposes the imitation gap into the following terms.

$$V_{r^\star}^{\pi^{\mathrm{E}}} - V_{r^\star}^{\bar{\pi}} \leq \underbrace{\frac{1}{K} \sum_{k=1}^{K} \left( \left( \widehat{V}_{r^\star}^{\pi^{\mathrm{E}}} - V_{r^\star}^{\pi^k} \right) - \left( \widehat{V}_{r^k}^{\pi^{\mathrm{E}}} - V_{r^k}^{\pi^k} \right) \right)}_{:=\mathrm{Term\ I}} + \underbrace{\frac{1}{K} \sum_{k=1}^{K} \left( V_{r^k}^{\pi^{\mathrm{E}}} - V_{r^k}^{\pi^k} \right)}_{:=\mathrm{Term\ II}}$$
$$+ \underbrace{2 \max_{r \in \mathcal{R}} \left| \widehat{V}_r^{\pi^{\mathrm{E}}} - V_r^{\pi^{\mathrm{E}}} \right|}_{:=\mathrm{Term\ III}}.$$

Lemmas 4, 5, and 6 in (Shani et al., 2021) upper bound Terms I, II, and III, respectively.

$$\text{Term I} \precsim \sqrt{\frac{|\mathcal{S}||\mathcal{A}|H^2}{K}}, \ \text{Term II} \precsim \sqrt{\frac{H^4 \log(|\mathcal{A}|)}{K}}, \ \text{Term III} \precsim \sqrt{\frac{|\mathcal{S}||\mathcal{A}|H^3 \log(1/\delta)}{N}}.$$

Finally, OAL attains the imitation gap bound of

$$\widetilde{\mathcal{O}} \left\{ \min \left\{ \sqrt{\frac{|\mathcal{S}||\mathcal{A}|H^2}{K}} + \sqrt{\frac{H^4 \log(|\mathcal{A}|)}{K}} + \sqrt{\frac{|\mathcal{S}||\mathcal{A}|H^3}{N}}, H \right\} \right\}.$$

Here $H$ represents the maximum value for the imitation gap. In comparison, CoPT-AIL attains the bound of

$$\widetilde{\mathcal{O}} \left( \min \left\{ \frac{C|\mathcal{S}||\mathcal{A}|H^2}{KN} + \sqrt{\frac{|\mathcal{S}||\mathcal{A}|H^4}{KN}} + \sqrt{\frac{|\mathcal{S}||\mathcal{A}|H^2}{N}}, H \right\} \right).$$

It is direct to derive that CoPT-AIL can achieve an improved imitation gap bound when $N \succsim C\sqrt{|\mathcal{S}||\mathcal{A}|H^2/K}$.

## B USEFUL LEMMAS

First, we provide the basic theoretical guarantee on BC. Following (Tiapkin et al., 2024), we consider the BC algorithm formulated as

$$\pi^{\mathrm{BC}} \in \underset{\pi \in \Pi}{\operatorname{argmax}} \sum_{h=1}^{H} \left( \sum_{i=1}^{N} \log(\pi_h(a_h^i \mid s_h^i)) + \mathcal{R}_h(\pi_h) \right). \tag{11}$$

Here $\mathcal{D} = \{(s_1^i, a_1^i, \ldots, s_H^i, a_H^i)\}_{i=1}^{N}$ denotes expert demonstrations and $\mathcal{R}_h(\pi_h) = \sum_{(s,a) \in \mathcal{S} \times \mathcal{A}} \log(\pi_h(a|s))$ is the regularizer. Tiapkin et al. (2024) proved theoretical bounds on the forward KL divergence between $\pi^{\mathrm{E}}$ and $\pi^{\mathrm{BC}}$. In the sequel, we provide a bound on the reverse KL divergence, which could be of independent interest.

**Lemma 1.** *Consider Eq.(11). With probability at least $1 - \delta$, it holds that*

$$\forall (s, h) \in \mathcal{S} \times [H], \ D_{\mathrm{KL}}(\pi_h^{\mathrm{BC}}(\cdot|s), \pi_h^{\mathrm{E}}(\cdot|s)) \leq \frac{H|\mathcal{A}| \log(4|\mathcal{S}||\mathcal{A}|H(N+1)/\delta)}{N_h(s) + |\mathcal{A}|}.$$

*Here $N_h(s) := \sum_{i=1}^{N} \mathbb{I}\{s_h^i = s\}$ denotes the number of times that states $s$ appears in demonstrations.*

*Proof.* The optimization problem in Eq.(11) admits the closed-form solution of

$$\pi_h^{\mathrm{BC}}(a|s) = \frac{N_h(s,a) + 1}{N_h(s) + |\mathcal{A}|}. \tag{12}$$

Here $N_h(s,a)$ represents the number of times that the state-action pair $(s,a)$ is visited in $\mathcal{D}$. We first analyze the case where $N_h(s) > 0$. We aim to upper bound the probability of $\mathbb{P}(D_{\mathrm{KL}}(\pi_h^{\mathrm{BC}}(\cdot|s), \pi_h^{\mathrm{E}}(\cdot|s)) \geq \varepsilon)$ for each $(s,h) \in \mathcal{S} \times [H]$. To analyze this probability, we reformulate $\pi^{\mathrm{BC}}$ as a mixture of two distributions.

$$\pi_h^{\mathrm{BC}}(a|s) = \frac{N_h(s)}{N_h(s) + |\mathcal{A}|} \cdot \frac{N_h(s,a)}{N_h(s)} + \frac{|\mathcal{A}|}{N_h(s) + |\mathcal{A}|} \cdot \frac{1}{|\mathcal{A}|}$$

$$= \frac{N_h(s)}{N_h(s) + |\mathcal{A}|} \cdot \widehat{\pi}_h(a|s) + \frac{|\mathcal{A}|}{N_h(s) + |\mathcal{A}|} \cdot p(a).$$

Here $\hat{\pi}$ denotes the empirical distribution from $\mathcal{D}$ and $p$ denotes the uniform distribution over $\mathcal{A}$. Furthermore, based on the convexity of KL divergence, we have that

$$D_{\mathrm{KL}}(\pi_h^{\mathrm{BC}}(\cdot|s), \pi_h^{\mathrm{E}}(\cdot|s)) \leq \frac{N_h(s)}{N_h(s) + |\mathcal{A}|} D_{\mathrm{KL}}(\widehat{\pi}_h(\cdot|s), \pi_h^{\mathrm{E}}(\cdot|s))$$

$$+ \frac{|\mathcal{A}|}{N_h(s) + |\mathcal{A}|} D_{\mathrm{KL}}(p(\cdot), \pi_h^{\mathrm{E}}(\cdot|s)).$$

Therefore, the event of $D_{\mathrm{KL}}(\pi_h^{\mathrm{BC}}(\cdot|s), \pi_h^{\mathrm{E}}(\cdot|s)) \geq \varepsilon$ implies that

$$D_{\mathrm{KL}}(\widehat{\pi}_h(\cdot|s), \pi_h^{\mathrm{E}}(\cdot|s)) \geq \frac{N_h(s) + |\mathcal{A}|}{N_h(s)} \cdot \left( \varepsilon - \frac{|\mathcal{A}|}{N_h(s) + |\mathcal{A}|} D_{\mathrm{KL}}(p(\cdot), \pi_h^{\mathrm{E}}(\cdot|s)) \right).$$

We define that

$$\varepsilon' := \frac{N_h(s) + |\mathcal{A}|}{N_h(s)} \cdot \left( \varepsilon - \frac{|\mathcal{A}|}{N_h(s) + |\mathcal{A}|} D_{\mathrm{KL}}(p(\cdot), \pi_h^{\mathrm{E}}(\cdot|s)) \right).$$

Then we have that

$$\mathbb{P}(D_{\mathrm{KL}}(\pi_h^{\mathrm{BC}}(\cdot|s), \pi_h^{\mathrm{E}}(\cdot|s)) \geq \varepsilon) \leq \mathbb{P}(D_{\mathrm{KL}}(\widehat{\pi}_h(a|s), \pi_h^{\mathrm{E}}(\cdot|s)) \geq \varepsilon').$$

According to Sanov's Theorem (Lemma 2), we have that

$$\mathbb{P}(D_{\mathrm{KL}}(\widehat{\pi}_h(a|s), \pi_h^{\mathrm{E}}(\cdot|s)) \geq \varepsilon') \leq (N_h(s) + 1)^{|\mathcal{A}|} \exp(-N_h(s)\varepsilon').$$

Setting the term in the RHS as the failure probability $\delta$ yields that

$$\varepsilon' = \frac{|\mathcal{A}| \log(N_h(s) + 1) + \log(1/\delta)}{N_h(s)},$$

$$\varepsilon = \frac{|\mathcal{A}| \log(N_h(s) + 1) + \log(1/\delta) + |\mathcal{A}| D_{\mathrm{KL}}(p(\cdot), \pi_h^{\mathrm{E}}(\cdot|s))}{N_h(s) + |\mathcal{A}|}.$$

This implies that for $N_h(s) > 0$, with probability at least $1 - \delta$,

$$D_{\mathrm{KL}}(\pi_h^{\mathrm{BC}}(\cdot|s), \pi_h^{\mathrm{E}}(\cdot|s)) \leq \frac{|\mathcal{A}| \log(N_h(s) + 1) + \log(1/\delta) + |\mathcal{A}| D_{\mathrm{KL}}(p(\cdot), \pi_h^{\mathrm{E}}(\cdot|s))}{N_h(s) + |\mathcal{A}|}$$

$$\overset{(a)}{\leq} \frac{|\mathcal{A}| \log(N_h(s) + 1) + \log(1/\delta) + H|\mathcal{A}| \log(4|\mathcal{A}|)}{N_h(s) + |\mathcal{A}|}$$

$$\leq \frac{H|\mathcal{A}| \log(4|\mathcal{A}|(N + 1)/\delta)}{N_h(s) + |\mathcal{A}|}.$$

Here inequality (a) follows Lemma 3.

When $N_h(s) = 0$, we have that $\pi_h^{\mathrm{BC}}(\cdot|s) = p(\cdot)$ according to Eq.(12). With Lemma 3, we can have that

$$D_{\mathrm{KL}}(\pi_h^{\mathrm{BC}}(\cdot|s), \pi_h^{\mathrm{E}}(\cdot|s)) = D_{\mathrm{KL}}(p(\cdot), \pi_h^{\mathrm{E}}(\cdot|s)) \leq H \log(4|\mathcal{A}|) \leq \frac{H|\mathcal{A}| \log(4|\mathcal{A}|(N + 1)/\delta)}{N_h(s) + |\mathcal{A}|}.$$

By combining the above two cases, we can conclude that with probability at least $1 - \delta$,

$$D_{\mathrm{KL}}(\pi_h^{\mathrm{BC}}(\cdot|s), \pi_h^{\mathrm{E}}(\cdot|s)) \leq \frac{H|\mathcal{A}| \log(4|\mathcal{A}|(N + 1)/\delta)}{N_h(s) + |\mathcal{A}|}.$$

Applying the union bound over $(s,h) \in \mathcal{S} \times [H]$ finishes the proof. $\qquad\square$

**Lemma 2** (Sanov's Theorem). *Suppose that $Q$ is a distribution over an alphabet $\mathcal{X}$ and $E$ is a set of distributions over $\mathcal{X}$. Let $\mathcal{D} = \{X_1, X_2, \ldots, X_N\}$ be i.i.d. samples drawn from the distribution $P$. Then*

$$\mathbb{P}(\widehat{P}_{\mathcal{D}} \in E) \leq (N+1)^{|\mathcal{X}|} \exp(-N D_{\mathrm{KL}}(P^\star, Q)),$$

*where $\widehat{P}_{\mathcal{D}}$ denote the empirical distribution from $\mathcal{D}$ and $P^\star = \operatorname{argmin}_{P \in E} D_{\mathrm{KL}}(P, Q)$.*

**Lemma 3.** *For any $(s, h) \in \mathcal{S} \times [H]$, consider $p$ is an uniform distribution over $\mathcal{A}$, we have that*

$$D_{\mathrm{KL}}(p(\cdot), \pi_h^{\mathrm{E}}(\cdot|s)) \leq H \log(4|\mathcal{A}|).$$

*Proof.* For any fixed $(s, h) \in \mathcal{S} \times [H]$,

$$D_{\mathrm{KL}}(p(\cdot), \pi_h^{\mathrm{E}}(\cdot|s)) = \sum_{a \in \mathcal{A}} \frac{1}{|\mathcal{A}|} \log\left(\frac{1/|\mathcal{A}|}{\pi_h^{\mathrm{E}}(a|s)}\right) = -\log(|\mathcal{A}|) - \frac{1}{|\mathcal{A}|} \sum_{a \in \mathcal{A}} \log(\pi_h^{\mathrm{E}}(a|s)).$$

According to Eq.(1), we can further obtain that

$$D_{\mathrm{KL}}(p(\cdot), \pi_h^{\mathrm{E}}(\cdot|s)) = -\log(|\mathcal{A}|) - \frac{\sum_{a \in \mathcal{A}} Q_h^{\star,\mathrm{soft}}(s, a)}{|\mathcal{A}|} + V_h^{\star,\mathrm{soft}}(s).$$

Notice that

$$Q_h^\star(s, a) = \mathbb{E}\left[\sum_{h'=h}^{H} r_{h'}^\star(s_{h'}, a_{h'}) + \sum_{h'=h+1}^{H} H(\pi_{h'}^{\mathrm{E}}(\cdot|s_{h'})) \middle| s_h = s, a_h = a, \pi^{\mathrm{E}}\right] \geq 0,$$

$$V_h^\star(s) = \mathbb{E}\left[\sum_{h'=h}^{H} (r_{h'}^\star(s_{h'}, a_{h'}) + H(\pi_{h'}^{\mathrm{E}}(\cdot|s_{h'}))) \middle| s_h = s, a_h = a, \pi^{\mathrm{E}}\right]$$
$$\leq (H - h + 1)(1 + \log(|\mathcal{A}|)).$$

Then we have that

$$D_{\mathrm{KL}}(p(\cdot), \pi_h^{\mathrm{E}}(\cdot|s)) \leq (H - h + 1)(1 + \log(|\mathcal{A}|)) \leq H(1 + \log(|\mathcal{A}|)) \leq H \log(4|\mathcal{A}|).$$

$\square$

**Lemma 4.** *Suppose $n \sim \mathrm{Bin}(N, p)$ where $N \geq 1$ and $p \in [0, 1]$. Then with probability at least $1 - \delta$, we have*

$$\frac{p}{\max\{n, 1\}} \leq \frac{12 \log(2/\delta)}{N}.$$

*Proof.* According to the Chernoff bound (Wainwright, 2019), with probability at least $1 - \delta$,

$$\left|\frac{n}{N} - p\right| \leq \sqrt{\frac{3p \log(2/\delta)}{N}}.$$

This implies a quadratic inequality regarding $x = \sqrt{p}$.

$$x^2 - bx - c \leq 0, \ b = \sqrt{\frac{3 \log(2/\delta)}{N}}, c = \frac{n}{N}.$$

Solving this inequality yields that

$$\sqrt{p} = x \leq \frac{b + \sqrt{b^2 + 4c}}{2} \leq b + \sqrt{c} = \sqrt{\frac{3 \log(2/\delta)}{N}} + \sqrt{\frac{n}{N}} \leq 2\sqrt{\frac{3 \max\{n, 1\} \log(2/\delta)}{N}}.$$

This directly implies that

$$p \leq \frac{12 \max\{n, 1\} \log(2/\delta)}{N}.$$

Rearanging the above inequality finishes the proof. $\square$

## C  EXPERIMENT DETAILS

### C.1  IMPLEMENTATION DETAILS OF CoPT-AIL

In this part, we present the detailed implementation of CoPT-AIL, which is outlined in Algorithm 3. In the pretraining stage, we first pretrain the policy via BC.

$$\pi^1 \leftarrow \pi^{\mathrm{BC}}, \ \pi^{\mathrm{BC}} = \operatorname*{argmax}_{\pi \in \Pi} \sum_{i=1}^{N} \sum_{h=1}^{H} \log\bigl(\pi_h(a_h^i \mid s_h^i)\bigr).$$

Then, according to the analysis in Section 4.1, we pretrain the reward by setting

$$r_h^1(s, a) = \log\left(\pi_h^1(a|s)\right).$$

For continuous action space, we parameterize policies with Gaussian distributions and utilize the log probability of the Gaussian distribution.

After the pretraining phase, we conduct the online AIL process, which alternates between policy and reward updates. In iteration $k$, for the policy update, we first learn the Q-function by minimizing the temporal difference learning objective.

$$\min_{Q \in \mathcal{Q}} \ell^k(Q) := \mathbb{E}_{\tau \sim \mathcal{D}^k} \left[ \sum_{h=1}^{H} \left( Q_h(s_h, a_h) - r_h^k(s_h, a_h) - \overline{Q}_{h+1}^k(s_{h+1}, \pi^k) \right)^2 \right] \tag{13}$$

Here $\mathcal{D}^k$ is the replay buffer consisting of all historical online trajectories and $\overline{Q} = \{\overline{Q}_1, \ldots, \overline{Q}_H\}$ is the delayed target Q-function. Besides, we define that $\overline{Q}_{h+1}^k(s_{h+1}, \pi^k) := \mathbb{E}_{a' \sim \pi_{h+1}^k(\cdot|s_{h+1})}[\overline{Q}_{h+1}(s_{h+1}, a')]$. With the newly learned Q-function $Q^{k+1}$, we update the policy by minimizing the objective of $\ell^k(\pi) := -\mathbb{E}_{\tau \sim \mathcal{D}^k}[\sum_{h=1}^{H} Q_h^{k+1}(s_h, \pi)]$.

For the reward update, the objective function is formulated by

$$\ell^k(r) := \mathbb{E}_{\tau \sim \pi^{k+1}} \left[ \sum_{h=1}^{H} r_h(s_h, a_h) + \beta \exp(-r_h(s_h, a_h)) \right] - \mathbb{E}_{\tau \sim \mathcal{D}^{\mathrm{E}}} \left[ \sum_{h=1}^{H} r_h(s_h, a_h) \right]. \tag{14}$$

Here we add a regularization term $\exp(-r_h(s_h, a_h))$ to improve the stability of reward training, and $\beta > 0$ is the regularization coefficient.

---

**Algorithm 3** Practical Implementation of CoPT-AIL

---

**Input:** Demonstrations $\mathcal{D}^{\mathrm{E}}$, replay buffer $\mathcal{D}^1 = \emptyset$.
 1: Pre-train a policy via BC based on Eq.(2): $\pi^1 \leftarrow \pi^{\mathrm{BC}}$.
 2: Pre-train a reward through $r_h^1(s, a) = \log(\pi_h^{\mathrm{BC}}(a|s))$.
 3: **for** $k = 1, 2, \ldots, K-1$ **do**
 4:     Update the Q-value function by $Q^{k+1} \leftarrow Q^k - \eta_Q \nabla \ell^k(Q)$ from Eq. (13).
 5:     Update the policy by $\pi^{k+1} \leftarrow \pi^k - \eta_\pi \nabla \ell^k(\pi)$, where $\ell^k(\pi) := \mathbb{E}_{\tau \sim \mathcal{D}^k}[\sum_{h=1}^{H} Q_h^{k+1}(s_h, \pi)]$.
 6:     Apply $\pi^{k+1}$ to roll out a trajectory $\tau^{k+1}$ and append it to the replay buffer $\mathcal{D}^{k+1} = \mathcal{D}^k \cup \{\tau^{k+1}\}$.
 7:     Update the reward function by $r^{k+1} \leftarrow r^k - \eta_r \nabla \ell^k(r)$ from Eq. (14).
 8:     Update the target Q-value by $\overline{Q}^{k+1} \leftarrow \tau Q^{k+1} + (1 - \tau)\overline{Q}^k$.
 9: **end for**

---

### C.2  ARCHITECTURE AND TRAINING DETAILS

The experiments are conducted on a machine with 64 CPU cores and 4 RTX4090 GPU cores. Each experiment is replicated three times using different random seeds. For each task, we adopt online DrQ-v2 (Yarats et al., 2021) to train an agent with sufficient environment interactions and regard the resultant policy as the expert policy. Specifically, we use 3M environment interactions for `Hopper`

`Hop`, and `Walker Run`, and 1M environment interactions for other tasks. Then we roll out this expert policy to collect expert demonstrations. The number of expert trajectories used for training in each task is provided in 1. The architecture and training details of CoPT-AIL and all baselines are listed below.

**CoPT-AIL:** Our codebase of CoPT-AIL extends the open-sourced framework of IQLearn. We retain the structure and parameter design of the critic from the original framework, and employ SAC (Haarnoja et al., 2018) with a fixed temperature coefficient for policy update. Note that CoPT-AIL pretrains the reward function using the BC policy. Therefore, we implement the reward model with the same architecture as the actor model. A comprehensive enumeration of the hyperparameters of CoPT-AIL is provided in Table 2.

**BC:** We implement BC based on our codebase. The actor model is trained using Mean Squared Error (MSE) loss over 10k training steps.

**PPIL:** We use the author's codebase, which is available at https://github.com/lviano/p2il. A comprehensive enumeration of the hyperparameters of PPIL is provided in Table 3.

**IQLearn:** We use the author's codebase, which is available at https://github.com/Div99/IQ-Learn. A comprehensive enumeration of the hyperparameters of IQLearn is provided in Table 4.

**FILTER:** We use the author's codebase, which is available at https://github.com/gkswamy98/fast_irl. A comprehensive enumeration of the hyperparameters of FILTER is provided in Table 5.

**HyPE:** We use the author's codebase, which is available at https://github.com/gkswamy98/hyper. A comprehensive enumeration of the hyperparameters of HyPE is provided in Table 6.

Table 1: Number of expert trajectories for each task.

| Task | Expert Trajectories |
|---|---|
| Walker Stand | 30 |
| Walker Run | 50 |
| Cartpole Swingup | 10 |
| Hopper Hop | 10 |
| Hopper Stand | 10 |
| Finger Spin | 50 |

Table 2: CoPT-AIL Hyper-parameters.

| Parameter | Value |
|---|---|
| discount factor | 0.99 |
| reward regularization coefficient $\beta$ | 1 |
| temperature coefficient | $10^{-2}$ |
| replay buffer size | $5 \cdot 10^5$ |
| batch size | 256 |
| optimizer | Adam |
| *Reward* | |
| learning rate | $1 \cdot 10^{-5}$ |
| number of hidden layers | 2 |
| number of hidden units per layer | 1024 |
| activation | ReLU |
| *Actor* | |
| learning rate | $3 \cdot 10^{-5}$ |
| number of hidden layers | 2 |
| number of hidden units per layer | 1024 |
| activation | ReLU |
| *Critic* | |
| learning rate | $3 \cdot 10^{-4}$ |
| number of hidden layers | 2 |
| number of hidden units per layer | 256 |
| activation | ReLU |

Table 3: PPIL Hyper-parameters.

| Parameter | Value |
|---|---|
| discount factor | 0.99 |
| gradient penalty coefficient | 10 |
| replay buffer size | $1 \cdot 10^6$ |
| batch size | 256 |
| optimizer | Adam |
| *Actor* | |
| learning rate | $3 \cdot 10^{-4}$ |
| number of hidden layers | 2 |
| number of hidden units per layer | 256 |
| activation | ReLU |
| *Critic* | |
| learning rate | $3 \cdot 10^{-4}$ |
| number of hidden layers | 2 |
| number of hidden units per layer | 256 |
| activation | ReLU |

Table 4: IQLearn Hyper-parameters.

| Parameter | Value |
|---|---|
| discount factor | 0.99 |
| gradient penalty coefficient | 10 |
| replay buffer size | $1 \cdot 10^6$ |
| batch size | 256 |
| optimizer | Adam |
| *Reward* | |
| learning rate | $3 \cdot 10^{-4}$ |
| number of hidden layers | 2 |
| number of hidden units per layer | 256 |
| *Actor* | |
| learning rate | $3 \cdot 10^{-4}$ |
| number of hidden layers | 2 |
| number of hidden units per layer | 256 |
| activation | ReLU |
| *Critic* | |
| learning rate | $3 \cdot 10^{-4}$ |
| number of hidden layers | 2 |
| number of hidden units per layer | 256 |
| activation | ReLU |

Table 5: FILTER Hyper-parameters.

| Parameter | Value |
|---|---|
| discount factor | 0.98 |
| gradient penalty coefficient | 10 |
| replay buffer size | $1 \cdot 10^6$ |
| batch size | 256 |
| optimizer | Adam |
| *Reward* | |
| learning rate | $8 \cdot 10^{-4}$ |
| batch size | 4096 |
| number of hidden layers | 2 |
| number of hidden units per layer | 256 |
| *Actor* | |
| learning rate | $7.3 \cdot 10^{-4}$ |
| number of hidden layers | 2 |
| number of hidden units per layer | 256 |
| activation | ReLU |
| *Critic* | |
| learning rate | $7.3 \cdot 10^{-4}$ |
| number of hidden layers | 2 |
| number of hidden units per layer | 256 |
| activation | ReLU |

Table 6: HyPE Hyper-parameters.

| Parameter | Value |
|---|---|
| discount factor | 0.98 |
| gradient penalty coefficient | 10 |
| replay buffer size | $1 \cdot 10^6$ |
| batch size | 256 |
| optimizer | Adam |
| *Reward* | |
| learning rate | $8 \cdot 10^{-4}$ |
| batch size | 4096 |
| number of hidden layers | 2 |
| number of hidden units per layer | 256 |
| *Actor* | |
| learning rate | $7.3 \cdot 10^{-4}$ |
| number of hidden layers | 2 |
| number of hidden units per layer | 256 |
| activation | ReLU |
| *Critic* | |
| learning rate | $7.3 \cdot 10^{-4}$ |
| number of hidden layers | 2 |
| number of hidden units per layer | 256 |
| activation | ReLU |

## D  ADDITIONAL EXPERIMENTAL RESULTS

### D.1  EXPERIMENTAL RESULTS ON ADDITIONAL TASKS

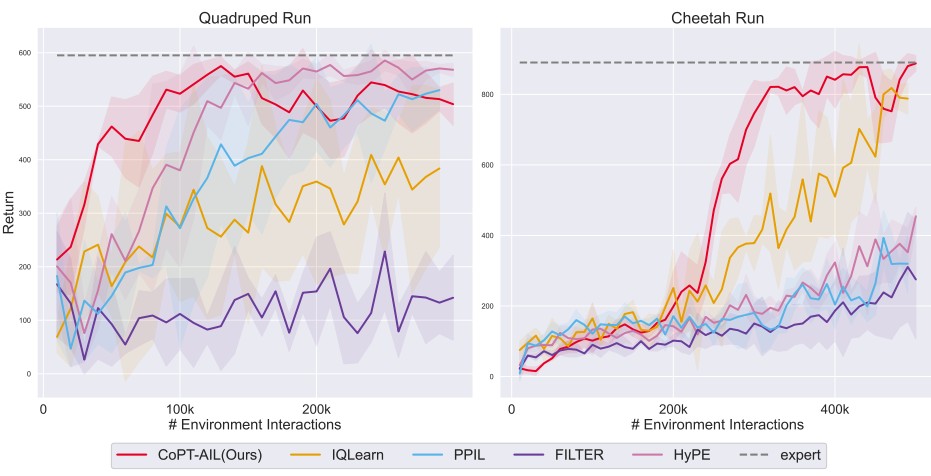

Figure 3: Learning curves of CoPT-AIL and the other baselines in two additional DMControl benchmark tasks. Here the $x$-axis is the number of environment interactions and the $y$-axis is the return.

Here we evaluate CoPT-AIL against baselines on two more challenging DMC tasks: `Quadruped Run` and `Cheetah Run`. The updated learning curves are provided in Figure 3. We use 10 expert trajectories in `Quadruped Run` and 50 expert trajectories in `Cheetah Run`. We observe that CoPT-AIL consistently converges faster than prior state-of-the-art AIL baselines on both tasks. These

additional results further support our theoretical analysis, demonstrating that the proposed reward pre-training mechanism leads to a smaller imitation gap.

## D.2 COMPARISONS WITH OLLIE (YUE ET AL., 2024) AND GAIL+BC (JENA ET AL., 2021)

We additionally compare CoPT-AIL against two baselines: OLLIE (Yue et al., 2024) and GAIL+BC (Jena et al., 2021). Unlike our setting, OLLIE assumes access to both expert demonstrations and a supplementary dataset. To adapt OLLIE to our setting, we directly set the supplementary dataset to an empty set. For both baselines, we use the official implementations provided in their repositories: `https://github.com/HansenHua/OLLIE-ICML24` and `https://github.com/rohitrango/BC-regularized-GAIL`.

The resulting learning curves are shown in Figure 4. We observe that CoPT-AIL matches or surpasses the convergence rates of OLLIE and GAIL+BC across all tasks. These results further support our theoretical analysis that the proposed reward pre-training mechanism leads to a smaller imitation gap and improved learning efficiency.

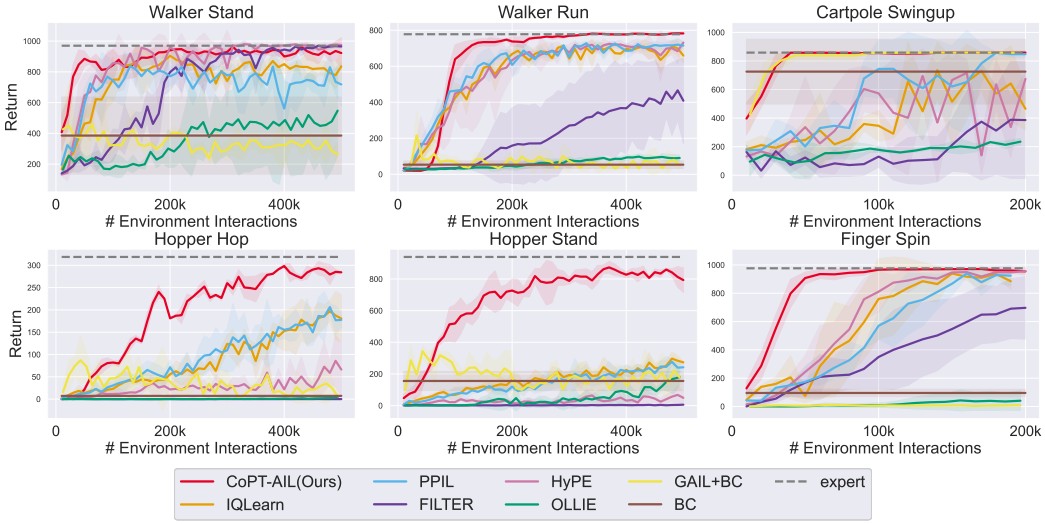

Figure 4: Comparisons with OLLIE (Yue et al., 2024) and GAIL+BC (Jena et al., 2021). Here the $x$-axis is the number of environment interactions and the $y$-axis is the return.

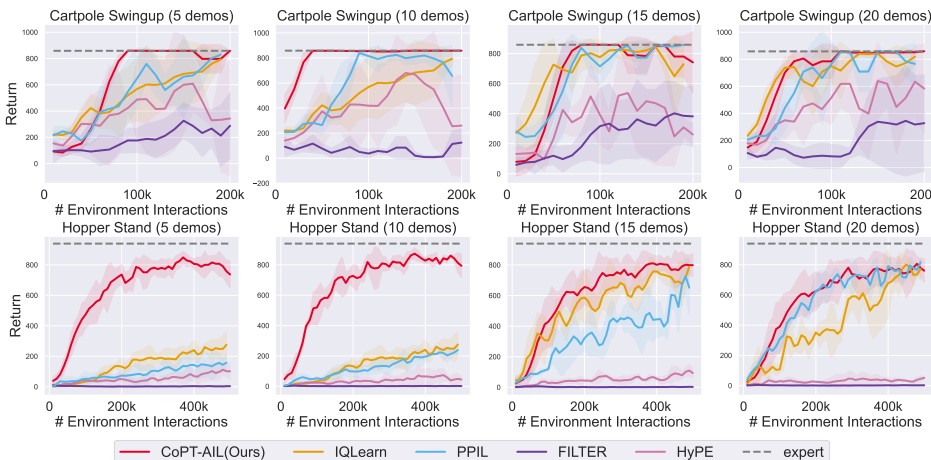

Figure 5: Learning curves of CoPT-AIL and the other baselines under different numbers of expert trajectories. Here the $x$-axis is the number of environment interactions and the $y$-axis is the return.

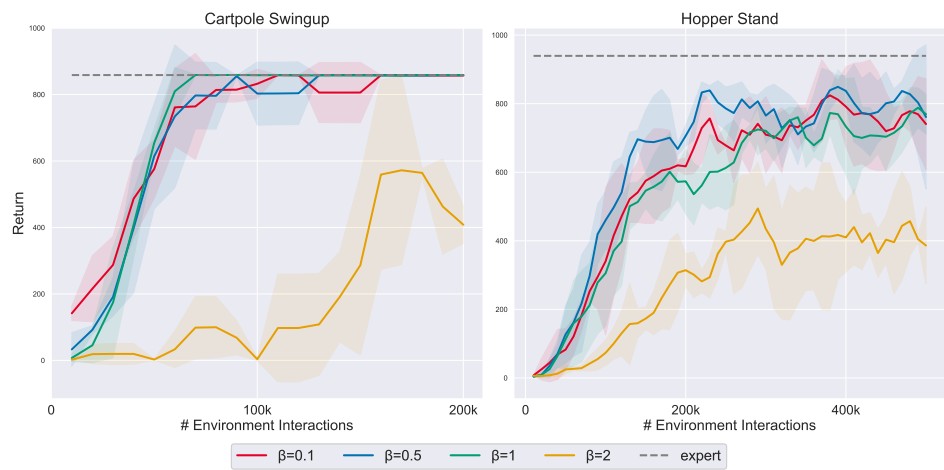

Figure 6: Learning curves of CoPT-AIL with different regularization coefficient $\beta$. Here the $x$-axis is the number of environment interactions and the $y$-axis is the return.

### D.3 SENSITIVITY ANALYSIS

#### D.3.1 SENSITIVITY ANALYSIS ON THE NUMBER OF DEMONSTRATIONS

We conduct a sensitivity analysis on the number of expert trajectories $N$ in $\mathcal{D}^E$. Specifically, we evaluate CoPT-AIL and other baselines with $N \in \{5, 10, 15, 20\}$, and the corresponding learning curves are presented in Figure 5. We observe that CoPT-AIL matches or surpasses the convergence rates of prior SOTA AIL methods across different number of expert trajectories.

#### D.3.2 SENSITIVITY ANALYSIS ON THE REGULARIZATION COEFFICIENT

We also conduct a sensitivity analysis on the regularization coefficient $\beta$ in Eq. (14). Specifically, we evaluate CoPT-AIL with $\beta \in \{0.1, 0.5, 1, 2\}$, and the corresponding learning curves are presented in Figure 6. CoPT-AIL maintains strong performance for $\beta \in [0.1, 1]$. When $\beta$ is large (e.g., $\beta = 2$), performance degrades because the regularization term begins to dominate the reward-training objective and can misguide the reward model.

### D.4 CONSISTENCY BETWEEN REWARD PRE-TRAINING AND REWARD FINE-TUNING

In CoPT-AIL, reward learning consists of two stages: BC-based reward pre-training and AIL-based reward fine-tuning. We investigate whether any "unlearning" occurs between these stages.

To this end, we first evaluate the gradient cosine similarity between the BC and AIL reward objectives. The resultant curves are shown in Figure 7. We observe that the gradient cosine similarity maintains a high value $\geq 0.8$ (with a maximal possible value of 1) throughout online training. This indicates that the BC and AIL reward objectives are closely aligned rather than working against each other.

Besides, the ultimate goal of reward learning is to produce a reward function that assigns high values to expert data and low values to non-expert data. To verify that the BC and AIL objectives contribute synergistically to this goal, we track the reward gap between expert data and replay buffer data $\mathbb{E}_{(s,a)\sim\mathcal{D}^E}[r_\phi(s,a)] - \mathbb{E}_{(s,a)\sim\mathcal{D}^{\text{replay}}}[r_\phi(s,a)]$ throughout the entire reward learning process. The curves are reported in Figure 8. The first 100K reward gradient steps correspond to the reward pre-training procedure while the remaining gradient steps correspond to the reward fine-tuning procedure. Importantly, when transitioning from pre-training to fine-tuning, the reward gap stays at its previously high level rather than decreasing, indicating that no noticeable unlearning occurs during the switch. This further confirms that the BC and AIL reward objectives operate synergistically, not adversarially.

### D.5 LANDSCAPE OF THE REWARD MODEL

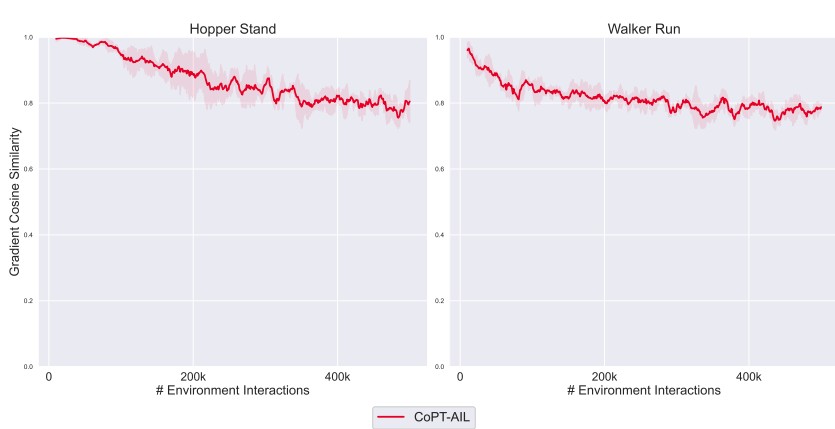

Figure 7: Gradient cosine similarity of reward pre-training and fine-tuning objectives in CoPT-AIL. Here the $x$-axis is the number of environment interactions and the $y$-axis is the gradient cosine similarity.

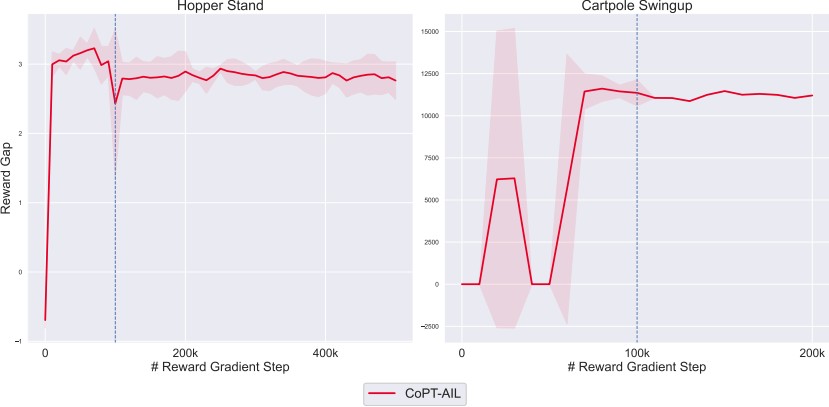

Figure 8: Reward gaps between expert data and replay data during the entire reward learning procedure in CoPT-AIL. Here the $x$-axis is the number of reward gradient steps and the $y$-axis is the reward gap $\mathbb{E}_{(s,a)\sim\mathcal{D}^{\mathrm{E}}}[r(s,a)] - \mathbb{E}_{(s,a)\sim\mathcal{D}^{\mathrm{replay}}}[r(s,a)]$. The first 100K reward gradient steps correspond to the reward pre-training procedure while the remaining gradient steps correspond to the reward fine-tuning procedure.

In Figure 1, CoPT-AIL shows an initial performance plateau on `Walker Run`. One may hypothesize that reward pre-training might produce a non-smooth reward landscape, potentially explaining the short plateau at the beginning of training. To evaluate this concern, we directly measure the Lipschitz coefficient of the reward model (with respect to the state–action input) in Walker-Run for two methods: with reward pre-training and without reward pre-training. The results are shown in Figure 9. We find that the Lipschitz coefficients in the two settings are highly comparable, indicating that reward pre-training does not introduce additional non-smoothness into the reward landscape. Moreover, the reward model under CoPT-AIL maintains a relatively small Lipschitz coefficient throughout online training. This is likely aided by the introduced exponential-based regularizer, which helps stabilize reward updates. Moreover, standard regularization methods (e.g., gradient penalties) can further address potential non-smoothness. Overall, we believe that reward pre-training offers a beneficial initialization without introducing notable non-smoothness.

To further understand the cause of this behavior, we investigate whether it arises from the specific set of demonstrations. We re-sample the same number of expert trajectories from the full expert dataset and re-run all methods using these new demonstrations. The resulting learning curves are shown in Figure 10. The plateau no longer appears, indicating that the behavior observed in Figure 1 is primarily due to the original demonstration set rather than the reward pre-training mechanism in CoPT-AIL.

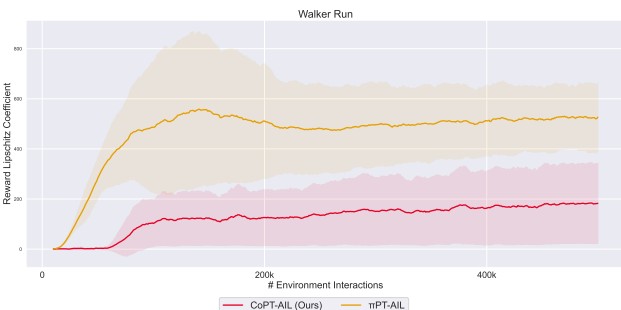

Figure 9: Lipschitz coefficients of the reward model in CoPT-AIL on Walker Run. Here the $x$-axis is the number of environment interactions and the $y$-axis is the Lipschitz coefficients.

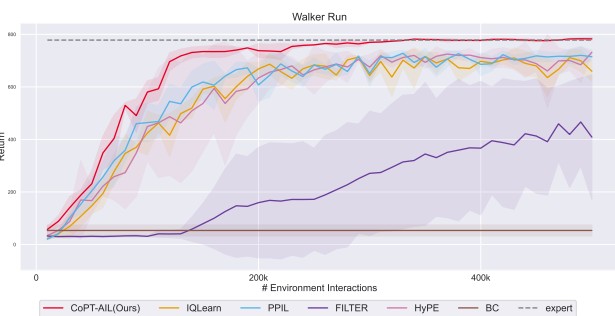

Figure 10: Learning curves of different methods on Walker Run. Here the $x$-axis is the number of environment interactions and the $y$-axis is the return.

### D.6  EMPIRICAL EVALUATIONS OF RELATIVE POLICY EVALUATION ERROR

The main theoretical prediction from our theory is that the proposed reward pre-training mechanism can reduce the relative policy evaluation error. Here we empirically validate this theoretical prediction by comparing the relative policy evaluation error under the pre-trained reward model and randomly initialized reward model. Recall that the relative policy evaluation error is defined as $(V_{r^\star}^{\pi^E} - V_{r^\star}^{\pi^1}) - (V_r^{\pi^E} - V_r^{\pi^1})$, where $\pi^1$ is the initial policy. Here we use the ground-truth reward defined in DMC tasks as $r^\star$ and approximate the policy value through Monte Carlo estimation using 20 trajectories. Below, we report the normalized relative policy evaluation error divided by the horizon length.

| Task | Walker Stand | Walker Run | Cartpole Swingup | Hopper Hop | Hopper Stand | Finger Spin |
|------|-------------|-----------|------------------|-----------|--------------|-------------|
| CoPT-AIL | 1.51 | 1.49 | -10.77 | 0.63 | 1.84 | 1.93 |
| Baseline | 56.76 | 52.44 | 2.69 | 35.21 | 7.30 | 27.66 |

Table 7: Relative policy evaluation error of CoPT-AIL and the baseline across DMControl tasks.

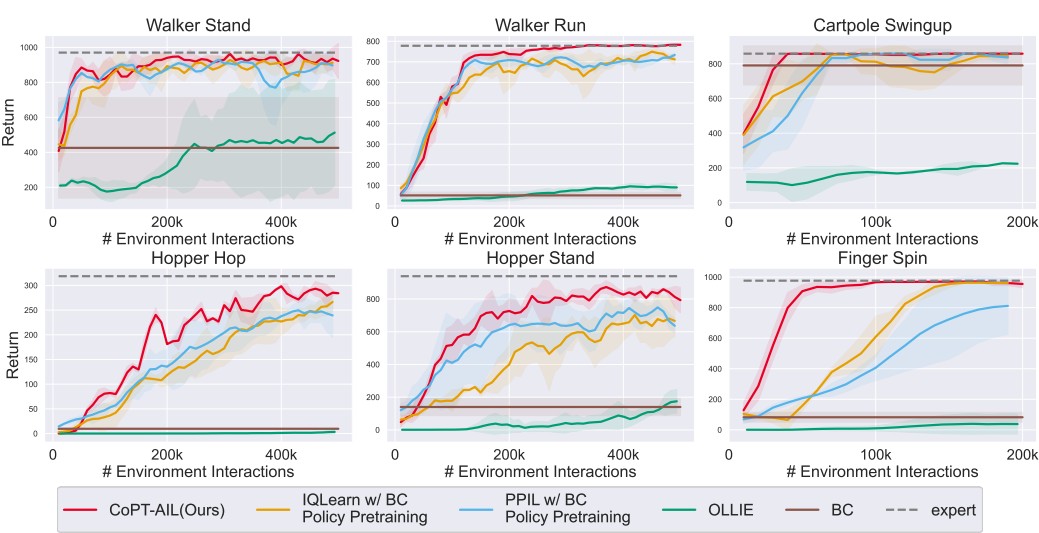

Figure 11: Comparisons against baselines with pre-training. Here the $x$-axis is the number of environment interactions and the $y$-axis is the return.

### D.7 COMPARISONS AGAINST BASELINES WITH PRE-TRAINING

We compare CoPT-AIL against prior state-of-the-art AIL methods augmented with pre-training. OLLIE (Yue et al., 2024) includes both policy and reward pre-training components by design. In contrast, most existing SOTA AIL algorithms (e.g., IQLearn (Garg et al., 2021) and PPIL (Viano et al., 2022)) do not originally incorporate any pre-training. We therefore add BC policy pre-training to both IQLearn and PPIL. The resulting learning curves are shown in Figure 11. Compared against OLLIE and baselines incorporated with BC policy pre-training, CoPT-AIL consistently achieves faster convergence in the evaluated tasks. This corroborates our theoretical analysis that the proposed reward pre-training mechanism yields substantial efficiency improvements.

## E USE OF LARGE LANGUAGE MODELS

A large language model (LLM) was utilized during the preparation of this manuscript solely to polish the writing. The tool was used to improve grammar, clarity, and readability. The LLM was not used for any substantive aspects of the research, such as literature retrieval, discovery, or the generation of research ideas. All intellectual content, analysis, and conclusions are the original work of the authors.

