# OpenReview forum: "Provably Efficient Policy-Reward Co-Pretraining for Adversarial Imitation Learning"
_ICLR.cc/2026/Conference — Submitted to ICLR 2026_

### Official Review · Reviewer_uaDi · 2025-10-29

**Soundness:** 3
**Presentation:** 2
**Contribution:** 3
**Rating:** 6
**Confidence:** 4

**Summary:**

This paper investigates the theoretical foundations of pretraining in adversarial imitation learning (AIL). The authors first theoretically reveal the fundamental reason why policy-only pretraining fails: reward error becomes the bottleneck. Based on this insight, they propose CoPT-AIL, which leverages reward shaping theory to jointly pretrain both policy and reward through a single BC procedure. Theoretically, they prove that when the number of expert demonstrations satisfies N ≳ C√(|S||A|H²/K), the method achieves a tighter imitation gap bound compared to standard AIL. Experiments validate the effectiveness of the approach on 6 DMControl tasks.

**Strengths:**

1. Clear motivation and important problem: The paper addresses a practical pain point in AIL—the requirement for extensive online interactions—and provides theoretical explanation for why existing policy-only pretraining approaches have limited effectiveness. The problem formulation is valuable.

2. Solid theoretical contributions: The error decomposition in Proposition 1 clearly reveals the bottleneck role of reward error. The use of reward shaping theory to circumvent the reward ambiguity issue is clever. Theorem 1 provides the first theoretical guarantee for the benefits of pretraining in AIL, filling an important theoretical gap

3. Simple and elegant method: The derivation of r̃*(s,a) = log π^E(a|s) from the expert's soft-optimality property is natural, enabling joint pretraining of policy and reward through a single BC procedure. The method is simple to implement and computationally efficient.
Reasonable experimental design: The paper includes comparisons with multiple SOTA methods, and ablation studies adequately validate the necessity of joint pretraining.

**Weaknesses:**

The theoretical analysis is explicitly limited to tabular MDPs (using |S|, |A| notation), which the authors acknowledge in the conclusion: "this work focuses on the standard tabular setup". However, the experiments use continuous state-action spaces in DMControl tasks, which do not match the theoretical setting. There is no discussion on how to extend the theory to function approximation and continuous spaces. Moreover, the constant C = max d^{πBC}(s)/d^{πE}(s) can be large when BC quality is poor, potentially weakening the practical utility of the theoretical results.

Setting r¹(s,a) = log π^BC(a|s) depends on BC quality, which may be poor when demonstrations are limited. For continuous action spaces, Algorithm 3 does not specify how to compute log π^BC(a|s). If π^BC deviates significantly from π^E, the pretrained reward may mislead subsequent training.

The paper only tests on 6 DMControl tasks, lacking environmental diversity. It is missing sensitivity analysis with respect to the number of expert demonstrations N (which the theoretical condition depends on). Eq. 14 introduces regularization term β exp(-r), but Table 1 does not report the value of β, and there are no ablation experiments to validate its importance. Additionally, there is no analysis of whether the condition N ≳ C√(|S||A|H²/K) is satisfied in the experiments.

The derivation from r*(s,a) to r̃*(s,a) in Section 4.1 is somewhat abrupt; the choice of shaping function Φ could be explained more explicitly. Implementation details for baselines in the experimental section are insufficient, such as specific hyperparameter settings for each method. While Proposition 2 is based on the classic result of Ng et al. (1999), its application in the current problem is reasonable.

**Questions:**

1. Your theory explicitly targets tabular MDPs (original text: "standard tabular setup"), but experiments use continuous state-action spaces. Can you (a) discuss how the theory extends to function approximation settings, or (b) at least provide heuristic analysis or experimental validation of theoretical predictions in continuous spaces?

2. For continuous actions, how do you compute r¹(s,a) = log π^BC(a|s)? Do you assume π^BC has a specific parametric form (e.g., Gaussian policy)? Algorithm 3 should explicitly clarify this implementation detail.

3. Practical impact of constant C: In your experimental tasks, what are typical values of C = max d^{πBC}/d^{πE}? How does this affect the tightness of the theoretical results?

4. Regarding the regularization coefficient β in Eq. 14: (a) Why is the value of β not listed in Table 1? (b) Can you provide ablation experiments to validate the importance of this regularization term?

5. The theory requires N ≳ C√(|S||A|H²/K). Can you verify whether this condition is satisfied in your experiments, or show sensitivity analysis of algorithm performance with respect to N?

6. Can you provide experimental comparisons with Jena et al. 2021 "Augmenting GAIL with BC for Sample Efficient Imitation Learning"? This method also combines BC and AIL, and comparison would more clearly demonstrate the advantages of CoPT-AIL.

7. Can you discuss the dependence of the method on the soft-optimality assumption for the expert policy (Eq. 1)? If the expert is suboptimal, does the derivation of r̃*(s,a) = log π^E(a|s) still hold?

Note: Satisfactory answers to questions 1-5 would strengthen the paper significantly and could raise my recommendation to Accept. Questions 6-7 are less critical but would further enhance the contribution.

---

> ### Author Response · Authors · 2025-11-21
> **Response (Part I)**
>
> We appreciate your time to review and provide positive feedback for our work.
>
> **Question 1**:  The theoretical analysis is explicitly limited to tabular MDPs (using |S|, |A| notation), which the authors acknowledge in the conclusion: "this work focuses on the standard tabular setup". However, the experiments use continuous state-action spaces in DMControl tasks, which do not match the theoretical setting. There is no discussion on how to extend the theory to function approximation and continuous spaces.
>
> **Answer 1**: Our theoretical analysis could be extended to the linear function approximation set-up with continuous spaces [R1]. In this set-up, the reward and policy are linear functions regarding a certain feature function.
> \begin{align*}
> r_h (s, a) = \langle \theta_h, \phi (s, a) \rangle,  \pi_h (a|s) = \frac{\exp (\langle w_h, \phi (s, a) \rangle)}{\sum_{a^\prime \in \mathcal{A}} \exp (\langle w_h, \phi (s, a^\prime) \rangle)}.
> \end{align*}
> Here $\phi: \mathcal{S} \times \mathcal{A} \rightarrow \mathbb{R}^d$ is the feature function satisfying $\\| \phi (s, a) \\|\_2 \leq 1, \forall (s, a) \in \mathcal{S} \times \mathcal{A}$  and $\\{ \theta_h \\}\_{h=1}^H, \{ w_h \}_{h=1}^H$ are parameters satisfying $\| \theta_h \|_2 \leq \sqrt{d}, \\| w_h \\|_2 \leq \sqrt{d}$, $\forall h \in [H]$.
> First, we would like to point out that the choice of tabular or linear function approximation set-up primarily influences the analysis for statistical errors arising from finite demonstrations. The main difference is to replace the dependence of $|\mathcal{S}| |\mathcal{A}|$ with the feature dimension $d$. Below, we outline how each theoretical result extends to the linear function approximation setting.
>
> 1. **Extension of Proposition 1.**
>      * **Extension of Eq.(4).**  We need to re-analyze the statistical error in reward error, which affects the second term in the RHS of Eq.(4). Specifically, we aim to analyze the statistical error $| V^{\pi^{\text{E}}}\_{r} - \widehat{V}^{\pi^{\text{E}}}\_{r}  |$, where $\widehat{V}^{\pi^{\text{E}}}\_{r}$ is an estimation of $V^{\pi^{\text{E}}}\_{r}$ based on $\mathcal{D}^{\text{E}}$. We can relate the statistical error in the value space to that in the feature space.
>   \begin{align*}
>   |V^{\pi^{\text{E}}}\_{r} - \widehat{V}^{\pi^{\text{E}}}\_{r}| \leq d \sum\_{h=1}^H  \\| \mathbb{E}\_{(s_h, a_h) \sim d^{\pi^{\text{E}}}\_h } [\phi (s_h, a_h)] - \mathbb{E}\_{(s_h, a_h) \sim \widehat{d}^{\pi^{\text{E}}}\_h } [\phi (s_h, a_h)]     \\|\_{\infty}
>   \end{align*}
>   With Hoeffding's inequality and a union bound, we can upper bound the statistical error in the feature space. With probability at least $1-\delta$,
>   \begin{align*}
>   d \sum\_{h=1}^H  \\| \mathbb{E}\_{(s_h, a_h) \sim d^{\pi^{\text{E}}}\_h } [\phi (s_h, a_h)] - \mathbb{E}\_{(s_h, a_h) \sim \widehat{d}^{\pi^{\text{E}}}\_h } [\phi (s_h, a_h)]     \\|\_{\infty}  \leq H \sqrt{\frac{\log (dH/\delta)}{N}}.
>   \end{align*}
>   Combining the above two inequalities yields that $| V^{\pi^{\text{E}}}\_{r} - \widehat{V}^{\pi^{\text{E}}}\_{r} | \leq H d \sqrt{\frac{\log (dH/\delta)}{N}}$. Then the second term in the RHS of Eq.(4) becomes $\mathcal{O} (H d \sqrt{\frac{\log (dH/\delta)}{N}})$.
>    *   **Extension of Eq.(5).** We need to re-analyze the KL divergence between the expert policy and BC policy with linear function approximation (i.e., the third term in the RHS of Eq.(4)). In particular, [R2, R3] provide the theoretical analysis of BC under function approximation. By leveraging their results, we could prove that the KL divergence can be bounded by $\widetilde{\mathcal{O}} (\frac{d H}{N})$. Then the last term in the RHS of Eq.(5) becomes $\widetilde{\mathcal{O}} ( \sqrt{\frac{d H^4}{K N}})$.
> 2. **Extension of Proposition 2.** Proposition 2 holds as stated. Its proof is conducted entirely in function space and does not depend on whether the underlying MDP is tabular or continuous.
> 3. **Extension of Theorem 1.** To extend Theorem 1, the key is to analyze the "relative policy evaluation error". Built upon Proposition 2, we can relate the "relative policy evaluation error" with the KL divergence between the expert policy and BC policy. Applying BC guarantees under linear function approximation [R2, R3] again gives an upper bound of $\widetilde{\mathcal{O}} (\frac{d H^2}{N})$, which in turn modifies the last term in the RHS of Eq.(7) to $\widetilde{\mathcal{O}} ( \sqrt{\frac{d H^4 }{K N}})$.

---

> ### Author Response · Authors · 2025-11-21
> **Response (Part II)**
>
> **Question 2**: Moreover, the constant $C = \max d^{\pi^{\text{BC}}}_h (s) / d^{\pi^{\text{E}}}_h (s)$ can be large when BC quality is poor, potentially weakening the practical utility of the theoretical results.
>
> **Answer 2**: First, we argue that the appearance of this distribution ratio is mainly due to the fact that the "relative policy evaluation error" depends on the value of **the initial policy (BC policy)**. Specifically, the key quantity we analyze is the relative policy evaluation error, which is inherently defined with respect to **the initial policy, i.e., the BC policy**. Since reward pre-training is performed using expert demonstrations (the only data in the offline stage), the pre-trained reward is guaranteed to be accurate on **the expert distribution**. Consequently, when evaluating the reward under the BC policy, a distribution shift arises between $d^{\pi^{\text{BC}}}_h$ and $d^{\pi^{\text{E}}}_h$. **The dependence on the ratio is therefore structurally unavoidable rather than an artifact of looseness in our analysis.**
>
>
>
>
> To further address your concern, we empirically calculate this distribution ratio in our tasks. We choose the Cartpole Swingup task, where the state-action space is low-dimensional such that accurate distribution ratio estimation is allowed. We apply a discriminative method in the literature of GAN to directly learn the distribution ratio. In particular, we learn a discriminator in the following way.
> \begin{align*}
> \max\_{D: \mathcal{S} \rightarrow (0, 1)} \mathbb{E}\_{s \sim \widehat{d^{\pi^{\text{E}}}_h} (\cdot) } [\log (D(s))] + \mathbb{E}\_{s \sim \widehat{d^{\pi^{\text{BC}}}_h} (\cdot) } [\log (1-D(s))] .
> \end{align*}
> Here $\widehat{d^{\pi^{\text{E}}}_h} (\cdot)$ and $\widehat{d^{\pi^{\text{BC}}}_h} (\cdot)$ are empirical distributions based on finite samples. It is direct to obtain that the optimal discriminator satisfies that $D^* (s) = \widehat{d^{\pi^{\text{E}}}_h} (s) / (\widehat{d^{\pi^{\text{E}}}_h} (s)  + \widehat{d^{\pi^{\text{BC}}}_h} (s) )$. Then we can approximate the distribution ratio by $(1-D^{\star}(s)) / D^{\star} (s)$. In this way, we obtain that $C$ maintains a moderate value $4.575$, suggesting that the distribution shift is well-controlled and does not undermine the practical relevance of our theoretical results.
>
> **Question 3**: Setting $r^1 (s, a) = \log (\pi^{\text{BC}} (a|s))$ depends on BC quality, which may be poor when demonstrations are limited. For continuous action spaces, Algorithm 3 does not specify how to compute $\log (\pi^{\text{BC}} (a|s))$. If $\pi^{\text{BC}}$ deviates significantly from $\pi^{\text{E}}$, the pretrained reward may mislead subsequent training.
>
> **Answer 3**: We agree that the quality of the pretrained reward indeed depends on the quantity of demonstrations. This dependence is natural: **reward pre-training is fundamentally an attempt to extract as much information as possible from the offline expert dataset to provide a good initialization for AIL**. From this information-theoretic perspective, it is expected that the performance of any pretrained reward model scales with the amount of available demonstrations.
>
> Importantly, all IL methods are affected by demonstration quantity. The key question is therefore which method uses a fixed dataset more effectively. Our argument is that **CoPT-AIL makes superior use of the demonstrations to construct a good reward initialization.** We support this claim from both theoretical and empirical perspectives:
>
>
> * Theoretically, as discussed in Section 4.1, in order to reduce the "relative policy evaluation error", the key is to pre-train toward the shaping reward $\log (\pi^{\text{E}} (a|s))$, which is the log-probability of the expert policy. This reduces to the problem of imitating the expert policy. From the perspective of imitating policies, [R4] has proved that BC is already minimax optimal in the offline setting. As such, we conjecture that our reward pre-training method, which leverages the BC policy, may have reached the performance limit for reward pre-training.
> * Empirically, we evaluate CoPT-AIL with various numbers of expert trajectories. The results are shown in Figure 5 in the revision. We observe that CoPT-AIL consistently outperforms prior SOTA AIL methods across all data regimes, demonstrating its robustness even when demonstrations are limited.
>
>
>
> For continuous action spaces, we parameterize the policy with a Gaussian distribution, $\pi_{\theta} (a|s) = \mathcal{N} (a; \mu_{\theta} (s), \sigma_{\theta} (s))$ and calculate the log-probability of the Gaussian distribution. We have clarified this point in the revision.

---

> ### Author Response · Authors · 2025-11-21
> **Response (Part III)**
>
> **Question 4**: The paper only tests on 6 DMControl tasks, lacking environmental diversity. It is missing sensitivity analysis with respect to the number of expert demonstrations N (which the theoretical condition depends on). Eq. 14 introduces regularization term $\beta \exp (-r)$, but Table 1 does not report the value of $\beta$, and there are no ablation experiments to validate its importance. Additionally, there is no analysis of whether the condition $N \succsim C \sqrt{|\mathcal{S}| |\mathcal{A}| H^2/K}$ is satisfied in the experiments.
>
> **Answer 4**: We address the reviewer’s concerns regarding environmental diversity, sensitivity analyses, and the theoretical condition on N as follows:
> 1. **Environmental Diversity**. To mitigate your concerns, we test two additional DMControl tasks Quadruped Run and Cheetah Run. The learning curves are displayed in Figure 3 in the revision. CoPT-AIL exhibits a consistently faster convergence rate than prior SOTA AIL methods on these two tasks. This validates our theoretical analysis that the proposed reward pre-training mechanism yields a better imitation gap in CoPT-AIL.
> 2. **Sensitivity Analysis of $N$**. We evaluate CoPT-AIL with various numbers of expert trajectories. The results are shown in Figure 5 in the revision. We observe that CoPT-AIL matches or surpasses the convergence rates of prior SOTA AIL methods across different number of expert trajectories.
> 3. **Sensitivity Analysis of $\beta$**. We set the regularization coefficient $\beta = 1$ across all tasks. Besides, we conduct a sensitivity analysis on $\beta \in \\{0.1, 0.5, 1, 2 \\}$. The results are shown in Figure 6 in the revision. We observe that CoPT-AIL is not sensitive to the value of $\beta$ and maintains strong performance for $\beta \in [0.1, 1]$. When $\beta$ is large (e.g., $\beta = 2$), performance degrades because the regularization term begins to dominate the reward-training objective and can misguide the reward model.
> 4. **Condition on $N$**. It is difficult to calculate this condition in experiments. We emphasize that this condition serves as a **sufficient (not necessary)** condition ensuring that CoPT-AIL achieves a better imitation-gap bound in theory. Its primary purpose is to provide qualitative guidance: CoPT-AIL benefits more when a reasonable amount of expert data is available. We have clarified this point in the revision.
>
> **Question 5**: The derivation from $r^\star (s,a)$ to $\widetilde{r}^\star (s,a)$ in Section 4.1 is somewhat abrupt; the choice of shaping function Φ could be explained more explicitly. Implementation details for baselines in the experimental section are insufficient, such as specific hyperparameter settings for each method. While Proposition 2 is based on the classic result of Ng et al. (1999), its application in the current problem is reasonable.
>
> **Answer 5**: Thanks for your suggestion. In the revision, we have explicitly explained the choice of $\Phi$ and have added the implementation details for baselines.
>
> **Question 6**: Your theory explicitly targets tabular MDPs (original text: "standard tabular setup"), but experiments use continuous state-action spaces. Can you (a) discuss how the theory extends to function approximation settings, or (b) at least provide heuristic analysis or experimental validation of theoretical predictions in continuous spaces?
>
> **Answer 6**: First, we could extend our theory to the function approximation setting; please refer to **Answer 1** for the detailed discussion.
>
> Besides, the main theoretical prediction from our theory is that the proposed reward pre-training mechanism can reduce the "relative policy evaluation error". Here we empirically validate this theoretical prediction by comparing the "relative policy evaluation error" under the pre-trained reward model and the randomly initialized reward model. Recall that the "relative policy evaluation error" is defined as $(V^{\pi^{\text{E}}}\_{r^\star} - V^{\pi^{1}}\_{r^\star}) - (V^{\pi^{\text{E}}}\_{r} - V^{\pi^{1}}\_{r})$, where $\pi^1$ is the initial policy. Here we use the ground-truth reward defined in DMC tasks as $r^\star$ and approximate the policy value through Monte Carlo estimation using 20 trajectories. Below, we report the normalized "relative policy evaluation error" divided by the horizon length.
>
> | Task | Walker Stand  | Walker Run | Cartpole Swingup |Hopper Hop |Hopper Stand |Finger Spin |
> | -------- | -------- | -------- |-------- |-------- |-------- |-------- |
> |CoPT-AIL    |    1.51 |  1.49 |  -10.77| 0.63  | 1.84| 1.93|
> | Baseline   | 56.76   | 52.44 | 2.69  | 35.21 | 7.30| 27.66 |
>
> Across all six tasks, the relative policy evaluation error under CoPT-AIL is substantially smaller than under the baseline, clearly supporting our theoretical prediction.

---

> ### Author Response · Authors · 2025-11-21
> **Response (Part IV)**
>
> **Question 7**: For continuous actions, how do you compute $r^1 (s, a) = \log (\pi^{\text{BC}} (a|s))$? Do you assume $\pi^{\text{BC}}$ has a specific parametric form (e.g., Gaussian policy)? Algorithm 3 should explicitly clarify this implementation detail.
>
> **Answer 7**: Yes. We parameterize the policy with a Gaussian distribution, i.e., $\pi_{\theta} (a|s) = \mathcal{N} (a; \mu_{\theta} (s), \sigma_{\theta} (s))$ and calculate the log-probability of the Gaussian distribution. We have clarified this detail in the revision.
>
>
>
> **Question 8**: Practical impact of constant C: In your experimental tasks, what are typical values of $C = \max d^{\pi^{\text{BC}}}_h (s) / d^{\pi^{\text{E}}}_h (s)$? How does this affect the tightness of the theoretical results?
>
> **Answer 8**: Please refer to **Answer 2** for a detailed explanation.
>
>
>
> **Question 9**: Regarding the regularization coefficient $\beta$ in Eq. 14: (a) Why is the value of $\beta$ not listed in Table 1? (b) Can you provide ablation experiments to validate the importance of this regularization term?
>
> **Answer 9**: Please refer to the part "**Sensitivity Analysis of $\beta$**" in **Answer 4**.
>
>
>
>
> **Question 10**: The theory requires $N \succsim C \sqrt{|\mathcal{S}| |\mathcal{A}| H^2/K}$. Can you verify whether this condition is satisfied in your experiments, or show sensitivity analysis of algorithm performance with respect to N?
>
> **Answer 10**: Please refer to the parts "**Condition on $N$**" and "**Sensitivity Analysis on $N$**" in **Answer 4**.
>
>
>
>
> **Question 11**: Can you provide experimental comparisons with Jena et al. 2021 "Augmenting GAIL with BC for Sample Efficient Imitation Learning"? This method also combines BC and AIL, and comparison would more clearly demonstrate the advantages of CoPT-AIL.
>
> **Answer 11**: We evaluate the recommended baseline GAIL+BC on Hopper Stand and Cartpole Swingup. The learning curves are shown in Figure 4 in the revision. We observe that CoPT-AIL matches or surpasses the convergence speed of this baseline in both tasks.
>
>
>
> **Question 12**: Can you discuss the dependence of the method on the soft-optimality assumption for the expert policy (Eq. 1)? If the expert is suboptimal, does the derivation of $\widetilde{r}^\star (s,a) = \log \pi^{\text{E}} (a|s)$ still hold?
>
> **Answer 12**: If the expert is sub-optimal, $\widetilde{r}^\star (s,a) = \log (\pi^{\text{E}} (a|s))$ does not hold. In general, if the expert is sub-optimal, we cannot make any inference on the underlying reward without additional information, because the expert policy could be an arbitrary distribution. Nevertheless, if additional information such as the degree of sub-optimality [R5] is available, we could infer the underlying reward and extend CoPT-AIL to this set-up. We have discussed this promising direction in the revision.
>
>
> References:
>
> [R1] Chi Jin et al. "Provably efficient reinforcement learning with linear function approximation." COLT 2020.
>
> [R2] Daniil Tiapkin et al. "Demonstration-regularized RL." ICLR 2024.
>
> [R3] Dylan J Foster et al. "Is behavior cloning all you need? understanding horizon in imitation learning." NeurIPS 2024.
>
> [R4] Nived Rajaraman et al. "Toward the fundamental limits of imitation learning." NeurIPS 2020.
>
> [R5] Riccardo Poiani et al. "Inverse Reinforcement Learning with Sub-optimal Experts". arXiv:2401.03857.
>
> ---
>
> We hope that the above answers can address your concerns satisfactorily. We would be grateful if you could re-evaluate our paper based on the above responses. We are also willing to address any further concerns, if possible.

---

> ### Author Response · Authors · 2025-11-28
> **Follow-up on Rebuttal: Addressed Questions 1, 2, 4, 6, 7 for Potential Score Increase to Acceptance**
>
> Dear Reviewer uaDi,
>
> Thank you again for your constructive review. We are writing to ensure you have had a chance to review our response, specifically regarding the points you identified as key to raising your evaluation.
>
> You mentioned that **you would consider raising the score to acceptance if we satisfactorily addressed Questions 1, 2, 4, 6, and 7 (corresponding to Questions 1-5 in your original review).** We represent a summary of how we have fully resolved these specific concerns:
>
> - **Theory with Function Approximation and Empirical Validation of Theoretical Predictions (Q1 & Q6):**
>     - **Theory:** We have demonstrated that our theoretical analysis can be extended to the **linear function approximation setting** in continuous spaces (**Answer 1**).
>     - **Empirical Validation:** We empirically measured the "relative policy evaluation error" (**Answer 6 and Table 7 in the revision**), showing CoPT-AIL significantly outperforms baselines, directly validating our theoretical predictions.
> - **Distribution Ratio C (Q2):**
>     - **Theoretical Justification:** We clarified that the dependence on this ratio is **intrinsic to the problem structure** (evaluation relative to the initial BC policy) rather than an artifact of analysis looseness (**Answer 2**).
>     - **Empirical Calculation:** Using a discriminator-based estimation, we found that C maintains a **moderate value of ≈4.575 (Answer 2**). This confirms that this distribution ratio does not undermine the practical relevance of our theory.
> - **Experimental Completeness (Q4):**
>     - **Diversity:** We added **two new tasks** (Quadruped Run & Cheetah Run), showing consistent improvements of our method CoPT-AIL over prior SOTA (**Figure 3 in the revision**).
>     - **Sensitivity:** We conducted sensitivity analyses on demonstration count N (**Figure 5 in the revision**) and regularization coefficient $\beta$ (**Figure 6 in the revision**), demonstrating the robustness of CoPT-AIL across varying data regimes and hyperparameter settings.
> - **Implementation Details (Q7):**
>     - We clarified that for continuous actions, we use a Gaussian policy parameterization to compute log-probabilities (**Answer 7**).
>
> With these concerns resolved, **we hope the paper now meets the bar for the higher score you suggested**. In light of these comprehensive revisions, **we would greatly appreciate it if you could re-evaluate our work and consider updating the score to reflect these improvements.**
>
> Best regards,
>
> The Authors

---

### Official Review · Reviewer_zfqh · 2025-10-29

**Soundness:** 2
**Presentation:** 3
**Contribution:** 2
**Rating:** 2
**Confidence:** 4

**Summary:**

This paper proposes a theoretical analysis on the effects of pre-training both policy and reward function before using AIL.

**Strengths:**

The paper proposes an interesting theoretical analysis of AIL with policy pretraining alone. Furthermore, it quantifies the regret arising from not pre-training the reward function (Proposition 1, Eq. (5)) and provides an improved bound to showcase the effects of pre-training the reward function.

**Weaknesses:**

1. The paper’s contributions are mainly incremental and overlap with existing analyses in the AIL literature.

2. Previous theoretical work on AIL has already highlighted its reliance on reward updates (e.g., Lemma 3.2 and Theorem 4.5 in [1]).

3. The reward pretraining contribution is not entirely novel, as the connection between policy pretraining and reward learning has already been explored in [2].

4. The choice of experimental baselines is not fully justified. Given the close connection to OLLIE in [2], which similarly combines pretraining with AIL, including OLLIE as a baseline would have been more appropriate. Since CoPT-AIL emphasizes stronger theoretical analysis, the comparison against OLLIE would have been particularly interesting to demonstrate the practical benefits of the added theory.

5.  The results in Fig. 2 are not fully convincing and do not clearly substantiate the paper’s claims regarding CoPT-AIL’s superiority over prior AIL approaches.

**References**:

[1] Chen Y, Giammarino V, Queeney J, Paschalidis IC. Provably efficient off-policy adversarial imitation learning with convergence guarantees. arXiv preprint arXiv:2405.16668, 2024.

[2] Yue S, Hua X, Ren J, Lin S, Zhang J, Zhang Y. OLLIE: Imitation learning from offline pretraining to online finetuning. arXiv preprint arXiv:2405.17477, 2024.

**Questions:**

1. Setting aside the theoretical analysis, what are the main differences between OLLIE in [2] and CoPT-AIL? Are there reasons why one approach should be preferred over the other?

2. Can you make concrete claims about the quality and quantity of data required for CoPT-AIL to succeed? In particular, how does it perform with suboptimal demonstrations or with only a small number of expert trajectories?

---

> ### Author Response · Authors · 2025-11-21
> **Response (Part I)**
>
> We appreciate your time and effort for reviewing our work. Below, we begin with a general response to the key issues raised, before presenting our specific point-by-point answers.
>
> **General Response:** We identify that the reviewer’s primary concern lies in assessing the significance of our contributions relative to prior AIL theory and to the empirical work OLLIE [2].
>
> 1. **Comparison with prior AIL theory (Questions 1, 2)**.
>
>     Prior AIL theory [R1-R4] analyzes AIL initialized with **random policies and rewards**, and focuses on **the iterative online learning process** in AIL. To the best of our knowledge, none of these works (i) theoretically identify or highlight the error from initial policies and rewards, or (ii) provide theoretical guarantees for any pre-training strategy.
>
>     **In contrast, this work provides the first systematic theoretical treatment of reward pre-training in AIL.** Specifically, we (i) theoretically reveal the critical impact of reward pre-training on AIL performance (Section 3), (ii) introduce a new reward pre-training mechanism grounded in shaping-reward invariance (Section 4.1), and (iii) provide theoretical guarantees for the proposed method (Section 4.2). Please refer to **Answer 1** for details.
> 2.  **Comparison with OLLIE (Questions 3, 4, 6)**.
>     As an empirical work, OLLIE [2] attributes the failure of AIL with BC-pre-trained policies to random reward initialization based on **experimental observations**, without offering a theoretical explanation. In contrast, this work makes **theoretical** advances toward a principled understanding of reward pre-training in AIL. Methodologically, this paper differs from OLLIE in three fundamental aspects: **problem set-up, methodological design principle, and the resultant learning algorithm.**
>     * **Problem set-up.** This paper studies the standard IL set-up where the learner has access to an expert dataset (Section 2). In contrast, OLLIE studies IL with a supplementary dataset, where the learner has an expert dataset and an additional supplementary dataset.
>     * **Methodological design principle.** Our method is motivated by reducing the relative policy evaluation error, identified through a rigorous theoretical analysis of AIL with BC-pretrained policies (Proposition 1). We further show that this error remains invariant under reward shaping (Proposition 2), which allows us to define a shaping reward that directly motivates the final policy-reward co-pretraining algorithm CoPT-AIL.
>
>       OLLIE, on the other hand, is motivated by the closed-form solution of the GAIL discriminator, which depends on the distribution densities of the expert and imitating policies. OLLIE approximates this solution via a Lagrangian method to estimate the densities.
>     * **Learning algorithm.** CoPT-AIL uses a single BC procedure to simultaneously pre-train both policies and rewards (Algorithm 2). On the other hand, OLLIE involves multiple stages: (i) learning an auxiliary reward function, (ii) solving a min-max problem to estimate the distribution densities, which are used to construct the pre-trained reward and (iii) pre-training the policy based on the estimated densities.
>
>          Empirically, we compare CoPT-AIL against OLLIE in our set-up. We observe that CoPT-AIL achieves faster convergence rates than OLLIE (Figure 4 in the revision).

---

> ### Author Response · Authors · 2025-11-21
> **Response (Part II)**
>
> **Question 1**: The paper’s contributions are mainly incremental and overlap with existing analyses in the AIL literature.
>
> **Answer 1**: We respectfully disagree. Prior AIL theory [R1-R4] analyzes AIL initialized with **random policies and rewards**, and focuses on **the iterative online learning process** in AIL. To the best of our knowledge, none of these works (i) theoretically identify or highlight the error from initial policies and rewards, or (ii) provide theoretical guarantees for any pre-training strategy.
>
> **In contrast, this work provides the first systematic theoretical treatment of reward pre-training in AIL,** introducing new theoretical insights, algorithmic principles and theoretical guarantees on reward pre-training that are not covered by existing AIL analyses. We summarize our three key contributions below:
>
> 1. **We develop a theoretical analysis to diagnose why AIL with policy pre-training alone fails (Section 3).** In Proposition 1 (Eq. (4)), we prove that AIL with BC-pretrained policies suffers from a large reward-error term (called "relative policy evaluation error") caused by randomly initialized rewards—an effect that persists regardless of how good the pretrained policy is. Existing theory [R1–R4] only analyzes the iterative learning process and overlooks this initialization error. While some empirical works [R5, R6] observe that policy pretraining can fail, none provide a theoretical explanation. Our work is the first to theoretically diagnose why this failure occurs.
> 2. **We introduce a new policy-reward co-pretraining mechanism based on shaping-reward invariance (Section 4.1).** In Proposition 2, we establish that pretraining toward a shaping reward is sufficient for reducing the "relative policy evaluation error". Combined with the characterization that the log-probability of the expert policy is a shaping reward, we propose a novel policy-reward co-pretraining mechanism, which simultaneously pre-trains rewards and policies through a single learning procedure. However, prior AIL works do not leverage reward shaping to construct a theoretically grounded reward initialization from policies.
> 3. **We provide the first theoretical guarantee for policy-reward co-pretraining improving AIL (Section 4.2).** Theorem 1 demonstrates that CoPT-AIL provably reduces the "relative policy evaluation error", thereby achieving an improved imitation gap bound. To the best of our knowledge, this is the first theoretical guarantee for the efficiency gains of pretraining in AIL. Prior AIL analysis [R1-R4] does not provide guarantees for any pre-training strategy.

---

> ### Author Response · Authors · 2025-11-21
> **Response (Part III)**
>
> **Question 2**: Previous theoretical work on AIL has already highlighted its reliance on reward updates (e.g., Lemma 3.2 and Theorem 4.5 in [1]).
>
> **Answer 2**: We agree that the **reward update** procedure has been widely studied in prior theoretical works such as [1]. Different from those works, this paper makes concrete advances toward a systematic theoretical understanding of **reward pre-training** in AIL, which is absent in prior works. In particular, we (i) theoretically reveal the critical impact of reward pre-training on AIL performance (Section 3), (ii) introduce a new reward pre-training mechanism grounded in shaping-reward invariance (Section 4.1), and (iii) provide theoretical guarantees for the proposed method (Section 4.2); see **Answer 1** for details. We have clarified this distinction and explicitly discussed [1] in the revised manuscript.
>
> **Question 3**: The reward pretraining contribution is not entirely novel, as the connection between policy pretraining and reward learning has already been explored in [2].
>
> **Answer 3**: We respectfully disagree. While [2] attributes the failure of AIL with BC-pretrained policies to random initial rewards **empirically**, our work provides the first **theoretical analysis** of this phenomenon (Section 3). Specifically, we demonstrate that randomly initialized rewards can induce a large relative policy evaluation error, which directly inflates the imitation gap. This offers a theoretical explanation that was absent in prior work.
>
> In addition, as discussed in Answer 2, we propose a novel policy–reward co-pretraining mechanism based on shaping-reward invariance (Section 4.1) and provide the first theoretical guarantee for policy-reward co-pretraining improving AIL (Section 4.2). Together, these contributions constitute the novel **theoretical advances** on reward pre-training presented in this paper.
>
> **Question 4**: The choice of experimental baselines is not fully justified. Given the close connection to OLLIE in [2], which similarly combines pretraining with AIL, including OLLIE as a baseline would have been more appropriate. Since CoPT-AIL emphasizes stronger theoretical analysis, the comparison against OLLIE would have been particularly interesting to demonstrate the practical benefits of the added theory.
>
> **Answer 4**: We did not include OLLIE as a baseline because **our problem setting is fundamentally different from OLLIE.** In particular, we study the standard imitation learning set-up where the learner has access to an expert dataset. In contrast, OLLIE studies imitation learning with a supplementary dataset, where the learner has an expert dataset and an additional supplementary dataset. As stated in [2], one of the key ideas in OLLIE is to utilize the dynamics information in the supplementary dataset.
>
>
> Nonetheless, we can adapt OLLIE to our setting by treating the supplementary dataset as empty. We test OLLIE using [the official open-source implementation](https://github.com/HansenHua/OLLIE-ICML24) on Hopper Stand and Cartpole Swingup. The learning curves are presented in Figure 4 in the revision. We observe that CoPT-AIL achieves faster convergence than OLLIE in both tasks.
>
> **Question 5**: The results in Fig. 2 are not fully convincing and do not clearly substantiate the paper’s claims regarding CoPT-AIL’s superiority over prior AIL approaches.
>
>
> **Answer 5**: **We clarify that Fig. 2 is an ablation study, not the main evidence for superiority over prior AIL methods. Superiority over prior AIL methods is shown in Fig. 1.** Figure 1 compares CoPT-AIL with prior SOTA AIL methods. CoPT-AIL consistently matches or outperforms all prior AIL methods, achieving notably faster convergence rates on tasks such as Cartpole Swingup, Hopper Hop, Hopper Stand, and Finger Spin. These are the results that substantiate the paper’s main empirical claim. Figure 2 is an ablation study, which shows that the proposed reward pre-training mechanism leads to faster and more stable convergence. We will clarify this distinction in the revision.

---

> ### Author Response · Authors · 2025-11-21
> **Response (Part IV)**
>
> **Question 6**: Setting aside the theoretical analysis, what are the main differences between OLLIE in [2] and CoPT-AIL? Are there reasons why one approach should be preferred over the other?
>
> **Answer 6**: The main differences between CoPT-AIL and OLLIE lie in three key aspects: **problem set-up, methodological design principle, and the resultant learning algorithm.**
> 1. **Problem set-up.** As discussed in **Answer** 3, the problem formulations in CoPT-AIL and OLLIE are different. CoPT-AIL studies the standard IL set-up where the learner has access to an expert dataset. In contrast, OLLIE studies IL with a supplementary dataset, where the learner has an expert dataset and an additional supplementary dataset.
> 2. **Methodological design principle.** The design principles behind the two methods differ substantially:
>     * CoPT-AIL is motivated by reducing the relative policy evaluation error, identified through a rigorous theoretical analysis of AIL with BC-pretrained policies (Proposition 1). We further show that this error remains invariant under reward shaping (Proposition 2), which allows us to define a shaping reward that directly motivates the policy-reward co-pretraining algorithm CoPT-AIL.
>     * OLLIE, on the other hand, is motivated by the closed-form solution of the GAIL discriminator, which depends on the distribution densities of the expert and imitating policies. OLLIE approximates this solution via a Lagrangian method to estimate the densities.
> 3. **Learning algorithm.** The resultant learning algorithms of CoPT-AIL and OLLIE are substantially different.
>     * CoPT-AIL uses a single BC procedure to simultaneously pre-train both policies and rewards (Algorithm 2).
>     * OLLIE involves multiple stages: (i) learning an auxiliary reward function, (ii) solving a min-max problem to estimate the distribution densities, which are used to construct the pre-trained reward and (iii) pre-training the policy based on the estimated densities.
>
> Implications for method choice:
> 1. CoPT-AIL is explicitly designed for the standard IL problem and targets to reduce the "relative policy evaluation error" that appeared in the imitation gap. Experiments in **Answer 4** show that CoPT-AIL outperforms OLLIE in this setting. Therefore, CoPT-AIL is preferred when only an expert dataset is available. OLLIE is specifically designed to leverage additional supplementary datasets and is preferred in such scenarios.
> 2. CoPT-AIL requires only a single pre-training procedure, whereas OLLIE involves multiple stages (min-max optimization and policy extraction). Consequently, CoPT-AIL is more computationally efficient, especially for large-parameter models.
>
> In summary, CoPT-AIL is tailored for standard IL with efficient co-pretraining, while OLLIE is suited for scenarios with extra data and more complex density-based pre-training.
>
> **Question 7**: Can you make concrete claims about the quality and quantity of data required for CoPT-AIL to succeed? In particular, how does it perform with suboptimal demonstrations or with only a small number of expert trajectories?
>
> **Answer 7**: For demonstration quality, our work focuses on the standard IL setting, where the expert is assumed to be near-optimal. While imitation learning with imperfect or suboptimal demonstrations is an interesting and important direction, it is beyond the scope of this paper.
> For demonstration quantity, we assume that a considerable number of expert trajectories is available, under which pre-training can provide meaningful performance gains. The exact number of expert trajectories used in our experiments is summarized below:
>
> | Task | Walker Stand  | Walker Run |Finger Spin | Cartpole Swingup |Hopper Hop |Hopper Stand |
> | -------- | -------- | -------- |-------- |-------- |-------- |-------- |
> | Number of Expert Trajectories     | 30     | 50    | 50     | 10     | 10    | 10    |
>
>
> Besides, we evaluate CoPT-AIL with various numbers of expert trajectories. The results are shown in Figure 5 in the revision. We observe that CoPT-AIL exhibits a consistently better performance than prior SOTA AIL methods across different number of expert trajectories.

---

> ### Author Response · Authors · 2025-11-21
> **Part V**
>
> References:
>
> [R1] Lior Shani et al. "Online apprenticeship learning". AAAI 2022.
>
> [R2] Zhihan Liu et al. "Provably efficient generative adversarial imitation learning for online and offline setting with linear function approximation". ICML 2022.
>
> [R3] Luca Viano et al. "Imitation learning in discounted linear MDPs without exploration assumptions". ICML 2024.
>
> [R4] Yilei Chen et al. "Provably efficient off-policy adversarial imitation learning with convergence guarantees". arXiv:2405.16668.
>
> [R5] Rohit Jena et al. "Augmenting gail with bc for sample efficient imitation learning". CoRL 2021.
>
> [R6] Sheng Yue et al. "Ollie: Imitation learning from offline pretraining to online finetuning". ICML 2024.
>
>
>
> ---
> We hope that the above answers can address your concerns satisfactorily. We would be grateful if you could re-evaluate our paper in light of these clarifications. We are also willing to address any further concerns, if possible.

---

> > ### Comment · Reviewer_zfqh · 2025-11-24
> >
> > I would like to thank the authors for their responses. I have a few follow-up remarks:
> >
> > **Contributions**
> >
> > In my initial review, part of my concern was about how Contribution (1) is formulated:
> >
> > > First, we develop a theoretical analysis for AIL with policy pretraining alone, uncovering
> > a critical but theoretically unexplored limitation: the absence of reward pretraining. Our
> > analysis decomposes the imitation gap into two fundamental components: policy error and
> > reward error. While policy pretraining reduces policy error, we demonstrate that reward error
> > remains substantial due to random reward initialization. This creates a notable bottleneck
> > that inflates the overall imitation gap, particularly during early training phases.
> >
> > Given that prior work already shows that the regret of an AIL algorithm can be decomposed into the regret from policy updates and the regret from reward updates (e.g., Yilei Chen et al., Lemma 3.2), does this not make Contribution (1) somewhat straightforward?
> >
> > Once we assume a behavioral cloning initialization for the policy, it is natural that the regret of policy updates is reduced, while the regret of reward updates remains. In that light, it seems almost immediate that reward pretraining becomes a desirable strategy.
> >
> > In my view, the more interesting and non-trivial aspect is the theoretical justification of how reward pretraining is performed, rather than the observation that reward pretraining is needed.
> >
> > **Referring to OLLIE and the other baselines**
> >
> > Do any of the other evaluated baselines exploit a pretraining mechanism of any kind?
> >
> > If I understand correctly, your method performs an offline pretraining phase followed by online finetuning. One could certainly debate whether assuming access to an additional suboptimal dataset is a stronger or weaker assumption than assuming access to an expert dataset. However, I still do not fully follow the reluctance to provide a more direct comparison against OLLIE, which would seem to be the most natural point of reference under this setting.

---

> > > ### Author Response · Authors · 2025-11-26
> > > **Response (Round II-Part I): Clarification on Theoretical Contributions**
> > >
> > > We thank the reviewer for the further engagement.
> > >
> > > ---
> > > ### **Theoretical Justification of "How" to Pre-train Rewards is Precisely the Core of Our Work (Contributions 2 & 3)**
> > >
> > > We fully agree with the reviewer’s assessment that **"the theoretical justification of how reward pre-training is performed"** is the most interesting and non-trivial aspect of this problem.
> > >
> > > **However, we respectfully point out that this specific aspect is precisely the main focus of our Contributions 2 (Section 4.1) and 3 (Section 4.2), which constitute the majority of our theoretical analysis.**
> > >
> > > While the reviewer’s comment focuses heavily on the "triviality" of the motivation (Contribution 1), it appears to overlook that we went far beyond merely identifying the need for reward pre-training: Contributions 2&3 that have been listed in Answer 1 in Response (Part II). We provided the exact algorithmic design and theoretical justification that the reviewer asked for:
> > >
> > > 1. **We introduce a new policy-reward co-pretraining mechanism based on shaping-reward invariance (Contribution 2), which precisely addresses “How Reward Pre-training is Performed”.**
> > >
> > >     The reviewer mentioned that once the need is identified, the solution seems "desirable". However, pre-training a reward that can provably reduce the reward error (“relative policy evaluation error”) is **non-trivial.** To achieve this goal, we establish the invariance of “relative policy evaluation error” under reward shaping (Proposition 2), which implies that pre-training toward a shaping reward is sufficient for reducing the desired error. Combined with the characterization that the log-probability of the expert policy is a shaping reward, **we propose a novel policy-reward co-pretraining mechanism, which simultaneously pre-trains rewards and policies through a single learning procedure.** However, prior AIL works do not leverage reward shaping to construct a theoretically grounded reward initialization from policies.
> > >
> > > 2. **We provide the first theoretical guarantee for policy-reward co-pretraining improving AIL (Contribution 3), which precisely addresses “The Theoretical Justification”.**
> > >
> > >     Theorem 1 demonstrates that CoPT-AIL provably reduces the "relative policy evaluation error", thereby achieving an improved imitation gap bound. This proves *why* our specific method of "how to pre-train reward" mathematically leads to better imitation gap, rather than just intuitively expecting it to work. To the best of our knowledge, **this is the first theoretical guarantee for the efficiency gains of pre-training in AIL.** Prior AIL analysis [R1-R4] does not provide guarantees for any pre-training strategy.
> > >
> > > **Regarding the “Straightforwardness” of Contribution 1:**
> > >
> > > We respectfully clarify that the theoretical result (Proposition 1) in Contribution 1 is **not** a straightforward consequence of Lemma 3.2 in [R4]. While Lemma 3.2 indeed decomposes the bound into $\text{TotalRegret} \leq \sum_{k=1}^K  \text{PolicyReget}(k)  +  \sum_{k=1}^K \text{RewardRegret} (k)$ , the reviewer likely assumes the result is natural because BC policy pre-training trivially reduces **the initial policy regret PolicyRegret(1)**. However, our analysis establishes a **sharper and non-straightforward result**: we prove that BC policy pre-training reduces **the cumulative policy regret $\sum_{k=1}^K \text{PolicyRegret} (k)$** across the entire learning iteration K, which is reflected by the third term in the RHS of Eq.(5) in Proposition 1. This yields a tight theoretical guarantee for AIL with policy pre-training alone, establishing a sharp baseline result for the subsequent theoretical comparison with our approach.
> > >
> > > **Conclusion:**
> > >
> > > The reviewer mentioned that the "theoretical justification of how reward pre-training is performed" is the most valuable direction. **Our paper aligns closely with this standard: it moves beyond the motivation (Contribution 1) and dedicates the core sections to the algorithmic design (Contribution 2) and theoretical guarantee (Contribution 3) on how to perform reward pre-training.** We respectfully request that the reviewer re-evaluate the paper based on the **full scope** of our technical contributions, particularly Sections 4.1 and 4.2, which directly address the problems the reviewer find most critical.

---

> > > > ### Comment · Reviewer_zfqh · 2025-11-26
> > > >
> > > > Thank you for the detailed response.
> > > >
> > > > To clarify, my concern is not about whether the paper addresses how to pre-train rewards, nor about the technical difficulty of Contributions 2 and 3. It is specifically about how Contribution 1 is framed. As currently written, the main conceptual message is that policy pretraining mainly helps the policy part of the bound while leaving the reward part as the bottleneck, and therefore reward pretraining is desirable. Given existing regret decompositions in AIL, this conceptual message feels quite natural, which is why I described it as somewhat straightforward.
> > > >
> > > > I understand that Proposition 1 gives a sharper cumulative regret result, and I accept that this can be technically meaningful. However, at the conceptual level it still reads more as a refined baseline analysis than as a primary new insight. My suggestion is to present Contribution 1 explicitly as a sharpened baseline that prepares the ground for the main novelty in Contributions 2 and 3, rather than positioning it as a central conceptual contribution.

---

> > > > > ### Author Response · Authors · 2025-11-28
> > > > > **Response (Round III-Part I): The Framing of Contribution 1**
> > > > >
> > > > > We sincerely thank the reviewer for the continued engagement and the constructive suggestion.
> > > > >
> > > > > ### **1. Acknowledgment of Technical Consensus on Contributions 1-3.**
> > > > >
> > > > > We greatly appreciate that the reviewer has **explicitly clarified** that:
> > > > >
> > > > > 1. There are **no concerns regarding the technical difficulty and novelty of our core contributions (Contributions 2 and 3), which address "how to pre-train rewards".**
> > > > > 2. Contribution 1 is **technically meaningful** as it provides a "sharper cumulative regret result".
> > > > >
> > > > > ### **2. Revision on the Framing of Contribution 1.**
> > > > >
> > > > > We agree with your constructive suggestion regarding the **framing of Contribution 1**. Positioning it as a "sharpened baseline" rather than a primary conceptual discovery more accurately reflects its role in our theoretical framework.
> > > > >
> > > > > Revision: We have revised **the Contribution 1 paragraph in the Introduction, the discussion following Proposition 1** to explicitly frame Contribution 1 as establishing a sharp theoretical baseline. The text now states that this analysis **"serves a dual purpose: it mathematically isolates the reward error as the precise target for our algorithmic design, and establishes a tight theoretical baseline for the subsequent theoretical comparison."**
> > > > >
> > > > > This revision ensures that Contribution 1 is clearly presented as the foundational step that "prepares the ground" for the main novelty in Contributions 2 & 3, exactly as requested.
> > > > >
> > > > >
> > > > > ### **Conclusion.**
> > > > > Since the reviewer's remaining concern was specifically about the **framing** of Contribution 1 (rather than the technical soundness of Contributions 1-3), and given that:
> > > > >
> > > > > 1. We have **fully incorporated** the reviewer’s suggestion to refine the framing of Contribution 1;
> > > > > 2. The reviewer has **acknowledged the technical depth** of our core contributions;
> > > > >
> > > > > We respectfully believe the current score 2 does not reflect the agreed-upon technical merit of the work. **We kindly request that the reviewer re-evaluate the paper based on our technical contributions and the revised manuscript.**

---

> > > ### Author Response · Authors · 2025-11-26
> > > **Response (Round II-Part II): Clarification on Data Assumptions & Direct Comparison with OLLIE**
> > >
> > > ### **Clarification on Data Assumptions & Direct Comparison Showing the Clear Superiority of CoPT-AIL over OLLIE**
> > >
> > > We would like to clarify a factual misunderstanding regarding our problem setting and point the reviewer to the direct comparison with OLLIE that **is already provided** in our previous response.
> > >
> > > 1. **Correction: We do NOT use "additional suboptimal datasets".**
> > >
> > >     The reviewer mentioned: *"assuming access to an additional suboptimal dataset."* **This is a factual misunderstanding.** Our method strictly follows the **Standard IL setting: we only have access to the expert dataset $\mathcal{D}^{\text{E}}$ and the environment.** In contrast to this paper, OLLIE studies **IL with a supplementary dataset, where the learner has both an expert dataset $\mathcal{D}^{\text{E}}$ and a supplementary dataset $\mathcal{D}^{\text{S}}$,and can interact with the environment.**
> > >
> > > 2. **Direct Comparison against OLLIE (Already Provided).**
> > >
> > >     We have explicitly addressed the comparison with OLLIE in **Answer 4 of Response (Part III)** and **Figure 4 of the Revision**. We respectfully highlight that there was **no reluctance** to compare; the results are already included. We initially provided comparisons on Hopper Stand and Cartpole Swingup in the first revision and have since expanded this to cover **all remaining tasks** in the current version. All results are now available in Figure 4 in the revision.
> > >
> > >     - **Experimental Setup:** OLLIE is originally designed for a setting with supplementary data ( $\mathcal{D}^{\text{E}} + \mathcal{D}^{\text{S}}$). Since our work is in the standard setting ($\mathcal{D}^{\text{E}}$ only), a fair comparison requires adapting OLLIE to the standard setting by treating $\mathcal{D}^{\text{S}}$ as empty.
> > >     - **Results:** As shown in **Figure 4 in the revision**, CoPT-AIL exhibits **substantially faster convergence rates** than OLLIE across 6 tasks.
> > >
> > > **Conclusion:**
> > >
> > > Given that the "additional data" assumption was a misunderstanding and that the requested comparison with OLLIE has been fully provided (demonstrating our method's clear superiority), we respectfully request that the reviewer re-evaluate the paper based on these clarified facts and the empirical evidence.

---

> > > > ### Comment · Reviewer_zfqh · 2025-11-26
> > > >
> > > > Thank you for the clarification.
> > > >
> > > > To be precise, I did not mean to imply that your method uses an additional suboptimal dataset. I understand that CoPT-AIL uses only expert data and environment interactions. My concerns are:
> > > >
> > > > **Baselines and pretraining**. My original question was whether any of the other evaluated baselines use any form of pretraining. This remains unclear. It would help if the paper explicitly stated which baselines do and do not use pretraining.
> > > >
> > > > **Fairness of the OLLIE comparison**. OLLIE is designed for a setting with a supplementary dataset. Evaluating it with an empty supplementary dataset is analogous to comparing Behavioral Cloning to an AIL method but forcing the AIL method to use zero interactions so that both methods have “the same data.” This effectively tests a restricted version of OLLIE outside its intended regime. For this reason, I believe that OLLIE should also be evaluated in its original setting with a non-empty supplementary dataset.

---

> > > > > ### Author Response · Authors · 2025-11-28
> > > > > **Response (Round III-Part II): Baselines with Pre-training & The Fairness of Comparison**
> > > > >
> > > > > We thank the reviewer for the precise clarification. We address the corresponding concerns below.
> > > > >
> > > > > ### **1. Regarding “Baselines and Pre-training”.**
> > > > >
> > > > > To address the reviewer’s concerns, we have clarified and added baselines with policy pre-training.
> > > > >
> > > > > - **Existing Baseline with Pre-training:** In Figure 4 in the revision, **the tested OLLIE utilizes their proposed policy pre-training and reward pre-training components.**
> > > > > - **Added Baselines with Policy Pre-training:** Most prior SOTA AIL methods (e.g., IQLearn and PPIL) do not originally incorporate pre-training. To thoroughly address the reviewer's concern, we have now explicitly added **BC policy pre-training** to strong baselines **IQLearn and PPIL.**
> > > > > - **Results:** As shown in Figure 11 in the revision, **compared with OLLIE and baselines incorporated with BC policy pre-training, CoPT-AIL still exhibits faster convergence rates in the evaluated tasks.** This validates our theoretical analysis that the proposed reward pre-training mechanism provides notable efficiency gains.
> > > > >
> > > > >
> > > > > ### **2. Regarding "Fairness of OLLIE Comparison".**
> > > > >
> > > > >
> > > > > We respectfully disagree with the suggestion to evaluate OLLIE using a non-empty supplementary dataset $\mathcal{D}^{\text{S}}$ in our setting. A fundamental principle of rigorous scientific evaluation is that **a fair comparison requires all methods to have access to the identical set of information.**
> > > > >
> > > > > Our method operates under the Standard IL setting (access to $\mathcal{D}^{\text{E}}$ only), whereas OLLIE is originally designed for a setting with additional supplementary data ( $\mathcal{D}^{\text{E}}+\mathcal{D}^{\text{S}}$). Therefore, the only valid way to compare them fairly is to adapt OLLIE to the standard IL setting by treating $\mathcal{D}^{\text{S}}$ as empty. **Notably, the original OLLIE paper explicitly supports this adaptation**, **stating ”If no supplementary data is provided, then $\mathcal{D}^{\text{S}} = \phi$.”**
> > > > >
> > > > > We have already performed this strict, fair comparison in the revision. As shown in **Figure 4 in the revision**, CoPT-AIL exhibits **substantially faster convergence rates** than OLLIE across 6 tasks. Comparing our method against OLLIE with non-empty $\mathcal{D}^{\text{S}}$ would constitute an unfair "apples-to-oranges" comparison, as it would grant the baseline privileged information unavailable to our method.
> > > > >
> > > > > **Finally, we emphasize that OLLIE does not diminish our contribution due to fundamentally different settings.** While we investigate how to extract high-quality reward initialization solely from the **expert dataset $\mathcal{D}^{\text{E}}$** in Standard IL, OLLIE focuses on leveraging **additional supplementary data alongside expert data ($\mathcal{D}^{\text{E}}+\mathcal{D}^{\text{S}}$)**. This reliance on external resources places OLLIE in a completely different data regime, and the utilization of such supplementary data is explicitly **out of the scope of this work**. Furthermore, beyond the data setting, our approach also fundamentally differs from OLLIE in both its **methodological design principles** and the **resultant learning algorithm**, as detailed in Answer 6.
> > > > >
> > > > >
> > > > > ### **Conclusion.**
> > > > >
> > > > > We have fully addressed the concern regarding baselines by **adding BC-pretrained versions of IQLearn and FILTER, consistently demonstrating the superiority of CoPT-AIL.** Furthermore, we have clarified that **our comparison with OLLIE follows the strict principle of fair data access.** With these experimental concerns resolved and the fairness of our comparison established, **we kindly request that the reviewer re-evaluate the paper based on the updated empirical evidence and clarifications.**

---

### Official Review · Reviewer_YAjk · 2025-10-31

**Soundness:** 3
**Presentation:** 2
**Contribution:** 2
**Rating:** 4
**Confidence:** 4

**Summary:**

The paper tackles the failure mode of BC pre-training in adversarial imitation learning (AIL), where the AIL agent barely benefits from BC pretraining, if at all. This phenomemon has been observed and reported in prior works, but without identifying an actionable cause nor carrying out a rigourous theoretical study. The paper addresses this overlooked gap in the literature and idendifies the culprit as being the lack of reward pre-training in AIL: the reward starts from random initialization. They augment the AIL algorithm with a reward pre-training step, such that the policy and reward are co-pre-trained, hence naming their approach AIL with Policy-Reward Co-Pretraining (CoPT-AIL). Theoretical results show a tighter imitation gap. Empirical results show that CoPT-AIL can outperform standard AIL methods.

**Strengths:**

* The paper targets a crucial desideratum of AIL: taking advantage of potentially limited expert data to squeeze as much performance as possible while remaining offline.
* The detailed contribution rundown in the introduction is appreciated. This could be improved by indicating, for each point at least, where in the paper they are treated.
* The regret bound the authors end up with have are easy to interpret. For example, the new “relative policy evaluation error” bound decreases as 1/N as the number of expert trajectories N increases.
* Despite the weaknesses of the experimental setup, the proposed "fix" to BC-pretrained AIL is so simple that there is little reason for any practitioner not to use it.

**Weaknesses:**

* The authors mention "the telescoping argument" as a discovered property; it is not an incidental property but the reason reward shaping is constructed as such. It might be worth writing about the design principle behind the definition of reward shaping earlier in the main text or with more emphasis, especially considering how central it is to the solution developed in the paper.
* Experiments: going beyond DMC would be a big plus; but even in DMC, there are tasks like humanoid-walk/run, quadruped, and dog that could all be valuable additions. While the authors treat 6 environments, those are 6 tasks from 4 different environments (i.e. only 4 different underlying simulated robotics models). The environment with the highest degrees of freedom (leading to highest state space and action space dimensionalities) is the bipedal walker. A quadruped and/or a humanoid would me more convincing.
* When it comes to the practical use of the algorithm, it would be useful to see whether there is some “unlearning” happening in the reward, e.g. by showing how the reward gradients are behaving, or how aligned the BC and AIL gradients are, which would give empirical evidence that, with CoPT-AIL, BC and AIL are not working against each other anymore. In Walker-Run for example (the most complex task treated in the paper), there is a ~50k step initial phase where CoPT-AIL is unable to accumulated any reward when 3 baselines can. How I interpret this initial plateau at near-zero return where nothing happens: the reward pre-training with the log-probability of the BC-pre-trained policy as reward leads the reward to be peaked at its maximal value on the expert samples, and zero anywhere outside the support of the expert distribution, with very abrupt changes in value as we go in and out of the support. This is a tedious signal to optimize via RL (sparse) a priori. The typical AIL agent (especially in the off-policy setting,  cf. “Lipschitzness is all you need” by Blondé et al 2022) would employ a gradient penalty regularizer to smooth out the reward signal and make learning possible. The authors seem to do something akin to such penalization in their practical implementation (page 22). In L1159-1165, the authors introduce a reward regularizer to “improve the stability of reward training”. What the exponential-based regularizer does: the term penalizes very low or negative reward values, and suppresses excessively large ones, since exponentiation smooths the gradient and bounds the value to a small positive range. I would therefore be useful to show and discuss the reward landscape that the co-pre-training creates, which would at least answer the question of why there is a flat performance plateau at the very start of the learning curve for Walker-Run, only for CoPT-AIL. Note: using the reward smoothing regularlizer also in the reward pre-training phase might solve the issue already, since it would allow AIL to start with a non-random yet smooth reward landscape.

**Questions:**

* With CoPT-AIL, the policy and reward are co-pre-trained (first the policy, then the reward). By the end of the pre-training phases, Q is still randomly initialized. Would it not make even further progress toward your goal of avoiding any negating effect to also pre-train Q? For example, with a method akin to fitted Q iteration? That however assumes that the expert data is available in triplets (state, action, next-state), since the next state is required for the Q target. This condition is a fortiori statisfied if the expert trajectories are available whole and ordered, without subsampling.
* L95-96: aligning with how Ross and Bagnell 2010 (“Efficient Reductions in IL”) defined their non-stationary policies, I think it would be clearer for the authors to qualify the policies, dynamics, and reward structures introduced as non-stationary in the text, grounding the exposition in the pre-existing literature and improving precision. Do the concepts introduced by the authors not align with those in Ross and Bagnell 2010?
* The point made by the authors about the presence of a difference of value with an untrained reward in the bound of Prop 1 is sound. That being said, would it not be best to leave the second term of what the authors call “reward error” out of the curly brace? The same goes for the very last term of the bound.
* How many expert demonstrations were in use? (In the appendix the authors write how many expert demonstrations were collected from the experts, but it is unclear how many you used for the experiments shown on the plots.)
* How does the method fare w.r.t. various number of available demonstrations?
* Is the relative policy gap increasingly reduced compared to the baselines, as the developed theory would dictate?
* It would be interesting to see how CoPT-AIL reacts to different policy and reward architectures, considering how sensitive RL agents generally are to their reward landscape. For example, a reward designed from the error of a powerful diffusion model would, by its representational capacity, probably overfit the reward signal to the pre-training reward learning signal CoPT-AIL introduce, compared to a weaker model. I that process makes the reward landscape particularly non-smooth and peaked, AIL would have to deal with an initially sparse reward function, which would probably be more tedious to deal with than a randomly initialized reward.

Style, typos, suggestions:
* The shorthand “relative policy evaluation error” is rather confusing, as it is part of the reward error, which is distinct from the policy error. It could simply be called reward error if, in Prop. 1, the authors would call the first term of the bound the (scaled) reward error and the second term of the bound the “approximation error” since it grows to 0 as the dataset size tends to infinity.
* The authors should add what the dotted horizontal lines represent in the plot legends .
* [minor] Exposition is very clear; first part of 4.1 however is unnecessarily long for how well-known its contents are.

---

> ### Author Response · Authors · 2025-11-21
> **Response (Part I)**
>
> Thank you for taking the time to review and for your insightful comments. Below, we begin with a general response to the key issues raised, before presenting our specific point-by-point answers.
>
> **General Response**:  We identify two major themes in the reviewer’s comments: **(1) potential negative effects of reward pre-training** and **(2) completeness of the experimental evaluation**.
>
> 1. **Potential negative effects from reward pre-training (Questions 3, 4, 11).**
>     * **Possible "unlearning" during reward pre-training and fine-tuning.**
>       The reviewer is concerned about whether pre-training could inadvertently cause “unlearning” during subsequent fine-tuning. Our empirical analysis indicates that this is not the case. **As shown in Figure 7 in the revision, the gradient cosine similarity between the pre-training and fine-tuning objectives consistently remains high, demonstrating alignment rather than conflict between the two phases.** Additionally, we track the reward gap between expert data and replay buffer data throughout the entire reward learning process. The curves are shown in Figure 8 in the revision. **When transitioning from reward pre-training to reward fine-tuning, the reward gap stays at its previously high level rather than decreasing, indicating that no noticeable unlearning occurs during the switch.** Together, these results suggest that the two stages reinforce each other and that no noticeable unlearning occurs. Details are provided in Answer 3.
>     * **Potential non-smoothness introduced by reward pre-training.**
>       The reviewer also raises the possibility that pre-training could introduce non-smoothness in the reward landscape. To examine this, we compare the Lipschitz coefficients of reward models trained with and without pre-training. As shown in Figure 9 in the revision, **the coefficients are comparable across both settings, indicating that pre-training does not meaningfully degrade smoothness.** Moreover, standard regularization methods (e.g., gradient penalties) can further address potential non-smoothness. Please refer to **Answers 4 & 11** for details. Overall, reward pre-training offers a beneficial initialization without introducing notable negative side effects.
> 2.  **Experimental completeness (Questions 2, 8, 9, 10).**
>      The reviewer also asks for additional empirical evidence along four axes: (1) performance on more tasks, (2) the number of demonstrations used, (3) robustness under varying demonstration counts, and (4) empirical evaluation of the “relative policy evaluation error.” We thoroughly address these points by evaluating two additional tasks (Figure 3 in the revision), reporting the number of expert trajectories (Table 1 in the revision), conducting experiments across different demonstration counts (Figure 5 in the revision), and empirically measuring the "relative policy evaluation error" (Table 7 in the revision). Please refer to the corresponding answers for full details.
>
> **Question 1**: The authors mention "the telescoping argument" as a discovered property; it is not an incidental property but the reason reward shaping is constructed as such. It might be worth writing about the design principle behind the definition of reward shaping earlier in the main text or with more emphasis, especially considering how central it is to the solution developed in the paper.
>
>
> **Answer 1**: We agree with the reviewer’s observation. The telescoping structure is indeed the fundamental design principle behind reward shaping, rather than an incidental property. Our paper does not claim to introduce this structure; instead, we build upon it to show how pre-training a shaping reward can reduce the "relative policy evaluation error" in AIL by ensuring that policy value differences remain invariant (Proposition 2). We already discussed this mechanism in the definition of reward shaping in the revision.
>
>
>
> **Question 2**: Experiments: going beyond DMC would be a big plus; but even in DMC, there are tasks like humanoid-walk/run, quadruped, and dog that could all be valuable additions. While the authors treat 6 environments, those are 6 tasks from 4 different environments (i.e. only 4 different underlying simulated robotics models). The environment with the highest degrees of freedom (leading to highest state space and action space dimensionalities) is the bipedal walker. A quadruped and/or a humanoid would me more convincing.
>
> **Answer 2**: Following the reviewer’s suggestion, we have added experiments on two more challenging DMC tasks: Quadruped Run and Cheetah Run. The updated learning curves are provided in Figure 3 in the revised submission. We observe that CoPT-AIL consistently converges faster than prior state-of-the-art AIL baselines on both tasks. These additional results further support our theoretical analysis, demonstrating that the proposed reward pre-training mechanism leads to a smaller imitation gap.

---

> ### Author Response · Authors · 2025-11-21
> **Response (Part II)**
>
> **Question 3**: When it comes to the practical use of the algorithm, it would be useful to see whether there is some “unlearning” happening in the reward, e.g. by showing how the reward gradients are behaving, or how aligned the BC and AIL gradients are, which would give empirical evidence that, with CoPT-AIL, BC and AIL are not working against each other anymore.
>
>
>
> **Answer 3**: We first clarify that CoPT-AIL does not introduce an additional reward pre-training stage. After policy pre-training, we directly initialize the reward model with the pre-trained policy through $r_{\phi} (s, a) = \log (\pi_{\phi} (a|s))$, where $\pi_{\phi}$ is an independent copy of the pre-trained policy. Thus, reward "pre-training" in CoPT-AIL is exactly the BC policy pre-training process.
>
>
> Regarding the reviewer’s question, we argue that **under CoPT-AIL, the BC and AIL objectives for reward learning are inherently aligned.** We support this claim from two complementary perspectives.
> 1. **Under CoPT-AIL, the BC and AIL reward gradients are aligned.** The BC and AIL reward objectives are
> \begin{align*}
> \mathcal{L}^{\text{BC}} (\phi) =  \mathbb{E}_{(s, a) \sim \mathcal{D}^{\text{E}}} [\log (\pi\_{\phi} (a|s))] = \mathbb{E}\_{(s, a) \sim \mathcal{D}^{\text{E}}} [r\_{\phi} (s, a)],  \mathcal{L}^{\text{AIL}} (\phi) = \mathbb{E}\_{(s, a) \sim \mathcal{D}^{\text{E}}} [r\_{\phi} (s, a)] - \mathbb{E}\_{(s, a) \sim \pi^k} [r\_{\phi} (s, a)].
> \end{align*}
> Here $\pi^k$ is the current policy. The first term of $\mathcal{L}^{\text{AIL}} (\phi)$ is exactly $\mathcal{L}^{\text{BC}} (\phi)$. The second term stems from the adversarial mechanism of AIL, encouraging the reward to assign lower values to non-expert behaviors. Hence, the BC gradient is not competing with AIL—it is literally a component of the AIL gradient. This implies that the BC and AIL reward objectives are intrinsically compatible in CoPT-AIL.
> Empirically, we compute the gradient cosine similarity between BC and AIL reward gradients over the course of training; see Figure 7 in the revision. The similarity remains consistently above 0.8 (with a maximum possible value 1), indicating strong alignment rather than conflict between the two objectives.
> 2. **Under CoPT-AIL, the BC and AIL reward learning procedures consistently separate expert and non-expert data.** Ultimately, reward learning aims to produce a reward function that assigns high values to expert data and low values to non-expert data. To verify that the BC and AIL objectives contribute synergistically to this goal, we track the reward gap between expert data and replay buffer data $\mathbb{E}\_{(s, a) \sim \mathcal{D}^{\text{E}}} [r\_{\phi} (s, a)] - \mathbb{E}\_{(s, a) \sim \mathcal{D}^{\text{replay}}} [r\_{\phi} (s, a)]$ throughout the entire reward learning process; see Figure 8 in the revision. The first 100K reward gradient steps correspond to the reward pre-training procedure while the remaining gradient steps correspond to the reward fine-tuning procedure. Importantly, when transitioning from pre-training to fine-tuning, the reward gap stays at its previously high level rather than decreasing, indicating that no noticeable unlearning occurs during the switch. This further confirms that the BC and AIL reward objectives operate synergistically, not adversarially.
>
> Together, these theoretical and empirical results demonstrate that CoPT-AIL ensures the BC and AIL reward learning objectives reinforce each other, rather than inducing any form of "unlearning".

---

> ### Author Response · Authors · 2025-11-21
> **Response (Part III)**
>
> **Question 4**: In Walker-Run for example (the most complex task treated in the paper), there is a ~50k step initial phase where CoPT-AIL is unable to accumulated any reward when 3 baselines can. How I interpret this initial plateau at near-zero return where nothing happens: the reward pre-training with the log-probability of the BC-pre-trained policy as reward leads the reward to be peaked at its maximal value on the expert samples, and zero anywhere outside the support of the expert distribution, with very abrupt changes in value as we go in and out of the support. This is a tedious signal to optimize via RL (sparse) a priori. The typical AIL agent (especially in the off-policy setting, cf. “Lipschitzness is all you need” by Blondé et al 2022) would employ a gradient penalty regularizer to smooth out the reward signal and make learning possible. The authors seem to do something akin to such penalization in their practical implementation (page 22). In L1159-1165, the authors introduce a reward regularizer to “improve the stability of reward training”. What the exponential-based regularizer does: the term penalizes very low or negative reward values, and suppresses excessively large ones, since exponentiation smooths the gradient and bounds the value to a small positive range. I would therefore be useful to show and discuss the reward landscape that the co-pre-training creates, which would at least answer the question of why there is a flat performance plateau at the very start of the learning curve for Walker-Run, only for CoPT-AIL. Note: using the reward smoothing regularlizer also in the reward pre-training phase might solve the issue already, since it would allow AIL to start with a non-random yet smooth reward landscape.
>
>
>
> **Answer 4**: We appreciate the reviewer’s careful analysis of the Walker-Run learning curve and the hypothesis that reward pre-training might produce a non-smooth reward landscape, potentially explaining the short plateau at the beginning of training.
>
> 1. **Evaluating smoothness of the reward landscape.** To evaluate this concern, we directly measure the Lipschitz coefficient of the reward model (with respect to the state–action input) in Walker-Run for two methods: with reward pre-training and without reward pre-training. The results are shown in Figure 9 in the revised version. **We find that the Lipschitz coefficients in the two settings are highly comparable, indicating that reward pre-training does not introduce additional non-smoothness into the reward landscape. Moreover, the reward model under CoPT-AIL maintains a relatively small Lipschitz coefficient throughout online training.** This is likely aided by the exponential-based regularizer highlighted by the reviewer, which helps stabilize reward updates. Moreover, standard regularization methods (e.g., gradient penalties) can further address potential non-smoothness. Overall, we believe that reward pre-training offers a beneficial initialization without introducing notable non-smoothness.
> 2. **Investigating the cause of the initial plateau.** To understand whether the plateau originates from the reward initialization or from properties of the demonstrations themselves, we repeat the Walker-Run experiments using a newly resampled set of expert trajectories (same quantity as in the original setting). The resulting learning curves are shown in Figure 10 in the revision. In this new run, the initial plateau disappears for CoPT-AIL, while all methods exhibit substantially more consistent early learning behavior. This indicates that the plateau observed in our original figure is primarily attributable to the specific realization of the expert dataset rather than the reward pre-training mechanism.
>
> Taken together, these experiments suggest that (i) reward pre-training does not produce notable non-smoothness in the reward landscape, and (ii) the short plateau in Walker-Run is dataset-specific rather than an inherent limitation of CoPT-AIL.

---

> ### Author Response · Authors · 2025-11-21
> **Response (Part IV)**
>
> **Question 5**: With CoPT-AIL, the policy and reward are co-pre-trained (first the policy, then the reward). By the end of the pre-training phases, Q is still randomly initialized. Would it not make even further progress toward your goal of avoiding any negating effect to also pre-train Q? For example, with a method akin to fitted Q iteration? That however assumes that the expert data is available in triplets (state, action, next-state), since the next state is required for the Q target. This condition is a fortiori statisfied if the expert trajectories are available whole and ordered, without subsampling.
>
> **Answer 5**: We agree with the reviewer’s insight. With a pre-trained reward, one can indeed pre-train a Q-function via fitted Q-iteration if the expert data is in triplets (state, action, next-state). Recent work [R1] in generalist robot learning has similarly explored value-function pre-training and demonstrated good performance. In our paper, we focus on reward pre-training because our theoretical analysis demonstrates that the reward error is the primary source of imitation gap in AIL. Nonetheless, incorporating Q-function pre-training is a natural and promising extension. Exploring this direction is an excellent suggestion and something we plan to investigate in future work.
>
>
>
>
>
> **Question 6**: L95-96: aligning with how Ross and Bagnell 2010 (“Efficient Reductions in IL”) defined their non-stationary policies, I think it would be clearer for the authors to qualify the policies, dynamics, and reward structures introduced as non-stationary in the text, grounding the exposition in the pre-existing literature and improving precision. Do the concepts introduced by the authors not align with those in Ross and Bagnell 2010?
>
> **Answer 6**: In the revision, we now explicitly state that the policy, transition dynamics, and reward functions introduced in our formulation are non-stationary, consistent with the terminology used in Ross & Bagnell (2010).
>
>
>
>
> **Question 7**: The point made by the authors about the presence of a difference of value with an untrained reward in the bound of Prop 1 is sound. That being said, would it not be best to leave the second term of what the authors call “reward error” out of the curly brace? The same goes for the very last term of the bound.
>
> **Answer 7**: Thanks for your suggestion. We have accordingly modified this part in the revision.
>
> **Question 8**: How many expert demonstrations were in use?
>
> **Answer 8**: The number of expert trajectories in use is listed as follows.
>
>
> | Task | Walker Stand  | Walker Run | Cartpole Swingup |Hopper Hop |Hopper Stand |Finger Spin |
> | -------- | -------- | -------- |-------- |-------- |-------- |-------- |
> | Number of Expert Trajectories     | 30     | 50    | 10     | 10    | 10    |50     |
>
> **Question 9**: How does the method fare w.r.t. various numbers of available demonstrations?
>
> **Answer 9**: We evaluate CoPT-AIL with various numbers of expert trajectories. The results are shown in Figure 5 in the revision. We observe that CoPT-AIL exhibits a consistently better performance than prior SOTA AIL methods across different number of expert trajectories.
>
> **Question 10**: Is the relative policy gap increasingly reduced compared to the baselines, as the developed theory would dictate?
>
> **Answer 10**: We evaluate the "relative policy evaluation error" of the pre-trained reward in CoPT-AIL versus the randomly initialized reward in the baseline methods. Recall that the "relative policy evaluation error" is defined as $(V^{\pi^{\text{E}}}\_{r^\star} - V^{\pi^{1}}\_{r^\star}) - (V^{\pi^{\text{E}}}\_{r} - V^{\pi^{1}}\_{r})$. Here we use the ground-truth reward defined in DMC tasks as $r^\star$ and approximate the policy value through Monte Carlo estimation using 20 trajectories. Below, we report the normalized "relative policy evaluation error" divided by the horizon length.
>
> | Task | Walker Stand  | Walker Run | Cartpole Swingup |Hopper Hop |Hopper Stand |Finger Spin |
> | -------- | -------- | -------- |-------- |-------- |-------- |-------- |
> |CoPT-AIL    |    1.51 |  1.49 |  -10.77| 0.63  | 1.84| 1.93|
> | Baseline   | 56.76   | 52.44 | 2.69  | 35.21 | 7.30| 27.66 |
>
> Across all six tasks, the relative policy evaluation error under CoPT-AIL is substantially smaller than under the baseline, clearly supporting our theoretical prediction.

---

> ### Author Response · Authors · 2025-11-21
> **Response (Part V)**
>
> **Question 11**: It would be interesting to see how CoPT-AIL reacts to different policy and reward architectures, considering how sensitive RL agents generally are to their reward landscape. For example, a reward designed from the error of a powerful diffusion model would, by its representational capacity, probably overfit the reward signal to the pre-training reward learning signal CoPT-AIL introduce, compared to a weaker model. I that process makes the reward landscape particularly non-smooth and peaked, AIL would have to deal with an initially sparse reward function, which would probably be more tedious to deal with than a randomly initialized reward.
>
> **Answer 11**: We agree that examining CoPT-AIL under different architectures, including diffusion-based rewards, is an interesting direction. Regarding the concerns about overfitting and non-smooth reward landscapes, standard regularization techniques such as gradient penalty can effectively mitigate these issues during pre-training. For example, DiffAIL [R2] successfully employs diffusion models for reward parameterization and uses gradient penalty to stabilize training and avoid reward overfitting in the online stage. Similar regularization strategies, such as gradient penalties or weight decay, are expected to address the same concerns when using high-capacity architectures like diffusion models in CoPT-AIL’s pre-training stage.
>
> In addition, diffusion models naturally capture multi-modal distributions [R3, R4]. When used to parameterize rewards, they can assign high values to diverse optimal actions, enabling the reward function to reflect a multi-modal reward structure. This property is particularly valuable for preserving behavioral diversity in imitation learning [R5].
>
> **Question 12**: The shorthand “relative policy evaluation error” is rather confusing, as it is part of the reward error, which is distinct from the policy error. It could simply be called reward error if, in Prop. 1, the authors would call the first term of the bound the (scaled) reward error and the second term of the bound the “approximation error” since it grows to 0 as the dataset size tends to infinity.
>
> **Answer 12**: Thanks for your suggestion. We have modified the shorthand in the revision.
>
> **Question 13**: The authors should add what the dotted horizontal lines represent in the plot legends.
>
> **Answer 13**: The dotted horizontal lines represent the return of the expert policy. We have refined this point in the revision.
>
>
> **Question 14**: Exposition is very clear; first part of 4.1 however is unnecessarily long for how well-known its contents are.
>
> **Answer 14**: We have revised the first part of Section 4.1 according to your suggestion.
>
>
> References:
>
> [R1] Ali Amin et al. "$\pi^*_{0.6}$: a VLA That Learns From Experience". 2025.
>
> [R2] Bingzheng Wang et al. "DiffAIL: Diffusion Adversarial Imitation Learning". AAAI 2024.
>
> [R3] Yang Song and Stefano Ermon. "Generative Modeling by Estimating Gradients of the Data Distribution". NeurIPS 2019.
>
> [R4] Cheng Chi et al. "Diffusion Policy: Visuomotor Policy Learning via Action Diffusion". RSS 2023.
>
> [R5] Yunzhu Li et al. "InfoGAIL: Interpretable Imitation Learning from Visual Demonstrations". NeurIPS 2017.
>
>
> ---
>
> We hope that the above answers can address your concerns satisfactorily. We would be grateful if you could re-evaluate our paper based on the above responses. We are also willing to address any further concerns, if possible.

---

> ### Author Response · Authors · 2025-11-28
> **Follow-up on Rebuttal: Evidence on Reward Pre-training Effects and Extended Experiments**
>
> Dear Reviewer YAjk,
>
> As the discussion period is drawing to a close, we would like to gently follow up to ensure that our response and the revised manuscript have fully addressed your concerns.
>
> We have made significant efforts to provide the additional empirical evidence you requested. Specifically:
>
> - **Regarding Potential Negative Effects from Reward Pre-training (Questions 3, 4, 11):** You raised valid concerns about "unlearning" and "non-smoothness" in reward pre-training.
>     - **Evidence against Unlearning:** In the revision, we have added **Figure 7** (gradient cosine similarity) and **Figure 8** (reward gap analysis), which explicitly demonstrate that the pre-training objective aligns with the fine-tuning objective, confirming that no noticeable unlearning occurs.
>     - **Evidence of Smoothness:** In the revision, we have added **Figure 9,** demonstrating that the Lipschitz coefficients of the pre-trained reward model remain comparable to random initialization, proving that smoothness is not compromised.
> - **Regarding Experimental Completeness (Questions 2, 8-10):** We have significantly expanded our evaluation to ensure robustness:
>     - **Broader Evaluation:** In the revision, we have incorporated two additional tasks (**Figure 3**), demonstrating the consistent superiority of CoPT-AIL over prior SOTA.
>     - **Detailed Metrics:** We explicitly reported the number of expert trajectories (**Table 1**), conducted robustness tests on demonstration counts (**Figure 5**), and empirically measured the "relative policy evaluation error" (**Table 7**).
>
> We believe these new results strongly support that our reward pre-training method offers a beneficial initialization without introducing notable negative side effects. **As we have addressed the specific issues that led to the initial score, we would greatly appreciate it if you could reconsider your evaluation.**
>
> Best regards,
>
> The Authors

---

### Author Response · Authors · 2025-12-01
**Summary of Rebuttal Discussion (Part I)**

Dear ACs, SACs, and PCs,

We sincerely appreciate your time and effort in handling our submission, especially given the unexpected interruption of the standard rebuttal process.

Before the rebuttal was halted, we had two rounds of discussion with Reviewer RZUf. However, we did not receive any feedback from Reviewers YAjk and uaDi. Thus, we would like to take this opportunity to summarize the current status of the rebuttal to facilitate your final decision-making.

---
### Summary of main contributions
First, we summarize the main contributions of this paper, which we hope will help clarify the context behind the reviewers’ concerns. **This work provides the first systematic theoretical treatment of reward pre-training in adversarial imitation learning (AIL):**

1. **Contribution 1:** We develop a theoretical analysis of AIL with policy pre-training alone, **identifying reward error as the dominant error source** and establishing a sharp theoretical baseline (Section 3).
2. **Contribution 2:** To reduce reward error, we introduce **a principled policy–reward co-pretraining mechanism, CoPT-AIL**, grounded in shaping-reward invariance (Section 4.1).
3. **Contribution 3:** We provide the first **theoretical guarantee** that policy–reward co-pretraining improves imitation-gap bounds (Section 4.2).
---
### Reviewer YAjk
Reviewer YAjk raised concerns in two main areas: (1) the potential negative effects of reward pre-training and (2) the completeness of the experimental evaluation.

1. **Potential negative effects of reward pre-training (Questions 3, 4, 11).**

    Reviewer YAjk raised valid concerns about whether potential “unlearning’’ or “non-smoothness’’ may occur during reward pre-training.

    - **Evidence against unlearning.** In the revision, we have added **Figure 7** (gradient cosine similarity) and **Figure 8** (reward gap analysis), which clearly show that the reward pre-training objective **aligns with** the fine-tuning objective, confirming that **no noticeable unlearning** occurs.
    - **Evidence against non-smoothness.** In the revision, we have added **Figure 9**, demonstrating that the Lipschitz coefficients of the pre-trained reward model remain **comparable to** those from random initialization, indicating that smoothness is not compromised.
2. **Experimental completeness (Questions 2, 8–10).**
    - **Question 2.** In the revision, we have included two additional tasks (**Figure 3**), which further demonstrate the **consistent superiority of CoPT-AIL** over prior SOTA methods.
    - **Question 8.** We have explicitly reported the number of expert trajectories in **Table 1**.
    - **Question 9.** We have conducted robustness tests on varying numbers of demonstrations (**Figure 5**), showing that CoPT-AIL consistently outperforms baselines across different demonstration counts.
    - **Question 10.** We have empirically measured the reward error (“relative policy evaluation error’’) in **Table 7**, validating our theoretical claim that the proposed reward pre-training mechanism reduces reward error.

We believe this empirical evidence strongly supports that **our reward pre-training method provides a beneficial initialization without introducing notable negative side effects**, directly addressing Reviewer YAjk’s concerns.

---

> ### Author Response · Authors · 2025-12-01
> **Summary of Rebuttal Discussion (Part II)**
>
> ### Reviewer zfqh
> The initial review from Reviewer zfqh raised concerns primarily about the novelty of our contributions relative to prior AIL theory and the empirical work OLLIE. After two rounds of discussion, **Reviewer zfqh acknowledged the technical validity of our three main contributions, explicitly clarifying that:**
>
> 1. There are **no concerns regarding the technical difficulty or novelty of our core contributions (Contributions 2 and 3).**
> 2. Contribution 1 is **technically meaningful**, as it provides a “sharper cumulative regret result.”
>
> With the main concern resolved, the remaining issues raised by Reviewer zfqh focus on (1) the framing of Contribution 1 and (2) the baselines that use pre-training.
>
> 1. **Framing of Contribution 1.**
>
>     Reviewer zfqh suggested presenting Contribution 1 as a “sharpened baseline” rather than a primary conceptual discovery. In response, we have revised **the Contribution 1 paragraph in the Introduction** and **the discussion following Proposition 1**. The text now states that this analysis **“serves a dual purpose: it mathematically isolates the reward error as the precise target for our algorithmic design, and establishes a tight theoretical baseline for subsequent theoretical comparison.”**
>
> 2. **Baselines with pre-training.**
>
>     Reviewer zfqh asked whether any evaluated baselines use pre-training. In the revision, we clarified that OLLIE uses both policy and reward pre-training as originally proposed. In addition, we have **explicitly added BC policy pre-training** to the strong AIL baselines **IQLearn** and **PPIL**. **As shown in Figure 11**, even when compared against OLLIE and baselines incorporated with BC policy pre-training, **CoPT-AIL still achieves faster convergence rates in the evaluated tasks.**
>
>     Reviewer zfqh also raised concerns regarding the fairness of the OLLIE comparison. We clarify that **our comparison is fair in that all methods have access to the identical set of information.** Our work considers the **standard IL setting** (access to $\mathcal{D}^{\mathrm{E}}$ only), whereas OLLIE was originally designed for IL with additional supplementary data ($\mathcal{D}^{\mathrm{E}} + \mathcal{D}^{\mathrm{S}}$). Thus, the only fair comparison is to adapt OLLIE to the standard IL setting by treating $\mathcal{D}^{\mathrm{S}}$ as empty. **Importantly, the original OLLIE paper explicitly supports this adaptation**, stating: *“If no supplementary data is provided, then $\mathcal{D}^{\mathrm{S}} = \varnothing$.”*
>
>
> In summary, Reviewer zfqh has **acknowledged the technical depth** of our core contributions. In addition, we have **fully incorporated** the reviewer’s suggestions to refine the framing of Contribution 1, and we have **clarified and expanded** the baselines with policy pre-training to strengthen the experimental evaluation. We believe that our responses satisfactorily address all remaining concerns.

---

> ### Author Response · Authors · 2025-12-01
> **Summary of Rebuttal Discussion (Part III)**
>
> ### Reviewer uaDi
>
> Reviewer uaDi raised concerns in two main areas: (1) theory under function approximation and (2) completeness of the experimental evaluation. Notably, **Reviewer uaDi stated that he/she would consider raising the score to Acceptance if we satisfactorily addressed Questions 1, 2, 4, 6, and 7 (corresponding to Questions 1–5 in the original review).**
>
> 1. **Theory with Function Approximation and Empirical Validation of Theoretical Predictions (Questions 1, 6).**
>
>     Reviewer uaDi asked whether our theoretical analysis can be extended to the function approximation setting and requested empirical validation of the theoretical predictions.
>
>     - **Theory:** We have shown that our theoretical analysis can be extended to the **linear function approximation setting** in continuous spaces (**Answer 1**).
>     - **Empirical validation:** We empirically measured the “relative policy evaluation error’’ (**Answer 6 and Table 7**). CoPT-AIL significantly outperforms the baselines, directly validating our theoretical predictions.
> 2. **Experimental Completeness (Questions 2, 4, 7).**
>     - **Distribution ratio C (Question 2).** Theoretically, we clarified that dependence on this ratio is **intrinsic to the problem structure**, not a result of loose analysis (**Answer 2**). Empirically, we found that C remains at a **moderate value of ≈4.575** (**Answer 2**), indicating that this ratio does not undermine the practical relevance of our theory.
>     - **Task diversity (Question 4).** We added **two new tasks** (Quadruped Run & Cheetah Run), showing consistent improvements of CoPT-AIL over prior SOTA methods (**Figure 3**).
>     - **Sensitivity analysis (Question 4).** We conducted sensitivity analyses on the demonstration count N (**Figure 5**) and the regularization coefficient $\beta$ (**Figure 6**), demonstrating the robustness of CoPT-AIL across different data regimes and hyperparameter settings.
>     - **Implementation details (Question 7).** We clarified that for continuous action spaces, we use a Gaussian policy parameterization to compute log-probabilities (**Answer 7**).
>
> In summary, we believe our responses have thoroughly addressed the concerns raised by Reviewer uaDi, **particularly those that he/she indicated would lead to an Acceptance score once resolved.**
>
> ---
> Our work introduces a principled policy–reward co-pretraining mechanism for imitation learning. Owing to its simplicity, this approach holds strong promise for integration into foundation-model-based generalist robot training frameworks. We believe our responses provide clear evidence addressing the raised concerns, and we would greatly appreciate a thorough evaluation from the area chairs.
>
> Best regards,
>
> Authors of Paper 17861

---

### Meta-Review · Area_Chair_aCNU · 2026-01-10

**Summary:**

There are several concerns from the reviewers about this work. The novelty of the work can be challenged because "previous theoretical work on AIL has already highlighted its reliance on reward updates," and "the reward pretraining contribution is not entirely novel, as the connection between policy pretraining and reward learning has already been explored." Also, OLLIE should serve as a baseline in the expeirments. There is concern that reward pre-training may cause the agent to unlearn good behaviors during the transition to online fine-tuning and have a non-smooth reward landscape. The experiments are narrow, primarily on the DMControl benchmark. Reviewer zfqh raised questions about the novelty of the contributions and suggested framing one of the contributions about reward error as a sharpened baseline rather than a primary discovery.

**Reviewer Concerns:**

During the rebuttal, the authors provide additional experiments showing alignment between pre-training and fine-tuning objectives, which do not show the unlearning phenomenon. They also demonstrate that the smoothness of the reward landscape is not compromised. Going beyond DMControl, the authors have added experiments on two more challenging DMC tasks: Quadruped Run and Cheetah Run. The authors revised their framing theoretical analysis about reward error accordingly. Overall, the authors have addressed the concerns of the reviewers. However, some concerns regarding the technical novelty compared to prior work on AIL, especially OLLIE, are still outstanding. The missing experimental comparison to OLLIE is also outstanding. The AC understands that the problem setting may be different, but OLLIE can be evaluated under a different setting. Given the current form of the work, the AC does not recommend acceptance to ICLR.

**Reviewer Scores:**

This paper received ratings of 2,4,6. Two reviewers may change the score, but I doubt the overall scores will be very positive.

---

### Decision · Program_Chairs · 2026-01-26

Reject